

# A production-tagged aerosol module for earth system models, OsloAero5.3 – extensions and updates for CAM5.3-Oslo

Alf Kirkevåg[1], Alf Grini[1], Dirk Olivié[1], Øyvind Seland[1], Kari Alterskjær[2,3], Matthias Hummel[3], Inger H. H. Karset[3], Anna Lewinschal[4], Xiaohong Liu[5], Risto Makkonen[6], Ingo Bethke[7], Jan Griesfeller[1], Michael Schulz[1], Trond Iversen[1,2]

[1]Norwegian Meteorological Institute, P.O. Box 43, Blindern, 0313 Oslo, Norway
[2]CICERO Center for International Climate Research, 0349 Oslo, Norway
[3]Department of Geosciences, Section for Meteorology and Oceanography, University of Oslo, 1022 Oslo, Norway
[4]Department of Meteorology, Stockholm University, 106 91 Stockholm, Sweden
[5]Department of Atmospheric Science, University of Wyoming, Laramie, Wyoming, 82071, USA
[6]Dept. of Physics, University of Helsinki, P.O. Box 64, Helsinki, Finland
[7]Uni Research Climate, Bjerknes Centre for Climate Research, P.O. Box 7810, 5020 Bergen, Norway

*Correspondence to*: Alf Kirkevåg (alfk@met.no)

**Abstract.** We here document model updates, and present and discuss modelling and validation results, from a further developed production tagged aerosol module, OsloAero5.3, for use in earth system models. The aerosol module has in this study been implemented and applied in CAM5.3-Oslo. This model is based on CAM5.3/CESM1.2 and its own predecessor model version CAM4-Oslo. OsloAero5.3 has improved treatment of emissions, aerosol chemistry, particle lifecycle and aerosol-cloud interactions compared to its predecessor OsloAero4.0 in CAM4-Oslo. The main new features consist of improved aerosol sources, the module now explicitly accounting for aerosol particle nucleation and secondary organic aerosol production, with also new emissions schemes for sea-salt, dimethyl sulphide (DMS) and marine primary organics. Mineral dust emissions are updated as well, adopting the formulation of CESM1.2. The improved model representation of aerosol-cloud interactions now resolves heterogeneous ice nucleation based on black carbon (BC) and mineral dust calculated by the model, and treats activation of cloud condensation nuclei (CCN) as in CAM5.3. Compared to OsloAero4.0 in CAM4-Oslo, the black carbon (BC) mass concentrations are less excessive aloft, with better fit to observations. Near surface mass concentrations of BC and sea-salt aerosols are also less biased, while sulfate and mineral dust are slightly more biased. Although appearing quite similar for CAM5.3-Oslo and CAM4-Oslo, the validation results for organic matter (OM) are inconclusive, since both of the respective versions of OsloAero are equipped with a limited number of OM tracers for the sake of computational efficiency. Any information about the assumed mass ratios of OM to organic carbon (OC) for different types of OM sources is lost in the transport module. Assuming that observed OC concentrations scaled by 1.4 are representative for the modeled OM concentrations, CAM5.3-Oslo with OsloAero5.3 is slightly inferior for the very sparsely available



observation data. Comparing clear-sky column integrated optical properties with data from ground based remote sensing, we find a negative bias in optical depth globally, however not as strong as in CAM4-Oslo, while it is biased high for areas typically dominated by mineral dust emissions. Aerosol absorption has a larger negative bias than the optical depth globally, and is less overestimated in areas where mineral dust is the main contributor to absorption. Globally, the low bias in absorption is smaller

than in CAM4-Oslo. The Ångström parameter exhibits small biases both globally and regionally, suggesting that the aerosol particle sizes are reasonably well represented. Cloud top droplet number concentrations over oceans are generally underestimated compared to satellite retrievals, but seem to be overestimated downwind of major emissions of dust and biomass burning sources. Finally we find small changes in direct radiative forcing at top of the atmosphere, while the cloud radiative forcing due to anthropogenic aerosols is now more negative than in CAM4-Oslo, being on the strong side compared

to the multi-model estimate in IPCC AR5. Although not all validation results in this study show improvement for the present CAM5.3-Oslo version, the extended and updated aerosol module OsloAero5.3 is more advanced and applicable than its predecessor OsloAero4.0, as it includes new parameterizations which more readily facilitate sensitivity and process studies and use in climate and earth system model studies in general.

## 1 Introduction

Humans influence the production of aerosols (microscopic solid and liquid particles suspended in air) in various ways, giving rise to local and regional air pollution. Furthermore, Earth's climate can be influenced by aerosols, either directly through changes to the scattering and absorption of solar radiation, or more indirectly through the effects these particles have on cloud properties and precipitation. Numerical modelling of Earth's climate therefore requires a description of aerosols, where mass and number concentrations and chemical composition as function of size are important properties.

Even without going all the way in calculating how aerosols impact climate, including slow responses and feedbacks through atmospheric and ocean–atmosphere interactions that can be simulated in fully coupled climate models or Earth System Models (ESMs), one may quantify a first order effect on Earth's radiative budget in partly uncoupled model configurations through estimates of the so-called aerosol radiative forcing. It is common to distinguish between the traditional concepts of radiative

forcing (RF) and the effective radiative forcing (ERF), which include rapid adjustments that modify the radiative budget through fast atmospheric and surface changes (IPCC AR5: Boucher et al., 2013; Myhre et al., 2013). ERF from aerosols can furthermore be decomposed into a forcing term due to aerosol–radiation interactions (ERFari), which include the traditional direct effect and semi-direct effects (as rapid adjustments to atmospheric heating by absorbing aerosols), and an aerosol–cloud interactions term (ERFaci) (Boucher et al., 2013), which include the cloud albedo effect (Twomey, 1977) and associated

adjustments in the form of lifetime effects (e.g., Albrecht, 1989). In this study we follow the method outlined by Ghan (2013) for calculating the effective radiative forcing of aerosols, which is decomposed into a direct radiative forcing, a cloud radiative



forcing, and a surface albedo forcing term. In contrast to the terminology used in IPCC AR5, the semi-direct effect is here baked into the cloud radiative forcing term.

Traditionally, mainly two methods have been used to calculate aerosol size and chemical composition. Modal approaches (e.g., Binkowski and Shankar, 1995) approximate the aerosol size distribution as lognormal distributions. Sectional methods (e.g., Bergman et al., 2012) discretize the size distribution into fixed size intervals which have constant properties. In a sectional aerosol module the size distribution does not have to be lognormal or of any other specified shape, and is generally considered to be closer to "first principles".

An alternative "production tagged" aerosol module is used in the atmospheric component (CAM-Oslo) of the Norwegian Earth System model (NorESM), and in various predecessor model versions. This aerosol module has been documented in Kirkevåg et al. (2013) for CAM4-Oslo (NorESM1), and in earlier studies (Kirkevåg et al.,1999; Kirkevåg and Iversen, 2002; Iversen and Seland, 2002; Iversen and Seland, 2003; Kirkevåg et al., 2005, Kirkevåg et al., 2008; Seland et al., 2008). The production tagged method describes a number of "background" lognormal modes. These modes can change their size distribution due to condensation, coagulation and cloud processing. The corresponding aerosol microphysical calculations are performed in a detailed size-resolving model, run offline. A selection of results in terms of bulk properties from these aerosol microphysics calculations are stored in lookup tables, which during the NorESM model simulation provide information about aerosol optical parameters as well as size and composition, where this is needed (for details, see Sect. 2.1 in Kirkevåg et al., 2013). "Production-tagged" refers to the fact that the tracers which change the aerosol size distribution represent their production pathway (e.g., condensation, coagulation, and cloud processing). We will refer to the online aerosol module as OsloAero, and to the offline size-resolving model which produces the look-up tables as AeroTab. Although the aerosol module has been developed over many years and already been used in numerous model versions, it has previously not been given any name or version number. For the purpose of simplicity and clarity in the inter-comparison of the respective module versions, we hereafter denote the OsloAero module described and used by Kirkevåg et al. (2013) OsloAero4.0, and the present version OsloAero5.3. We similarly denote the respective versions of the offline size-resolving look-up table model AeroTab4.0 (Kirkevåg et al., 2013) and AeroTab5.3.

In this work we have ported OsloAero to the Community Atmospheric Model version CAM5.3 (Neale et al., 2012; Liu et al., 2016), so that it exists as an option alongside the CAM modal aerosol modules (MAM3 and MAM7).  We hereafter refer to the atmospheric model including OsloAero5.3 and the AeroTab5.3-produced look-up tables as CAM5.3-Oslo. CAM5.3 is part of the Community Earth System Model version 1.2, CESM1.2 (http://www.cesm.ucar.edu/models/cesm1.2). The Norwegian Earth System Model version based on CESM1.2, which we name NorESM1.2, uses CAM5.3-Oslo instead of CAM5.3, an updated MICOM version based on NorESM1 (Bentsen et al., 2013) instead of POP2 as the ocean model, while the land model CLM4.5, the sea-ice model CICE4, and the coupler CPL7, are all as in CESM1.2. In this study we do not make use of the fully



coupled model system, but prescribe sea-surface temperatures and sea-ice fractions (i.e., an AMIP set-up). In the following discussions we therefore just refer to the model as CAM5.3-Oslo.

CAM5.3-Oslo is after some final updates and tuning planned to be merged with the atmospheric component, CAM6, from the
upcoming    release    of    the    NCAR/DOE    Community    Earth    System    Model,    CESM2
(http://www.cesm.ucar.edu/working_groups/Atmosphere/). This merged version is expected to be the atmospheric component
of NorESM2. NorESM2 is planned to participate in the Coupled Model Intercomparison Project no. 6 (CMIP6). NorESM1.2
(using a further adapted and tuned version of CAM5.3-Oslo) is at the present a fallback version, and may be used in the early
phases of CMIP6 if NorESM2 is not finalized in time. Two versions of NorESM1, NorESM1-M (Bentsen et al., 2013; Iversen
et al., 2013; Kirkevåg et al., 2013) and NorESM1-ME (Tjiputra et al., 2013), contributed with results for CMIP5 and were
analysed together with the other CMIP5-contributing models in IPCC AR5 (Myhre et al., 2013). NorESM1-M is presently
also taking part in the ongoing Precipitation Driver Response Model Intercomparison Project (PDRMIP, see e.g., Samset et
al., 2016; Myhre et al., 2017).

The main purpose of this study is to document the changes in the treatment of aerosols and aerosol-cloud interactions since
the predecessor model version CAM4-Oslo, as well as summarizing the main principles behind the aerosol schemes applied
in earlier and the present model versions. We then evaluate CAM5.3-Oslo's performance with respect to various aerosol and
cloud droplet properties, and present and discuss new estimates of effective radiative forcing, both for comparison with results
from CAM4-Oslo and other CMIP5 models.

The article is organized as follows: Section 2 describes the model components which have changed since Kirkevåg et al.
(2013), with emphasis on the aerosol module. Section 3 describes the model configurations used in this study. Section 4
compares the aerosol and cloud droplet concentrations and optical properties to observations and remote retrievals, as well as
to previous studies, wherever feasible. Section 5 puts the results into a climate context by discussing the effective radiative
forcing due to aerosol–radiation and aerosol–cloud interactions, before presenting the summary and conclusions in Section 6.

## 2 Aerosol model description

OsloAero5.3, as it is implemented in CAM5.3, applies the same method of aerosol activation as Liu et al. (2012), with the
simplifications proposed by Ghan and Easter (2006) that cloud-borne aerosols are not advected, except by vertical turbulent
mixing. An important feature of CAM5.3 is that it includes a general chemical solver (CAM-Chem) as well as a standardized
chemical code pre-processor (MOZART, Emmons et al., 2010), which OsloAero5.3 (unlike earlier versions) makes use of.
The sulfur chemistry is now also as in Liu et al. (2012), except for the DMS + OH addition reaction where 75% of the reaction
product is $SO_2$ (as in Pozzoli et al., 2008), compared to 50% in Liu et al. (2012). However, the treatments of nucleation and



secondary organic aerosols differ, as in many other processes which are specific to CAM5.3-Oslo, i.e. to OsloAero5.3 and AeroTab5.3.

Since Kirkevåg et al. (2013) (CAM4-Oslo), several improvements have been made to OsloAero and AeroTab. These updates will be described in detail in this section, but may be briefly summarized as follows. Aerosol nucleation and secondary organic aerosols have been taken explicitly into account based on Makkonen et al. (2014), with some extensions. Sea-salt emissions and emission sizes have been changed to those of Salter et al. (2015). Dimethyl sulfide (DMS) and oceanic primary organics are now emitted from concentration and wind driven parameterizations (Nightingale et al., 2000; Vignati et al., 2010), and dust emissions are calculated online based on Zender et al. (2003). Aerosol hygroscopicity and a few other microphysical properties have also been changed since CAM4-Oslo. Finally, heterogeneous ice nucleation is implemented based on Wang et al. (2014), which was based on a modified version of the scheme in CAM3-Oslo (Hoose et al., 2010).

## 2.1 The production tagged aerosol module

The "production tagged" aerosol module has been used previously in many studies. The life-cycling component of the online aerosol module we now call OsloAero was first developed and described by Seland et al. (1999) and Iversen and Seland (2002, 2003). The offline size-resolving aerosol model we call AeroTab, including table look-ups and interpolations with respect to aerosol–radiation and aerosol–cloud interaction calculations in OsloAero, was first developed and described by Kirkevåg et al. (1999), and Kirkevåg and Iversen (2002), with some updates by Kirkevåg et al. (2005). Later versions of both components of the production tagged aerosol module as a whole are described by Seland et al. (2008), Kirkevåg et al. (2008), and by Kirkevåg et al. (2013), hereafter referred to as K13. The essential difference to other aerosol module treatments is the division of tracers into "background" and "process" tracers. Background tracers, which are mainly primary emitted particles (nucleation being the exception), form lognormal modes and contribute to the aerosol number concentration. The process tracers change the shape and chemical composition of the initially lognormal background modes. Examples of process tracers are sulfate condensate, sulfate coagulate, sulfate from cloud processing (aqueous phase chemistry in cloud droplets, followed by evaporation), and secondary organic aerosol (SOA) condensate. All tracers that are calculated explicitly are listed in Table 1.

For gas phase and aqueous aerosol chemistry, concentrations of OH, $NO_3$, $O_3$ and $HO_2$ are prescribed (see also Karset et al., 2018) as time-varying climatological 3D monthly-mean fields from simulations with the global stratosphere-troposphere chemistry model CAM-chem v3.5 in the study of Lamarque et al. (2010), representative for conditions in the year 2000. $H_2O_2$ is calculated as in Liu et al. (2012), and depends on the prescribed (monthly averaged) $HO_2$ concentrations.

As soon as the aerosol background modes have changed composition and shape, we refer to them as "mixtures". Because the resulting size-distribution from AeroTab is no longer lognormal and "modes" are traditionally used for aerosol size



distributions which are lognormal, the term mixture is used in order to avoid confusion. The resulting mixtures, which the look-up tables are based on, are given in Table 2. The table shows which tracers are assumed to be background tracers (log-normally distributed at the point of emission or production) and which tracers are purely size and composition modifying. OsloAero calculates how much of each "modifying" tracer should be distributed to a background mode within a time-step.

When that fraction is known, interpolations in the look-up tables (generated by AeroTab) return the optical properties or the best log-normal fit (in terms of modal median radius and standard deviation) of the final dry size distribution of that mode after growth. Assumed standard deviation of the initially log-normal size distributions as well as the accommodation coefficients for each of the mixtures are still as in Table 1 in K13.

Concerning the basic principles behind the production-tagged aerosol module (see K13 and references therein), we may look at it as a three stage process over a time-step in the model. First, during atmospheric transport the background tracers are assigned prescribed sizes (the sizes at the point of emission in Table 2, somewhat augmented to take into account atmospheric growth), and for computational efficiency the size-modifying tracers are lumped together, also with prescribed sizes. The assumed modal parameters, number median radius (NMR) and standard deviation (SIGMA), for the size-modifying tracers

during transport are NMR = 0.04 µm and SIGMA = 1.8 for SO4_A1, and NMR = 0.1 µm and SIGMA = 1.59 for SO4_A2, SO4_AC, OM_AC, BC_AC and SOA_A1. Secondly, when the size distribution resulting from aerosol microphysics is needed, the mass of the size-modifying tracers is distributed onto the different background size modes according to how large the sink is for the tracer in question, estimated online following Kirkevåg et al. (1999). For example, the amount of condensate added to a background mode is proportional to the background mode's condensation sink (prior to growth). Finally, the mass of these

mixture-apportioned tracers are fed into the interpolation code connected to the lookup-tables, giving us estimated sizes and optical properties. The look-up tables have been calculated offline by use of AeroTab5.3, based on the fully size-resolved (with 44 size-bins) solution to the continuity equations for particle number and mass concentrations (Kirkevåg et al., 1999) after aerosol growth. Note that the full size distribution (i.e., number concentration for each size-bin) is not stored in these look-up tables, but rather the subsequent bulk (i.e. size integrated) parameters which are required by the atmospheric model, such as

"single scattering albedo", "asymmetry factor", "mass specific extinction", in addition to log-normal fits to the dry size distributions after growth. Tabulated aerosol optical parameters include the effect of humidity swelling.

Using this technique, we lose information about which sizes were modified by which tracer in the past, since the detailed size information is lumped back into a limited number of tracers before atmospheric transport. However, we gain computational

efficiency since the technique requires fewer transported tracers. The size of the aerosol mixtures, i.e., of background tracers including growth by process tracers, could in principle be estimated by use of the tabulated size parameters for the particle mixtures in the previous time-step. Such a link has not yet been implemented in the model, but is something that should be investigated and tested in future model versions.



## 2.2 Secondary organic aerosols and nucleation

The treatment of secondary organic aerosol (SOA) and nucleation has been much improved since K13, where SOA was simply prescribed as a monthly surface source, and nucleation (sulfate only) was implicitly determined by the amount of available $H_2SO_4$ left after condensation during a model time step. The treatment is now based on Makkonen et al. (2014), herafter

referred to as M14, who implemented emissions of monoterpene and isoprene in a research version of NorESM1-M (see also Boy et al., in prep.). These SOA precursors are oxidized by OH, $O_3$ and $NO_3$.

The chemical reactions and assumed yields (0.15 and 0.05) are given below, with reaction rates (not shown) taken from IUPAC (Atkinson et al., 2004; 2006). These yields are similar to values used in other studies (e.g. Mann et al., 2010; Tsigaridis et al.,

10 2014).

$$monoterpene + O_3 \rightarrow 0.15 \cdot SOAG_{LV} \tag{R1}$$
$$monoterpene + OH \rightarrow 0.15 \cdot SOAG_{SV} \tag{R2}$$
$$monoterpene + NO_3 \rightarrow 0.15 \cdot SOAG_{SV} \tag{R3}$$
$$isoprene + OH \rightarrow 0.05 \cdot SOAG_{SV} \tag{R4}$$
$$isoprene + O_3 \rightarrow 0.05 \cdot SOAG_{SV} \tag{R5}$$
$$isoprene + NO_3 \rightarrow 0.05 \cdot SOAG_{SV} \tag{R6}$$

The idea of separating between $SOAG_{SV}$ and $SOAG_{LV}$ is that the SOA gas (SOAG) tracer labeled "SV" is assumed to be semi-

volatile, with an equilibrium vapor pressure which is too high to contribute to new particle formation but instead goes to condensation. In addition to contributing to condensation, the tracer labeled "LV" is assumed to be low volatile enough to also contribute to particle nucleation and subsequent aerosol growth below the number median radius of the SO4_N/SOA_N mixture (mixture no. 1 in table 2). Only low volatile products are assumed to take part in new particle formation as described by Kulmala et al. (2004). In M14, low volatile products are only assumed to form in the reaction between monoterpene and

$O_3$. This choice is supported by observed correlation of 7–20 nm (in diameter) aerosol growth and monoterpene ozonolysis (Yli-Juuti et al., 2011), as well as the relatively higher yield of Extremely Low Volatility Organic Compounds (ELVOCs) from $O_3$ as compared to OH reaction with monoterpenes (Jokinen et al., 2015). The fractions of monoterpene and isoprene which do not react to form SOA are "lost" from the model, assuming that they form other products which we do not track. This approach is a good way to resolve oxidant-mediated variations in SOA production, and is suitable for global aerosol models

with simplified aerosol precursor chemistry schemes (e.g. Spracklen et al., 2008). We also note that, since the model uses the "offline-oxidant approach", reactions (R1) to (R6) need only to resolve one product (on the right-hand-side of the equations). While methanesulfonic acid (MSA) in K13 was emitted directly into the OM_NI tracer as primary OM, we now also treat MSA as a biogenic VOC which may form SOA, assuming that 20% and 80% of the mass is added to the $SOAG_{LV}$ and $SOAG_{SV}$





tracers, respectively (see Fig. 1). The exact LV to SV ratio is unknown, but some of the MSA is of low enough volatility to contribute to nucleation and subsequent growth (Chen et al., 2017; Willis et al., 2016).

The concentrations of the condensable gases, $H_2SO_4$, $SOAG_{LV}$ and $SOAG_{SV}$, are calculated based on the production rates from

the gas phase chemistry solver MOZART (Horowitz et al., 2002). The solver is configured to use the chemical mechanism used in K13 with the additional reactions for SOA.

Furthermore, only a fraction of the $SOAG_{LV}$ oxidation products (50%, as in M14) is assumed low-volatile enough to nucleate or condense onto nucleation-sized particles, while the remaining fraction and the semi-volatile tracer is allowed to condense

on pre-existing particles. Binary nucleation of $H_2SO_4$ vapor is based on Vehkamaki et al. (2002). Boundary layer nucleation is implemented according to several semi-empirical parameterizations from Paasonen et al. (2010). For the present model version and the simulations in this study we have used Eq. 18 in Paasonen et al. (2010).

After nucleation, particles grow further by condensation of sulfuric acid and organic vapors. Growth of nucleated clusters to

the particle size of the corresponding mixture treated in the model (see Table 2) is based on Lehtinen et al. (2007). The organic vapors available for this transition have been found to be very important for the growth of atmospheric particles (Riipinen et al., 2011; Keskinen et al., 2013).

The condensation sink is known from the surface area of the background aerosols. After the gas phase chemistry is treated in

the model, the concentrations of the condensable gases are set back to their value from the start of the time-step, and the following equation is solved to obtain concentrations at the end of the time step:

$$\frac{dC_{gas}}{dt} = P_{gas} + L_{cond} \cdot C_{gas} - L_{nuc} \cdot C_{gas,} \tag{1}$$

where $L_{cond}$ is the loss rate [s$^{-1}$] for condensation and $L_{nuc}$ is the loss rate [s$^{-1}$] due to nucleation for the condensing gas. Since $L_{nuc}$ is dependent on the concentration we perform one iteration before the equation is solved with an Euler backwards method to obtain the concentration at the end of the time step $C_{gaa,new}$. In the first iteration, $L_{nuc}$ is zero. The resulting gas phase concentration from the first iteration is used to calculate the nucleation rate. When the concentration at the end of the time step has been found with the Euler backwards method, the tendency is calculated as

$$\frac{dC_{gas}}{dt} = (C_{gas,new} - C_{gas,old})/dt, \tag{2}$$



Nucleated particles from $SOAG_{LV}$ and $H_2SO_4$ have much smaller diameters ($d_{nuc}$ ~ 1–3 nm) than the respective aerosol mixture in CAM5.3-Oslo (mixture no. 1 in Table 2), which has a median modal diameter ($d_x$) of 23.6 nm. The smallest particles can either coagulate with the background particles, or grow by condensation of $SOAG_{LV}$ and $H_2SO_4$ until they reach sizes that have longer lifetime with respect to coagulation. The following formula (Eq. 7 in Lehtinen et al., 2007, see also M14), gives

the rate $J_x$ at which particles of size $d_x$ form, growing from nucleation size to that of the corresponding mixture (no. 1) in the model:

$$J_x = J_{nuc} \cdot exp(-\gamma \cdot d_{nuc} \frac{CoagS(d_{nuc})}{GR}) \; ; \; \gamma = \frac{1}{m+1}[(\frac{d_x}{d_{nuc}})^{m+1}-1] \tag{3}$$

Here $J_{nuc}$ is the nucleation rate of $d_{nuc}$ sized particles, CoagS is the coagulation sink, and GR is the rate of particle growth due to the condensation. The factor $\gamma$ is expressed as a function of $d_{nuc}$ and $d_x$, as well as a background size dependent exponent m. Here we simply let m = -1.6 (as in M14), which is a typical value for atmospheric conditions (Lehtinen et al., 2007). The formation rate is in other words determined by the concentration of sulfuric acid and organic vapors available for condensational growth and by the coagulation sink of the newly formed particles onto pre-existing aerosols.

There are four important differences in the SOA treatment compared to M14:

    1.    We close the mass balance both for $H_2SO_4$ and for organic vapors, while M14 put nucleated mass into the model as $H_2SO_4$, thus allowing sulfur mass to be produced by organic vapors. Unlike the M14 study, which focused on changes in aerosol life cycling but not on radiative effects of SOA, the look-up tables for optics and sizes with respect to aerosol-radiation

and aerosol-cloud interactions are now also taking into account SOA.

    2.    We add the non-nucleated vapor as condensate. The condensate is only added through condensation/diffusion on pre-existing particles and it does not produce new particles. In M14, non-nucleated vapor was added to the tracer representing primary organics. Since primary organics is a background tracer in OsloAero5.3, increasing primary organic mass also increases aerosol number concentration. In the updated treatment condensate does not increase particle number concentrations

(unless it leads to increased nucleation rates).

    3.    M14 assumed secondary organic aerosol formation only from monoterpenes. In this work both monoterpenes and isoprene are assumed to produce SOA mass. Still only monoterpene ozonolysis products are allowed to produce new particles by nucleation (via $SOAG_{LV}$).

    4.    We now also make use of interactive emissions of SOA precursors from CLM4.5, using the MEGAN-v2.1 (Guenther

et al., 2012) algorithm instead of reading them in from file. This allows us to study the effects of a changing climate on SOA formation and facilitates feedback studies. We lump 21 monoterpene species (myrcene, sabinene, limonene, 3-carene, t-β-ocimene, β-pinene, α-pinene, dimethyl styrene, p-cymene, o-cymene, α-phellandrene, α-thujene, α-terpinene, γ-terpinene,



terpinolene, β-phellandrene, camphene, bornene, α-fenchene, allo-ocimene and cis-β-ocimene) into one atmospheric monoterpene tracer.

The main advantages of the new treatment of SOA in this study, compared to M14, are that the atmospheric composition influences the aerosol size distribution and particle number, as well as its optical properties, that SOA is allowed to form outside the boundary layer, and that the use of interactive biogenic volatile organic compound (BVOC) emissions, including MSA from the ocean surface, facilitates studies of the effects of climate change on SOA formation, as well as on subsequent feedbacks.

## 2.3 Aerosol microphysics

Diffusion coefficients for condensable gases have been calculated based on Eq. 11-4.4 and Table 11-1 in Poling et al. (2001). For SOA, which was not explicitly treated in the predecessor model CAM4-Oslo (K13), we use a molecular weight of 168.2 (g mol$^{-1}$), corresponding to $C_{10}H_{16}O_2$ as our assumed representative SOA molecule. Due to lack of exact information about the large range of possible organic compounds we call SOA, we for simplicity and computational efficiency assume SOA to have the same microphysical properties (mass density, hygroscopicity, refractive index) as OM in the model, i.e., both in AeroTab5.3 and OsloAero5.3. A bug in the life cycle scheme (OsloAero4.0; K13) that produced too slow growth by condensation has also been found and rectified in OsloAero5.3. The effect of this is discussed to some degree by Iversen et al. (2018).

Mass densities and refractive indices are unchanged from K13, except for BC and mineral dust. For BC we have adopted the recommendations by Bond and Bergstrom (2006), using a monomer mass density of 1800 kg m$^{-3}$ and a refractive index of m = 1.95 - 0.79i (assumed to be wavelength independent). The refractive index for mineral dust has also been modified. This now follows Hess et al. (1998) for all wavelengths, which gives somewhat more light absorption by dust than in K13.

Modal number median radii and standard deviations for background tracers at the point of emissions (Table 2) are as in CAM4-Oslo, except for BC and sea-salt (SS_A1, SS_A2 and SS_A3). Sea-salt particle sizes have been changed to fit the new emission parameterization by Salter et al. (2015).

NMR for mixtures no. 2 and 12 (BC_A and BC_N from fossil fuel combustion) has been ca. doubled (to 24 nm) compared to CAM4-Oslo (11.8 nm), in order to account for some growth from the BC monomer size near the emission source to a more representative model grid mean value. This NMR is consistent with observations of somewhat aged BC mass size distributions of diesel exhaust and urban aerosol (Ning et al., 2013), and has also been shown to give more realistic aerosol number concentrations in a version of CAM4-Oslo with improved nucleation parametrization (M14). The new NMR is also more in



line with the Aitken mode fossil fuel carbonaceous particle size assumptions applied by the participating models in the multi-model AeroCom aerosol microphysics model inter-comparison study (Mann et al., 2014), which were in the range 15–40 nm. We note, however, that most of those models emitted particles as mixed BC-POM particles, so the size for a pure BC emission mode is not exactly comparable.

The externally mixed BC_AX mixture is a "fluffy", fractal structured agglomerate consisting of BC_N particles, assumed formed by rapid self-coagulation in the exhaust from fossil fuel combustion. We keep the standard deviation (SIGMA = 1.6) as in K13, but have reduced NMR from 0.1 to 0.0626 µm in order to conserve number concentrations as BC_AX gets coated and ages into BC_AI. We keep the assumed fractal dimension D (Ström et al., 1992) as in CAM4-Oslo, i.e., D = 2.5.

One aerosol tracer has been removed compared to CAM4-Oslo, namely the n-mode sulfate (SO4_N, originally mixture no. 11 in Table 2). This was done in order to save computational cost, and has been found to affect the all-over life-cycling properties with respect to e.g. sulfate concentrations and atmospheric residence times negligibly. This tracer was originally introduced to help mimic the growth in time from freshly nucleated sulfate particles (with a fixed size and composition) to aged particles.

Since the assumed chemical composition (with respect to life-cycling in OsloAero) in effect is quite similar to those of the aged particles, the division between those two aerosol tracers, despite somewhat different sizes, has been found unnecessary in OsloAero5.3.

Although the aerosol schemes are different from that of Liu et al. (2012), we use the same method for calculating aging of

externally mixed BC and organic aerosols. The layer thickness of SOA and sulfate condensate collected by the externally mixed species BC_N and BC_AX must exceed three monolayers (sulfate equivalent) before transitioning to the respective coated/aged particle mixtures is allowed. In K13 the BC_AX mixture was assumed to be large enough so that aging by condensation could be ignored, an assumption which was based on near surface measurements of BC in the remote Arctic. However, the extreme conditions in Arctic winter are not representative of conditions elsewhere, and this assumption

contributed to the somewhat exaggerated upper troposphere mass concentrations of BC that were modeled in CAM4-Oslo.

Hygroscopicites have also been modified somewhat, both with respect to internal consistency and simplicity. The new treatment makes sure that the hygroscopicity of an aerosol mixture for humidity swelling (for use with the offline optics calculations in AeroTab5.3) at slight sub-saturation (RH = 99.5%) is the same as the value used for calculating activation to

cloud droplets at supersaturated conditions (online in OsloAero5.3). These two cases were treated independently and could be slightly different in OsloAero4.0. The new growth factors (i.e., wet radius divided by dry radius) for RH values up to the cut-off value of 99.5%, hereafter referred to as $RH_{max}$, are shown in Fig. 2.



For BC we now assume a very low hygroscopicity of B = 5.0e-7 (Ghan et al., 2001) for all relative humidities. In CAM4-Oslo BC was assumed to be entirely hydrophobic (B = 0) in calculations of hygroscopic swelling, but B = 8.9e-7 with respect to CCN activation. Although the hygroscopicity for CCN activation is nearly halved since CAM4-Oslo, the values are already so small that the effect of this on cloud droplet production probably is negligible.

For ammonium sulfate we assume that B = 0.507 (Ghan et al., 2001) at $RH_{max}$ and at super-saturated conditions. This value is the same as in CAM4-Oslo with respect to CCN activation, but larger than what was used for hygroscopic growth at $RH_{max}$ (0.434). Instead of imposing a linear growth in the hysteresis domain, i.e. for RH = 37–80% (Tang and Munkelwitz, 1994; Tang, 1996) as in CAM4-Oslo, we here simply assume that B is reduced to the half (B=0.2535) between the points of

crystallization and deliquescence. Below the point of crystallization, the hygroscopicity is assumed to be the same as for BC (i.e. very low), compared to 0 in CAM4-Oslo.

While sulfate in OsloAero5.3 is consistently treated as ammonium sulfate, just as in CAM5.3 (Liu et al., 2012), in AeroTab5.3 we still (as in AeroTab4.0) treat both nucleated sulfate particles and condensate (SO4_NA and SO4_A1, respectively) as

sulfuric acid with respect to hygroscopicity. This hygroscopicity is now parameterized to vary with RH in such a way that the growth factor equals that of $H_2SO_4$ (for a range of RH values from 50 to 99%) in Table 2 in Köpke et al. (1997). By solving the Kohler equation, B is then estimated to be 0.534 at RH=99% (and assumed to be the same at $RH_{max}$), compared to 0.646 in CAM4-Oslo.

For sea-salt we have inferred the B values from Köpke et al. (1997), and then reduced the values by 50% in the hysteresis domain, i.e. for RH = 46–75% (Tang and Munkelwitz, 1994; Tang, 1996). This gives B = 1.20 at $RH_{max}$, which is slightly larger than the CAM4-Oslo B values of 1.15 at $RH_{max}$ and 1.16 for CCN activation (as in Ghan et al., 2001).

The OM hygroscopicity is assumed to be 0.14 (Ghan et al., 2001) for all RH values, slightly below the B value of 0.158 at

$RH_{max}$ but the same B value with respect to CCN activation as in CAM4-Oslo.

For mineral dust a B value of 0.069 has been chosen, consistent with a ca. 10% soluble mass fraction of dust. This is a high range value of the "less-hygroscopic" dust category in Koehler et al. (2009). In CAM4-Oslo much lower values were assumed, B = 4.8e-5 at $RH_{max}$, and B = 0.015 with respect to CCN activation. However, the new B value is still low compared to the

value 0.14 assumed by Ghan et al. (2001).

In this model version, as in CAM4-Oslo, hygroscopicity with respect to CCN activation is not calculated within AeroTab. AeroTab instead provides look-up tables of aerosol size parameters for each mixture, which in addition to B is used as input to the activation code (Abdul-Razzak and Ghan, 2000). The hygroscopicity is calculated as a mass weighted B for mixtures





which are uncoated or have a thin coating of soluble components (i.e. sulfate, OM and/or sea-salt), and as a mass weighted B of the coating itself when the coating is sufficiently thick. This threshold coating thickness is assumed to be 2 nm, as in K13.

## 2.4 Emission fluxes

DMS and biogenic OM emissions from the ocean have been updated to be wind-driven. In K13 DMS emissions were taken from Dentener et al. (2006), given as daily averages. Biogenic OM was assumed to have the same spatial distribution as the fine mode of sea-salt emissions given in Dentener et al (2006) and scaled to the global number in Spracklen et al. (2008). The DMS emissions are now instead given as the product of the transfer velocity and the ocean DMS molar concentration:

$$F_{DMS} = C \cdot k_{600} \cdot M_{DMS} \cdot C_{DMS} \tag{4}$$

Here $F_{DMS}$ is the flux of DMS [kg m$^{-2}$ s$^{-1}$], C is a unit conversion coefficient, $C_{DMS}$ is the DMS concentration in the ocean, given as monthly averages by Lana et al. (2011), $M_{DMS}$ is the molar mass of DMS, and $k_{600}$ is a transfer coefficient [cm h$^{-1}$] from Nightingale et al. (2000):

$$k_{600} = 0.222 \cdot U_{10}^2 + 0.333 \cdot U_{10} \tag{5}$$

where $U_{10}$ is the 10 m wind speed.

The flux of oceanic primary organic aerosols is given by O'Dowd (2008) and Vignati et al. (2010) to be proportional to the sub-micron sea-salt flux of the finest mode (SS_A1) and to the (monthly) organic matter concentration fraction in the water. Vignati et al. (2010) give the OM fraction as:

$$OM_{frac} = 0.435 \cdot \rho_{chl-a} + 0.13805 \tag{6}$$

$OM_{frac}$ is saturated at 90% according to O'Dowd (2008). $\rho_{chl-a}$ is the mass concentration of chlorofyll-a [mg m$^{-3}$] in the surface water, using SeaWiFS climatology (O'Reilly et al., 2000). A tuning constant has been added to the equation so that the OM flux from the ocean (still) matches the estimate of Spracklen et al. (2008) of approximately 8 Tg yr$^{-1}$.

The treatment of sea-salt fluxes in K13 has been changed to the formulation used for CAM4-Oslo in Salter et al. (2015), both being functions of near surface wind and sea-surface temperature. Dust sources were prescribed in K13. They are now wind-driven, calculated from the Dust Entrainment And Deposition (DEAD) model (Zender et al., 2003), which is implemented in the Community Land Model and is made available to OsloAero5.3. The parameterization is the same as that used by Liu et al. (2012), but fitted to the dust aerosol sizes used in OsloAero5.3.



As described in Sect. 2.2, the biogenic emissions of monoterpene and isoprene are calculated online (called every time step, which is 30 min.) from MEGAN (Guenther et al., 2012). The oxidant fields are prescribed as monthly averages but with a daily variation superimposed for OH and $HO_2$, and are therefore decoupled from the BVOC concentrations.

For aerosol and precursors not mentioned above, as in K13, the emissions are taken from CMIP5 (Lamarque et al., 2010) for the years 2000 (for simplicity called present day, PD), and 1850 (preindustrial, PI) conditions. The PD emissions and their vertical distribution are essentially the same as those used by Liu et al. (2012).

## 2.5 Heterogeneous ice nucleation

In this new version of CAM5.3-Oslo, the stochastic nature of freezing is considered for heterogeneous freezing in mixed-phase clouds, which is described according to classical nucleation theory (CNT, Pruppacher and Klett, 1997). Dust (DST_A2 and DST_A3) and black carbon (BC_A) can act as ice nucleating particles (INP). Water molecules can form small agglomerates of ice on the surface of INP, and if these ice clusters reach a critical size the thermodynamic energy barrier ΔG of the water–ice transformation is passed.

A common formulation for the ice nucleation rate is used for deposition and immersion freezing, as well as for contact nucleation, which is identical to Eq. (1) in Wang et al. (2014). Deposition freezing and contact nucleation takes place if the particles are uncoated or not completely coated. The coating thickness is calculated from the coated volume of the tracers and the volume of the dust and black carbon cores. The particles ability to act as INP in these mixtures is suppressed if the coated

volume exceeds the thickness of one monolayer of sulfate. Particles can be coated according to Table 2. Immersion freezing is allowed to take place on cloud-borne dust and black carbon, which becomes cloud-borne when interstitial particles merge with an already existing droplet or act as condensation nuclei themselves.

Two different approaches are considered for describing the contact angle for immersion freezing. The single contact angle ($\alpha$)

model is similar to previous descriptions with CNT (Hoose et al., 2010). Also an $\alpha$‑PDF model can be applied for dust immersion freezing, where the contact angle is formulated by a log-normal probability density function (Wang et al., 2014). Thus the inhomogeneity within the aerosol population can be represented by accounting for differences in the individual particle's ice nucleation properties (described in detail by Wang et al., 2014).

Compared to the study of Wang et al. (2014), we have used a small correction to the $\alpha$‑PDF model, which is also being taken into account in later releases of CAM versions by the National Center for Atmospheric Research (NCAR). The original



calculation of the probability p($\alpha$) in the $\alpha$-PDF model relies on a bin number of 101, which we have found to be too small to represent the log-normal distribution with a small standard deviation $\sigma$ (e.g. 0.01) properly (Fig. 3). This resulted in an unphysical lower limit of the activated fraction of INP, so that the INP activated fraction values were not able to fall below this limit and therefore stayed constant above a certain temperature (e.g. at ~$5 \cdot 10^{-5}$ for T > -15°C in Fig. 1 in Wang et al.,

2014). By increasing the bin number to 501, the distribution can be described more accurately (Fig. 3) and the unphysical behavior of the activated fraction is no longer present.

## 3 Model configuration and simulation set-up

All simulations have been run with 0.9° (latitude) by 1.25° (longitude) horizontal resolution and with 30 layers in the vertical. CAM5.3, and therefore also CAM5.3-Oslo, has two choices for microphysical schemes: MG1.0 (Morrison and Gettelman,

2008) and MG1.5 (Gettelman and Morrison, 2015), where MG1.5 is an update of the original formulation. We have in this study used the updated formulation. The land model CLM4.5 (Oleson et al., 2013) is configured with satellite observed phenology.

Two different configurations have been used to study and evaluate the aerosols: Nudged configuration (in the NUDGE_PD

and NUDGE_PI simulations) and AMIP configuration (in the AMIP_PD and AMIP_PI simulations), see Table 3 for an overview. The model has been run with aerosol and aerosol precursor emissions from year 2000 (PD) and 1850 (PI) for both configurations. We have here used PD oxidant levels also in the PI simulations, as in K13. The effects of using PI oxidant levels on the effective radiative forcing in CAM5.3-Oslo, and on the indirect effects in particular, are being studied by Karset et al. (2018). Only the aerosol and aerosol precursor emissions or concentrations differ between the PD and PI simulations,

while greenhouse gas concentrations, land-use, and prescribed SSTs and sea-ice concentrations are identical. Also the concentrations of DMS and biogenic OM in the ocean surface layer are the same, although the emissions of these into the atmosphere differ slightly due to different meteorological conditions.

The difference between the AMIP and the Nudged configuration is that the latter includes additional terms to the dynamical

equations which push (nudge) the model meteorology towards the observed (or reanalyzed, read in) meteorology, using a relaxation time scale of six hours. The main purpose of using the nudged configuration is to constrain natural variability, as a significantly higher number of simulated years are required to isolate statistically significant differences in cloud radiative forcing (due to anthropogenic aerosols) with the free AMIP configuration (Kooperman et al., 2012). Another objective is to obtain a model meteorology that more closely resembles actual meteorological conditions during the period of observations

which the model is compared with in the aerosol and cloud validation in Sect. 4. We have run both configurations in order to verify that results are coherent, and to be able to study how much the nudging affects the results.



In the Nudged configuration, we use meteorological data from ERA-interim (Berrisford et al., 2011) for the period 2004–2010. We nudge only to horizontal winds and surface pressures (Zhang et al., 2014). This way of nudging will allow the aerosols to influence temperatures and clouds. While nudging also to observed temperatures might improve the comparison of aerosol

properties with observations, leaving the temperature un-nudged is important for calculation of the indirect and semi-direct effect of aerosols (Zhang et al., 2014), since these are most realistically (or at least consistently) estimated with the model's own vertical temperature gradients, which again are crucial for atmospheric stability and vertical mixing.

## 4 Results and discussion

The predecessor model version CAM4-Oslo has been extensively validated and compared with other models through the

AeroCom project (Aerosol Comparisons between Observations and Models: http://aerocom.met.no), in studies by Jiao et al. (2014), Tsigaridis et al. (2014), Kipling et al. (2016), and Koffi et al. (2016), as well as in K13. A separate evaluation of CAM4-Oslo and other CMIP5 models by use of remote sensing of aerosols in the Arctic was made by Glantz et al. (2013). In this section we attempt to answer the question: how does CAM5.3-Oslo perform with respect to aerosol and aerosol related cloud properties, compared with observations? We first compare some of the results with CAM4-Oslo (K13) and other studies,

both in order to discuss properties which cannot easily (or at all) be compared with observations, and to be able to see whether the updates and extended physical parameterizations have improved the model performance with respect to aerosols or not. The latter question is not straight-forward, since the host model itself has undergone a great number of changes in moving from CAM4 to CAM5.3. Additionally, CAM4-Oslo was run with a coarser horizontal resolution of 2°.

### 4.1 Concentrations and budgets

#### 4.1.1 Budgets and vertical profiles

Table 4 shows the budgets for the different species in the model simulations. For each term in the table, results from both present day (PD) and preindustrial (PI) conditions are listed, together with the respective values found in K13. Unless otherwise stated, the discussed model values are from the NUDGE_PD simulation.

The result of the change in DMS emission parameterization described in Sect 2.4 is an almost doubled DMS column burden (34-35 Tg S yr$^{-1}$) compared to the 18.1 Tg S yr$^{-1}$ found in K13, accompanied by a similar increase in the SO$_2$ source term from oxidation of DMS. The main reason for the increase is that the DMS emissions in Dentener et al. (2006) (applied in K13) were based on the DMS climatology of Kettle and Andreae (2000), with generally lower DMS concentrations in sea-water than in the updated version of Lana et al. (2011). An experiment with wind-driven DMS emissions in a research version of CAM4-

Oslo using the same transfer function gave 22.0 Tg S yr$^{-1}$ with the Kettle and Andreae (2000) data and 34.2 Tg S yr$^{-1}$ with the Lana et al. (2011) data. The shorter lifetime of DMS (1.5 days) compared to K13 (2.4 days) is likely caused by the use of



different oxidant fields. Liu et al. (2012) obtain a lifetime of 1.3 days using nearly the same chemical mechanism (see Sect. 2) and the same oxidant fields as in the present work, but with emissions from Dentener et al. (2006).

The chemical source for $SO_4$ is divided into clear-air sources, through the $SO_2 + OH$-reaction, and production in cloud water.
The chemical sources of OM (via SOA) are mainly from monoterpene and isoprene. This gives a total of 78 Tg(OM) yr$^{-1}$ of SOA produced from terpenes, which lies within the range of AeroCom models published by Tsigaridis et al. (2014). For comparison, the total amounts of BVOC emitted as isoprene and monoterpene are 438 and 119 Tg yr$^{-1}$, respectively. There is also a source from oxidation of DMS to MSA which is assumed to form organics (ca. 9 Tg(OM) yr$^{-1}$), adding up to a total of 87 Tg(OM) yr$^{-1}$. As mentioned in Sect. 2.2, for the MSA contribution to SOA, 20% and 80% of the MSA mass is added to the
$SOAG_{LV}$ and $SOAG_{SV}$ tracers, respectively. The exact LV to SV ratio is unknown, but we find a quite low sensitivity of the anthropogenic change in cloud effective radiative forcing (i.e., the indirect effect, which is the most important in a climate change perspective) to assumed apportionment of MSA: test simulations indicate that the total short-wave and long-wave indirect effect only changes by about -0.03 W m$^{-2}$ if all MSA goes into $SOAG_{SV}$ (no nucleation), and by 0.00 W m$^{-2}$ if we instead feed all MSA into the $SOAG_{LV}$ tracer. The effect of neglecting the MSA contribution to SOA altogether is similarly
estimated to give a -0.10 W m$^{-2}$ change.

The zonal mean mass mixing ratios and their variation with height for $SO_2$, BC, OM, Sulfate, mineral dust and sea-salt (SS) are shown in Fig. 4, both for NUDGE_PD and AMIP_PD. The figure shows that some BC is transported to the stratosphere where the lifetime is longer. OM and sulfate also have this secondary maximum in the stratosphere, but the concentrations aloft are smaller in CAM5.3-Oslo than in CAM4-Oslo (not shown). Dust and sea-salt do not exhibit the same clear secondary
maxima in the stratosphere, since these particles are generally larger and more readily removed by sedimentation.

For BC we can compare the model with profiles from the HIAPER (High-Performance Instrumented Airborne Platform for Environmental Research) Pole-to-Pole Observations (HIPPO) of Carbon and Greenhouse Gases Study over the (mainly
remote) Pacific Ocean in January and November 2009, March/April 2010, and June/July and August/September 2011 (Wofsy et al., 2011; Schwarz et al., 2013), see Fig. 5. It is clear that the new model version does produce less excessive BC concentrations in the upper troposphere and in the stratosphere, globally (Fig 5 a), and that it now compares better with the HIPPO observations in the Pacific (Fig 5 b), although the concentrations are still too high in the upper troposphere and lower stratosphere for this region, similar to the findings for CAM5.3-MAM4 in Liu et al. (2016). This is probably related to the way
aerosols are transported and scavenged in deep convective clouds in the model. There are currently ongoing tests with alternative treatments of convective transport and mixing (cf. Sect. 2.1.5 of K13 for a sensitivity test on this in CAM4-Oslo), of which improved treatments will possibly be included in the upcoming CAM6-Oslo version for CMIP6.



Some of the changes in aerosol concentration fields are connected to changes in cloud microphysics in the host model. Two major factors that affect both aerosols and aerosol precursors are the amount of liquid cloud water and the cloud fraction. Globally averaged, CAM5.3-Oslo has only about one third as high cloud liquid water path (LWP) as CAM4-Oslo, while the precipitation rate is slightly (7%) larger. Since the loss rate of aerosol activated to cloud droplets in the model is assumed to

be proportional to the precipitation to LWP ratio, an increased scavenging efficiency and a subsequent reduction in aerosols away from source regions as a result of the reduced LWP is to be expected. A reduction in aerosol transport to remote regions is indeed found for all aerosol components, and is particularly pronounced in the Arctic and Antarctic regions. At the same time, the total (low) cloud cover has increased from 53% (34%) in CAM4-Oslo to 66% (43%) in CAM5.3-Oslo, with the largest changes at high latitudes. This increase in cloud cover likely also gives an increase in the frequency of precipitation

events, which tends to reduce aerosol lifetimes.

Even sea-salt burdens have been reduced away from its source regions, despite an almost 4 times increase in global lifetime, which is now 1.07 days. This is to a large degree due to the shift towards more long lived (i.e., accumulation mode) particle sizes (compare Table 2 with Table 1 in K13). While the lifetime is now longer, the emissions have decreased even more, so

that the overall sea-salt burdens are about 35% smaller than in CAM4-Oslo. In Liu et al (2012), the sea-salt lifetime lies between CAM4-Oslo and CAM5.3-Oslo, but is quite dependent on the aerosol microphysics (0.76 days in MAM3 and 0.55 days in MAM7).

As for BC, also the concentrations of OM, sulfate and mineral dust have dropped in the upper troposphere and lower

stratosphere, when going from CAM4-Oslo to CAM5.3-Oslo. This reduction is more substantial for carbonaceous aerosols than for the other species, however. In addition to the increased overall scavenging efficiency, BC and primary OM now experience a more rapid transition from external to internal mixtures, see Section 2.3. The lifetimes of BC and OM of approximately 5 days in CAM5.3-Oslo are now more comparable to the MAM7 values in Liu et al. (2012), which are 4.4, 4.9 and 4.1 days for BC, OM and SOA, respectively.

The situation for sulfur is more complex. While the scavenging efficiency of $SO_2$ is increased on the one hand, as for aerosols, the lower liquid water content in CAM5.3-Oslo on the other hand acts to reduce the aqueous phase reaction rates. The net effect of all changes is a ca. 20% increase in lifetime. Furthermore, while $SO_2$ (and thus the potential for formation of sulfate) is now transported higher into the atmosphere, the increase in aerosol activation scavenging tends to counteract the effect of

this enhanced transport. The combined effect of a longer lifetime for $SO_2$ and increased aerosol loss rates in the lower troposphere is just a 3% overall reduction in the atmospheric residence time of $SO_4$. The estimate at 3.70 days is very close to the sulfate lifetimes in Liu et al (2012): 3.72 days for MAM7 and 3.77 days for MAM3.





As for carbonaceous aerosols, also the lifetime of mineral dust is reduced. The main reason for this reduction is most likely the general increase in activation scavenging. Below-cloud collection efficiencies are still as in Seland et al. (2008), so that any changes in below cloud scavenging are due to changes in precipitation and aerosol life cycling. The relative amount of dust emitted in the accumulation mode (DST_A2) in the new emission parameterization (13%) is larger than for the prescribed

emissions in CAM4-Oslo (11%), which should rather contribute to a longer dust lifetime in CAM5.3-Oslo, due to reduced gravitational settling. A test simulation performed with an earlier model version showed that a tuning of the relative amount of emissions taking place through the accumulation mode from 13% to 20% lead to a 20% increase in lifetime, globally. The inherent assumption of OsloAero that there is a constant size background aerosol – the particles cannot shrink to smaller sizes than that of the background as the largest particles are deposited – may result in a shorter lifetime of the coarse mode, compared

to the modal aerosol schemes (MAM3 and MAM7) in Liu et al. (2012). Since Liu et al. (2012) calculate number and mass independently, the size of the coarse mode particles may decrease with time, thereby increasing the lifetime of that mode. The estimated dust lifetime of 1.9 days is shorter than in both MAM3 (2.6 days) and MAM7 (3.1 days) in Liu et al (2012).

According to Kok et al. (2017), mineral dust in global models is probably often too fine, based on constrained atmospheric

dust properties and abundance. AeroCom emission rates and loadings (Textor et al., 2006) are below the central estimates in Kok et al. (2017) of (1000–2700) Tg yr$^{-1}$ and (13–29) Tg, respectively. We get a slightly higher global emission rate of 3100 Tg yr$^{-1}$,in NUDGE_PD, but 2500 Tg yr$^{-1}$ in the AMIP_PD simulation. The estimated global dust burden of 13 (NUDGE_PD) or 16 Tg (AMIP_PD) which follows, however, falls within the central estimates of Kok et al. (2017). The global emission rate may be adjusted by a tuning factor (a constant in the emission flux term) in CAM5.3, but in the present version we have

retained the value used in the original CAM5.3 code.

Some of the aerosol burden changes from CAM4-Oslo to CAM5.3-Oslo are due to differences in meteorology. To roughly estimate the magnitude of such an effect, we compare the NUDGE_PD and AMIP_PD results in Table 4.

The globally averaged burdens of DMS and $SO_2$ differ by less than 2%, $SO_4$ and sea-salt by less than 1%, while BC and OM and mineral dust are ca. 5% and 7% lower in the free-running AMIP simulations, respectively. So for these species the differences between NUDGE and AMIP are quite small. We would probably have obtained even smaller changes if the model was self-nudged, i.e. being nudged to a meteorology produced by the model itself (e.g., the AMIP_PD simulation) instead of the ERA meteorology. In a similar comparison by Liu et al. (2016), they obtain as much as ca. 20% lower BC and OM burdens

with nudged (towards 1 year recurrent meteorology) vs. a free simulation (10 years). They partly attribute this to inter-annual variability, but mainly to (climatological) differences in the meteorological conditions between the free and nudged model simulations, which affect aerosol transport and cloud processing. Unlike the nudging procedure applied here, Liu et al. (2016) also nudged the model meteorology to reanalyzed temperatures (Tilmes et al., 2015), which may explain the larger difference in simulated aerosol burdens between their nudged and free AMIP simulations. A similar effect was found in an older version



of CAM5.3-Oslo as we went from nudging temperatures, specific humidity, as well as U, V, PS and some surface fields, to only nudging U, V and PS: the difference in globally averaged LWP between the nudged and free simulations was reduced by an order of magnitude.

The 3% increase in sea-salt emissions in going from NUDGE_PD to AMIP_PD, which is consistent with larger simulated 10m wind speeds in the extratropical storm track regions, is almost offset by a reduction in lifetime (more wet scavenging), giving only a 1% net increase in column burden. There is one exception for which the difference between NUDGE_PD and AMIP_PD seem to be important, however, namely for mineral dust. This is most readily seen from the global dust emissions, varying with wind speed and soil humidity, which are 19% lower in AMIP_PD than in NUDGE_PD, very close to the 18% difference
in atmospheric burden.

The contribution by inter-annual variations in the NUDGE_PD simulation to global aerosol or aerosol precursor burdens, here given as normalized standard deviations, is found to be about 3.6% for DMS, 0.8% for $SO_2$, 1.2% for $SO_4$, only 0.1% for BC, 1.0% for OM, 2.6% for sea-salt, and 2.5% for mineral dust. Hence, the above estimated changes in burdens from NUDGE_PD
to AMIP_PD are actually smaller than one standard deviation of the inter-annual variation (in NUDGE_PD) both for DMS, $SO_4$, and for sea-salt, so that only $SO_2$, BC, OM and mineral dust with some confidence can be said to be different (smaller) in the AMIP_PD than the NUDGE_PD simulation.

### 4.1.2 Evaluation of near surface mass concentrations

Column burdens cannot be measured and here observed surface concentrations are used for validating the aerosol masses in the model. Figure 6 and Table 5 show surface mass concentrations of BC, $SO_2$, OA (modeled OM vs. observed OC*1.4, see explanation in the figure caption and below), SS (sea-salt), $SO_4$ (sulfate) and DUST (mineral dust) in NUDGE_PD, compared with various observations as available via the AeroCom inter-comparison project (http://aerocom.met.no). Note that the amount of data and spatio-temporal coverage available for the different parameters is inhomogeneous because of data network
fluctuations and incomplete storage in the databases used (EBAS: Tørseth et al., 2012, see also http://ebas.nilu.no; AEROCE: Arimoto et al., 1995; Huneeus et al., 2011). Tables 6–8 give an overview of statistical evaluation of the NUDGE_PD and AMIP_PD simulations as well as a range of AeroCom Phase II (AP2) and AeroCom Phase III (AP3) models. These are compared for different years, both for individual years (meteorology of 2006 for AP2 and 2010 for AP3) and our model climatology against a climatology from the observational data.

We find that the model mainly overestimates $SO_2$ concentrations. One possible explanation for the large positive bias is the low vertical and horizontal resolution in the model. With such low resolution the model does not capture well the dispersion of primary emissions of $SO_2$ from large point sources or shipping routes. A part of this bias probably comes from the fact that



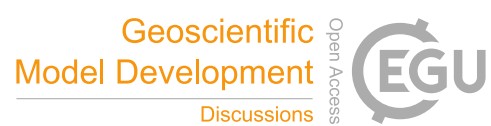

we are comparing concentrations at the mid-point of the lower-most model layer (~50 m) with ground-based observations (see discussion in Simpson et al., 2012). The Pearson correlation coefficient R (hereafter often just referred to as the correlation) is slightly better than for the climatological average in the un-nudged AMIP simulation, where instead the normalized mean bias NMB (hereafter often just referred to as the bias) is slightly better. The bias and correlation for each of the continents are 216% and 0.52 for Europe, 134% and 0.94 for North America, and -53% and -0.02 for Asia. None of the AP3 models have available $SO_2$ statistics, while 4 of the 5 AP2 models that do, exhibit higher biases than ours. The correlations are also lower than ours in all the AP2 models, while 3 of them have a higher percentage of monthly model values within a factor 2 of the observed values (Fact2).

Sulfate is also somewhat overestimated, with a positive bias of 22% and a correlation as high as 0.72 for the monthly climatological data, slightly above that of the free AMIP simulation. CAM4-Oslo exhibits a smaller, slightly negative bias, but is less correlated with the observations. The new model version still yields a lower Fact2 value, all in all performing slightly worse than the predecessor. Biases and correlations for each of the continents are 15% and 0.54 for Europe, 38% and 0.92 for North America, and -9% and 0.59 for Asia. The bias for sulfate is better than in 4 of the 8 AP3 models with available concentration data (for year 2010), while the correlation falls just below the AP3 range. Comparing against the 23 AP2 models (for year 2006), however, CAM5.3-Oslo has lower bias than only 6 of the AP2 models, while out-performing or matching 14 models with respect to correlation.

We see that the model mainly underestimates BC, especially the highest concentrations. The bias is -28% and the correlation 0.38, which also here is higher than for the AMIP simulation. CAM4-Oslo has an almost twice as large bias and a much lower correlation coefficient, so apparently there has been an improvement in modeled BC surface concentrations for the very limited number and geographical coverage of stations available (in Europe only). As much as 75% of the model values lie within a factor 2 of the observed values, compared to 68% for CAM4-Oslo. The BC bias is also better than in 6 of the 8 AP3 models. Although the correlations for BC are quite low for all the AP3 models, only one has a lower correlation than CAM5.3-Oslo, however. Similarly, comparing against the 23 AP2 models, CAM5.3-Oslo out-performs only 7 of the models bias-wise, and 6 with respect to correlation.

For calculation of mass concentrations of OM from OC the model does not distinguish between tracers from different source types, since they are lumped together for each of the background and size-modifying tracers. This has been done in order to limit the CPU requirements as much as possible, as the model (when fully coupled with the ocean and sea-ice modules) is built for use in long climate simulations. We here compare modeled OM with observed OC values which have been multiplied by 1.4 (defined as OA for the observations in Fig. 6, while OA simply means OM for the model values) to account for the conversion factor in going from fossil fuel OC to OM in the model (K13). For OM from biomass burning, defined as agricultural waste burning, grass fires and forest fires in the model, the respective conversion factor is assumed to be 2.6 (K13,



see also Formenti et al., 2003), i.e., 1.86 that of the fossil fuel emissions. If all OM originated from biomass burning, the bias would therefore be 19% instead of 122%. The latter value is simply based on the assumption of zero OC contribution from biomass burning. The truth concerning the validation probably lies somewhere between these two estimates, even though OM/OC ratios exceeding 2.6 might be more representative for some sources, such as MSA (cf. section 4.2.1 in K13). For

comparison, the respective bias values in CAM4-Oslo are 108% and 12%. The correlation coefficient for OM in CAM5.3-Oslo's NUDGE_PD is substantially lower than for both BC and SO$_4$, but very close to that for OM in both AMIP_PD and CAM4-Oslo. Regional bias and correlation values are 143% and 0.44 for North America, where most of the observation sites are located, and -26% and 0.01 for Europe.

Assuming that OA is representative for the modeled OM, in North America the concentrations are most overestimated in the months JJA, while being underestimated in DJF. In Europe OM is overestimated only in JJA. This may indicate that OC is overestimated in summer, or that sources with OM/OC ratios exceeding 1.4 dominate during summer, as should be expected since relative contributions to OM from SOA (e.g., Gelencsér et al., 2016) and forest-fires are generally larger in this season. As discussed in K13, in addition to the various OM/OC ratios in the model, as in nature, a further complicating factor comes

from use of different standards and methods for measuring OC mass concentrations. While being integrated over all particle sizes in the model, the measured quantities may be based on PM2.5 or PM10 values in different observation networks, as is the case for North America (PM2.5 in IMPROVE) vs. Europe (PM10 in EMEP). This hampers reliable validation of OM in the model in its present form. Ideally the model should carry separate tracers for OC from SOA (preferably speciated), fossil fuel and biomass burning sources, and also have separate mass diagnostics for the different size intervals, which would better

facilitate a more comprehensive evaluation of organic matter in the model.

Compared to the 8 AP3 models, the bias (i.e., modeled OM – measured OA) is found to be smaller than in all but one model. The correlation, however, is just below the range for the AP3 models. It is slightly negative for the whole year of 2010 (Europe only), as for the months MAM that year, while being 0.15 or higher in the other seasons. Comparing against the 22 AP2 models

(one model is missing surface concentration data), CAM5.3-Oslo has a smaller bias than only one of the models, while it performs better than 6 models with respect to correlations for the year 2006. The Pearson correlation in our model varies between 0.16 and 0.25 for years 2004–2006, when both North American and European stations are included, while being closer to zero or negative in 2007–2010 based only on European station data, where it varies between -0.12 and 0.12. It is surprising that there has been practically no change in correlation for the all year climatology since K13 (CAM4-Oslo in Table

7), where the SOA treatment was a very simplistic. This should be investigated in future studies.

For the sea-salt surface concentrations we obtain a bias of 22%, a correlation of 0.54, and 31% of the model values within a factor 2 of the observations. Compared to CAM4-Oslo this is much better bias-wise, but with nearly the same Fact2 value. The bias is also about half of that in the free running AMIP_PD simulation. Regional biases and correlations are 59% and 0.76 for





Europe, 19% and 0.72 for North America, and 31% and -0.04 for Asia. A considerable number of the observation stations for sea-salt are coastal and inland, however, and are perhaps therefore not very representative for sea-salt aerosol as such in the model. CAM5.3-Oslo performs better than all the AP3 models, bias-wise, and only one of the AP3 models has a higher Pearson correlation for sea-salt. Our model is also less biased than 20 of the 23 AP2 models, and with higher correlation than 21 models.

For mineral dust we only have climatological observations to compare with. The bias for all stations and months is found to be -39%, with a correlation of 0.52, which is here slightly lower than in the free AMIP_PD simulation. The observation stations for mineral dust surface concentrations are all quite distant from the largest dust source regions. Hence, the negative bias found in CAM5.3-Oslo may very well be a result of underestimated long range transport rather than too small emissions. This is
corroborated by the fact that aerosol optical depths in the largest source regions (see Sect. 4.2) are biased high compared to the remote retrieved values. Although the correlation coefficient is slightly better than in CAM4-Oslo, where mineral dust emissions are simply prescribed, CAM5.3-Oslo is more biased and has a lower Fact2 value. We note, however, that even for the nudged simulation, the year to year variation for mineral dust is large enough to affect these validation results. Comparing monthly data from each individual model year with the observed climatological dust concentrations, the bias here varies
between -46% and -23%, and the correlation between 0.29 and 0.71. Part of this variability may be due to a varying number of stations for which there are enough data to be included in the multi-year climatology. Compared to the 8 AP3 models, our model performs better than only 3 models bias-wise, but lies above the middle of the AP3 range with respect to correlations. It is also less biased than 14 of the 23 AP2 models, and has a higher correlation than 7 of the models

## 4.2 Optical properties

### 4.2.1 Mass specific extinction and absorption

Table 9 gives the modeled mass extinction coefficients (MEC) for each of the aerosol components, calculated as the component's aerosol optical depth at 550 nm divided by its atmospheric burden. What determines MEC for a mono-disperse aerosol consisting of spherical (which we assume) and homogeneous particles is the particle size (divided by the radiative
wavelength of interest), its mass density and refractive index. For an internally mixed component of an aerosol size distribution, the size-integrated and atmospheric column averaged MEC depends on a range of factors in the model. In addition to the refractive index of the components in a given mixture, and the mixture's log-normal modal parameters (median radii and standard deviations) at the point of emission or nucleation, the growth by added process tracers and by hygroscopic swelling also play important roles. Aerosol life times and aerosol life-cycling in general, including transport and deposition, can further
affect the results by shifting the "center of mass" of the aerosol components in question to areas and altitudes with different relative humidity, which consequently also affect the globally averaged MEC value.



Since neither the assumed mass density nor the initial log-normal modal parameters of the sulfate background modes in mixtures no. 1 and 5 have changed relative to the treatment in K13, i.e. in CAM4-Oslo, the ca 14% reduction in MEC globally must be due to changes in growth, including the effects of life-cycling on growth. As outlined in Sect. 2.3, the hygroscopicity of sulfuric acid has been reduced by about 17% for relative humidities close to $RH_{max}$, while for ammonium sulfate there has

been an equally large increase for these highest RH values but a larger reduction in large parts of the hysteresis domain (RH ≈ 50–80%), up to a 50% reduction at RH ≈ 80%. The net effect of this when introduced into the model at the time (in an older model version), was however small compared to the change from CAM4-Oslo to the present model version, which points to changes in meteorology and life-cycling as the main cause. Although the atmospheric residence times and burdens of sulfate are quite similar globally (Table 4), in CAM5.3-Oslo they are both considerably smaller at mid- to high latitudes and somewhat

larger in the subtropics. At these low latitudes the relative humidity (and cloud cover) in the lower troposphere is also somewhat lower in CAM5.3-Oslo. Hence the sulfate "center of mass" is in effect shifted towards typically less humid regions, which is consistent with less hygroscopic growth and the smaller MEC values found in CAM5.3-Oslo. Some of the reduction may in addition be a result of having relatively larger amounts of (less hygroscopic) OM internally mixed with sulfate in the present model version, both due to co-nucleation of sulfate and SOA (mixture no. 1) and to condensation of sulfuric acid and

$SOAG_{SV}$/$SOAG_{LV}$ onto larger particles (mixtures no. 1–10). The sulfate MEC estimates lie within the inter-model variability of the AeroCom Phase I models (Kinne et al., 2006) for both configurations of CAM5.3-Oslo, as for CAM4-Oslo.

MEC for OM aerosol has decreased by about 30% compared to CAM4-Oslo, also still within the range of the AeroCom I models, but now being closer to the AeroCom I median value. Looking back on results from earlier model versions of CAM5.3-

Oslo, we find that the larger part of this change is most likely due to a shift in OM burdens to less humid areas (mainly at lower latitudes), just as for sulfate. An additional change that might be of importance is that SOA now comes as nucleation/Aitken mode particles (mixture no. 1) and is distributed onto larger particles by condensation, instead of in the internally mixed primary OM/BC(a) mode (mixtures 4 and 14), which generally has a higher specific extinction. For instance, MEC is about 0.4 (0.6) $m^2$ $g^{-1}$ for mixture 1 if only consisting of nucleated OM at RH = 0% (80%), compared to 3.0 (4.5) $m^2$ $g^{-1}$ for mixtures

4 and 14 when only consisting of OM (and condensed water).

Despite a shift in burdens towards lower latitudes also for BC, the mass specific extinction for BC (7.6 $m^2$ $g^{-1}$) has increased by about 17% from CAM4-Oslo to CAM5.3-Oslo. This is also closer to the AeroCom I median value (8.9 $m^2$ $g^{-1}$). Regionally the increase is largest in areas downwind of relatively large sulfate and SOA/biomass burning sources in northern South

America (where MEC now is at its largest at about 20 $m^2$ $g^{-1}$) and Indonesia (~15 $m^2$ $g^{-1}$), as well as over and downwind of eastern North America to eastern Europe (~10–15 $m^2$ $g^{-1}$). As mentioned, there is more growth by condensation in CAM5.3-Oslo since SOA is no longer treated as primary particles as in CAM4-Oslo. It is reasonable to assume that this extra aerosol growth also may be linked to the increase in MEC. Most importantly, however, the changes in BC emissions size, mass density and refractive index (see Sect. 2.3) did change MEC for the pure and dry (RH=0%) background particles of mixtures no. 2



(when containing only BC) and 12 from about 7.0 to 8.5 m$^2$ g$^{-1}$, i.e. a 20% increase from the background tracer with the largest mass-wise contribution (90%) to fossil fuel BC emissions. For mixture no. 0, the fractal fossil fuel BC particles, the net change of MEC from altered size, density and refractive index is just a 0.3% increase to 8.2 m$^2$ g$^{-1}$, due to compensating effects. The increase in MEC is also very small (~3%) for fresh BC particles from biomass burning in mixtures no. 4 and 14, if we assume
that only BC is present in the OM/BC(a) mode.

MEC for mineral dust has increased by about 19% globally, and with a regional pattern quite similar to that of BC. Mass densities and particle sizes at the point of emission are here the same as in CAM4-Oslo for both tracers (DST_A2 and DST_A3). The effect of the change in refractive index (see Section 2.3) only yields a 0.4% increase in MEC at 550 nm for
pure dust both in mixture 6 and 7. Dust hygroscopicity has increased somewhat (see Sect 2.3), which together with the extra growth potential from SOA is consistent with an increase in MEC. We note, however, that MEC is now higher even in the most arid source regions (e.g., Sahara), due to a slightly larger fraction of accumulation mode (DST_A2) to total dust mass in the new emission parameterization (0.13) compared to CAM4-Oslo (0.11). With MEC = 2.44 m$^2$ g$^{-1}$ and 0.335 m$^2$ g$^{-1}$ for DST_A1 and DST_A2 (assuming no growth), this shift towards smaller sizes alone (i.e., before further growth and deposition)
can account for a 7% larger MEC for dust in CAM5.3-Oslo. Mineral dust MEC is still within the range of the AeroCom I models, although it is now closer to the highest model estimates. Note that the (common) assumption that dust particles are spherical leads to a substantial underestimate in MEC for coarse particles, while the error is much smaller for particles with geometric diameters below about 0.6 µm (Kok et al., 2017). The bias towards smaller emission sizes, however (cf. discussion above), should lead to an opposite directed bias in MEC, since coarse mineral dust has much lower MEC than sub-micron dust
(e.g., 86% lower for DST_A3 than for DST_A2).

The hygroscopicity of sea-salt has increased by about 4% for high ambient relative humidities, while now being smaller throughout much of the hysteresis domain compared to CAM4-Oslo (see Sect. 2.3). Together with changes in particle growth by the process tracers, such as by condensation of SOA (missing in K13 and M14), this might explain some of the changes in
sea-salt MEC in moving to CAM5.3-Oslo. The main cause of the about 63% increase, however, is the shift in particle effective radii towards sizes with higher specific extinction: Globally averaged MEC for sea-salt in CAM5.3-Oslo (5.04 m$^2$ g$^{-1}$) is just 1% lower than in the CAM4-Oslo development version of Salter et al. (2015), which used the same model parameters for sea-salt as in Table 2, while otherwise being the same as in K13.

Note that these MEC estimates are based on the common assumption that an internally mixed component's contribution to the total extinction increases linearly with its volume fraction, which in our model (in AeroTab) is allowed to vary with size. The same goes for the absorption or scattering, when we focus on either of their contributions to the extinction separately. In this way non-absorbing aerosols, such as sulfate and sea-salt, contribute to the total aerosol absorption wherever internally mixed with absorptive aerosols, such as BC. Although the total extinction/scattering/absorption is thus correctly found by summing



up the contributions from each of the aerosol components, the method is expected to give biased extinction estimates, especially for the absorption part, compared to in situ measurements for each aerosol component separately (or for less aged/internally mixed particles close to the sources). Furthermore, refractive indices of mixtures consisting of absorbing and non-absorbing components are calculated by use of the semi-empirical Maxwell–Garnett mixing rule, which gives less absorption (in better

agreement with measurements) than the volume mixing rule for homogeneous mixtures (Chýlek et al., 1998), but more absorption than for purely external mixtures (Chýlek et al., 1998, see also Fig. 6 in Kirkevåg et al., 2005).

To obtain a first rough estimate of the magnitude of at least parts of the uncertainty in connection with choice of methodology for calculating BC MEC and the corresponding mass specific absorption, MAC (defined as absorption aerosol optical depth

(AOD) divided by aerosol burden), we have also calculated the corresponding coefficients for the anthropogenic part (i.e. using PD – PI AODs and burdens). This means a shift towards sizes and specific extinctions more representative of fossil fuel sources. The anthropogenic MEC is found to be about 8% larger than for PD BC, 8.18 m$^2$ g$^{-1}$ for NUDGE_PD and 8.28 m$^2$ g$^{-1}$ for AMIP_PD, and 10% higher (7.14 m$^2$ g$^{-1}$) than for PD BC in CAM4-Oslo. Similarly, the anthropogenic MAC value is as much as 30% higher than for PD BC, 3.15 m$^2$ g$^{-1}$ for NUDGE_PD and 3.27 m$^2$ g$^{-1}$ for AMIP_PD, and 31% higher for

anthropogenic BC (3.15 m$^2$ g$^{-1}$) than for PD BC in CAM4-Oslo. We note that this is still low compared to measured values and the recommended range of 7.5±1.2 m$^2$ g$^{-1}$ for fresh, uncoated BC in Bond and Bergstrom (2006). According to that review paper, MAC can drop to about 5 m$^2$ g$^{-1}$ for collapsed BC aggregates, while coating by negligibly absorbing aerosol typically enhances MAC by 50% (to ca. 11 m$^2$ g$^{-1}$).

One may also calculate alternative MAC values from the PD simulations by assuming that non-absorptive or less absorptive components do not contribute to the light absorption of the mixture containing BC. First, leaving out only sulfate and sea-salt and letting MAC = ABS$_{(BC+SO4+SS)}$/B$_{BC}$, we find that MAC = 4.82 and 4.95 m$^2$ g$^{-1}$ in NUDGE_PD and AMIP_PD, respectively, compared to 5.07 m$^2$ g$^{-1}$ in CAM4-Oslo. MAC here exceeds 7 m$^2$ g$^{-1}$ over large areas (for all the above simulations) somewhat downstream of major BC emissions in North and South America, and over several smaller areas in South-East Asia. Similarly,

assuming that mineral dust and OM do not contribute to the absorption either (as in Stjern et al., 2017), which is a much less realistic assumption in many regions, we obtain global MAC values of 23.2 and 21.4 m$^2$ g$^{-1}$ in NUDGE_PD and AMIP_PD, respectively, and 13.6 m$^2$ g$^{-1}$ in CAM4-Oslo. Assuming that the truth lies somewhere between the two last assumptions we could obtain even globally averaged BC MAC values within the recommended range of Bond and Bergstrom (2006). The problem with this line of reasoning is, of course, that BC is not the only absorbing aerosol component, and that non-absorptive

components also add to, and even enhance (e.g., Chen et al., 2017) the total absorption for internal mixtures. Finally, although both mineral dust and OM individually have small MAC values, they have much larger atmospheric burdens than BC and thus also contribute considerably to the total absorption, even dominating regionally. In a test simulation with less absorptive mineral dust at most wavelengths – the imaginary refractive index at 550 nm is reduced from 0.0055 to 0.0024 – otherwise being identical to NUDGE_PD, the latter BC MAC estimate is reduced by 25% globally, from 23.2 to 17.5 m$^2$ g$^{-1}$. Assuming



linearity in MAC with respect to the imaginary part of the refractive index, MAC for BC partially internally mixed with non-absorptive dust can from this be estimated at ca. 10.1 m$^2$ g$^{-1}$. The absorption by OM is still included in this estimate, however.

### 4.2.2 Column integrated optical properties

Figure 7 shows aerosol optical depth and absorptive optical depth at 550 nm as well as Ångström parameter for wavelengths
440 to 870 nm in the NUDGE_PD simulation, compared with AERONET (Holben et al., 1998). The results discussed and referred to below are shown in Fig. 7 and Tables 6–9.

We first look at modeled clear-sky aerosol optical depth at 550 nm (OD550CS). This is in the model calculated as the all-sky optical depth weighted (at each time step in the simulation) with the clear-sky fraction, and with hygroscopic swelling
calculated from average grid-cell RH. This is the method adapted in K13, while a more common method for simulating the cloud-screened remote sensing assumes hygroscopic swelling based on the clear-sky RH fraction, bur for all-sky conditions (no sampling or weighting). Due to the relatively large coverage (which we here somewhat loosely call global, see Fig. 7 and http://aerocom.met.no for the actual coverage) we find an apparently wide spread for modeled vs. observed (monthly) values, but with a relatively low NMB of -16%, R = 0.59, and Fact2 = 42%. The all-sky values (OD550) look slightly better, with a
positive NMB of 15%, R = 0.64, and Fact2 = 68%. In comparison, CAM4-Oslo has a slightly stronger negative bias for OD550CS, and a slightly smaller, but negative, bias for OD550.

Across the various available observation years 2004–2010, MNB for OD550CS varies between -27% and -6%. Regionally OD550CS is most underestimated in East Asia (NMB = -59%), followed by North America (-56%), Europe (-38%), South
and Central America (-21%) and India (-19%). Europe is also here defined to include sites at the northern coast of Africa. For northern Africa, which again is defined to include sites at the Mediterranean coast in Europe, the bias is however positive (12%). The positive bias is even larger in Australia (71%), where mineral dust also is estimated to dominate, being the most optically thick aerosol. In spite of the apparent underestimation of near-surface dust mass concentrations discussed in Sect. 4.1.2, we may here add that the global all-sky optical depth contribution to mineral dust is biased by 65% (not shown, see
results at http://aerocom.met.no), i.e., much more than the 15% bias for total OD550. It is furthermore clear from Fig. 7 that OD550CS is under-estimated at high latitudes. What is not known, however, is how much of this negative bias that is caused by missing or inaccurate emissions (see e.g., Stohl et al., 2013), and how much of it is a result of other systematic biases such as, e.g., deficiencies in modelling of transport, aerosol chemistry, microphysics and subsequent scavenging or dry deposition.

Comparing OD550CS with the simulated aerosol optical depth from the same 8 AP3 models as in Sect. 4.1.2, we find that 4 of the models have approximately the same (one model) or larger (in absolute value) NMB values than ours. The correlations are higher in 6 of the AP3 models, and 5 models also exhibit higher Fact2 values. Comparing with the 20 models with available data among the 23 AP2 models, we find that only 7 of these are less biased, but the correlations are smaller than ours in only



of the models. Although CAM5.3-Oslo performs well in terms of NMB, the spread is so large that the Fact2 value is lower than in all of these AP2 models.

Moving on to the clear-sky absorption aerosol optical depth at 550 nm (ABS550CS), we similarly find that NMB = -25%, R = 0.47, and Fact2 = 47%. Also here the all-sky values look slightly better, with NMB = -20%, R = 0.47, and Fact2 = 50%. In comparison, CAM4-Oslo has a bit stronger negative biases both for ABS550CS and ABS550.

Across the individual years 2004–2010, MNB for ABS550CS only varies very little, between -41% and -36%. Regionally it is most underestimated in India (-52%), followed by East Asia (-44%), North America (-39%), Europe (defined as above, -29%), South and Central America (-20%), and Australia (-13%). For northern Africa (defined as above), the bias is slightly positive (4%). Regionally the biases in ABS550CS and OD550CS thus mainly have the same sign, which is consistent with too low or too high (depending on the sign of the bias) modeled aerosol burdens. Some exceptions are found, however, such as for Australia as a whole, and in (e.g.) some mineral dust dominated areas over and downwind of the Sahara desert where the absorption optical depth is underestimated while the optical depth (at 550 nm) is overestimated. This may indicate that the assumed imaginary part of the refractive index at 550 nm is too small, or that the effective size of the mineral dust particles is underestimated, which has been identified as a problem in many AeroCom models (Kok et al., 2017).

A few AP2 models and one AP3 model also have absorption data available (ABS550). Comparing with the AP3 model (MNB = -56%) CAM5.3-Oslo is less biased (-38%). Comparing with the 16 models with available data among the 23 AP2 models, we find that only 5 of these are less biased than our model. The correlations are higher than ours in all these AP2 models, however, and 9 models have larger Fact2 values.

Finally we look at the statistics for the clear-sky Ångström parameter, here defined through the clear-sky aerosol optical depths at the wavelengths 440 and 870 nm, OD440CS and OD870CS respectively:

$$ANG4487CS = \frac{-ln(OD870CS/OD440CS)}{ln(870/440)} \tag{7}$$

Globally, for ANG4487CS we obtain NMB = -17%, R = 0.75, and Fact2 = 83%, a quite decent result which indicates that the aerosol size for the clear-sky atmospheric column is fairly well modeled in terms of its relative abundancy of large vs. small particles. The all-sky equivalent ANG4487, however, yields a much poorer match with AERONET, having NMB = -44%, R=0.46, and Fact2=49%. In comparison, CAM4-Oslo has a smaller negative bias for ANG4487 (no clear-sky value is available from that model version), indicating that the effective particle sizes are indeed smaller there than in CAM5.3-Oslo. For all-sky




conditions the aerosol sizes are biased much more towards large particles (small ANG values), which is consistent with higher relative humidities and thus more extensive hygroscopic swelling.

Across the individual years 2004–2010, the bias varies as little as between -15% and -16%. Regionally, ANG4487CS is most

underestimated in Northern Africa (defined as above, -35%), followed by Europe (defined as above, -32%), Australia (-31%), India (-20%), East Asia (-10%), and smallest in North America (-7%). The pattern seen in Fig. 7 seems to point towards dust as the source of the largest negative biases, which is consistent with an excessive mineral dust contribution to the total aerosol (as also indicated by the regional OD550CS biases), or alternatively, overestimated dust particles sizes (opposite of what we found as a potential cause of the positive bias in ABS550CS). For South and Central America NMB = 15%, i.e., an overestimate

indicating slightly too fine particles. This positive bias is smallest (8%) for the SON months, i.e., late in the biomass burning season for the region, while it is largest (22%) for DJF. Since the negative biases for OD550CS and ABS550CS here are smallest (-5% and -6%, respectively) in JJA and largest (-32% and -35%) in DJF, there is still a theoretical possibility that the biomass burning aerosol contribution is exaggerated. This could be the case if contribution from other sources are generally underestimated, e.g., due to missing emissions or exaggerated scavenging. Just based on these results, however, we cannot

conclude whether this is the case or not, and more specifically, whether the assumed OM/OC ratio of 2.6 for biomass burning aerosols is too high or not.

None of the AP3 models but 13 of the AP2 models also have ANG4487 information available at http://aerocom.met.no. Comparing with the 13 AP2 models, we find that 5 have larger biases than in ours. The correlations are smaller in 6 models,

and the Fact2 values are also smaller than ours in 6 models.

Since the particle sizes globally seem to be well represented, based on modeled ANG4487CS, the consistent low bias in OD550CS and ABS550CS, assuming that the intrinsic optical properties and other factors that might affect the result are fairly well represented, imply that modeled aerosol column burdens are underestimated. For the surface concentrations, only BC and

mineral dust are underestimated compared to in situ observations, as discussed in Sect. 4.1.2. However, considering that the available in situ measurements are very sparsely distributed globally and that we know little about the model performance in terms of vertical distribution of mass concentrations (except for BC), we cannot expect these very different measures of model performance to fully agree.

Further aerosol model validation is taking place through ongoing multi-model studies that include results from the present model version. These studies are the AeroCom Control EXPERIMENT 2016, the Remote Sensing evaluation for AeroCom Control 2016, the AeroCom in situ Measurement Comparison (for optical properties) (https://wiki.met.no/aerocom/phase3-experiments), and the BACCHUS CCN global model intercomparison exercise (http://lists.met.no/pipermail/aerocom-modeller/2017-January/000109.html).



### 4.3 Cloud droplet concentrations

We also compare the modeled in-cloud droplet concentration (CDNC) to the dataset provided by Bennartz et al. (2017). To facilitate this comparison, we take out in-cloud droplet concentrations at the cloud top, defined as the first layer – starting from the model top – where the stratiform liquid cloud fraction in a grid-cell exceeds 10% and the temperature criterion of Bennartz et al. (2017) is fulfilled, i.e., 268 K < T < 300 K. The annually averaged result for the NUDGE_PD simulation is given in Fig. 8, which shows that, globally averaged, we calculate lower droplet number concentrations than what is observed. CAM5.3-Oslo mostly under-estimates cloud droplet concentrations over coastal ocean areas in East Asia, Europe and North America. The model over-estimates the droplet concentrations close to mineral dust and biomass burning dominated areas, typically downwind of Saudi Arabia and Africa. The results from AMIP_PD (not shown) are very similar with an average of 49.8 cm$^{-3}$, compared to 51.2 cm$^{-3}$ for NUDGE_PD. One possible reason for the discrepancies between model and observations is that we have not applied a satellite simulator, and the simple way of outputting the droplet concentration described above does not necessarily correspond to what the satellite is seeing. A comprehensive analysis of the discrepancies for the different regions is however beyond the scope of this study.

## 5 Interaction with radiation and clouds

The effective radiative forcing (ERF) of aerosols has been calculated using the method of Ghan (2013), where radiative fluxes for a "clean" (no aerosol extinction), and a "clear" (cloud-free, but including aerosol extinction) atmosphere are used together with the standard all-sky (including aerosol extinction) radiative fluxes in order to decompose the ERF into its separate components. Differences between the PD and PI simulations thus yield the anthropogenic ERF as a "direct radiative forcing", a "cloud radiative forcing" (note that this is the contribution by anthropogenic aerosols, not the total cloud forcing itself), and a "surface albedo forcing" term. We here only show and discuss results for the cloud forcing and the direct radiative forcing components. The surface albedo forcing is small on a global scale, and is not discussed. Neither is the semi-direct effect of aerosols, which is included as part of the cloud radiative forcing term (Ghan, 2013) but not calculated and shown separately, since this particular diagnostic requires extra sets of simulations in which (potentially) anthropogenic aerosols are assumed to be totally non-absorptive (Ghan et al., 2012). Results from such simulations with an earlier, slightly differently tuned model version, suggest that the semi-direct effect in CAM5.3-Oslo contributes very little to the total aerosol ERF, however. The SW + LW semi-direct radiative forcing was there estimated to be -0.02 W m$^{-2}$, globally averaged.

Figure 9 shows modeled shortwave (SW) and long-wave (LW) direct radiative forcing at the top of the atmosphere (TOA), annually averaged, both from the nudged simulations (i.e., NUDGE_PD – NUDGE_PI) and the longer AMIP simulations (AMIP_PD – AMIP_PI). Global averages are listed and compared to estimates from CAM4-Oslo (direct RF) and IPCC AR5 (direct RF and ERF) in Table 10. Regionally, the SW direct forcing is positive over some areas with high surface albedo or high cloud fractions for low clouds, mainly related to biomass burning activity which compared to PI conditions has led to




increased levels of light absorbing aerosols such as fossil fuel or biomass burning BC (e.g., Sahara, the Arctic, and off the west coasts of South America and Africa). The direct forcing term is also positive in some areas with reduced absorption, where the scattering aerosol optical depth (mainly from OM) has decreased even more (such as in eastern USA and parts of Australia and South America). However, negative SW direct forcing is dominating over the industrialized parts of the world, due to the

general increase in scattering aerosol of anthropogenic origin (sulfate and OM). The global and annual average is estimated at -0.095 W m$^{-2}$ (-0.092 W m$^{-2}$ for the AMIP simulations). The LW direct forcing is much smaller, having a regional maximum over the Middle East where both large mineral dust and (internally mixed) anthropogenic aerosol are abundant: the sulfate column burden has a local maximum in this region. The global and annual average is here 0.026 W m$^{-2}$ both for the NUDGE and the AMIP simulations. Just as for CAM4-Oslo, the estimated total (joint SW and LW) global direct radiative forcing in

CAM5.3-Oslo lies within the range of the ERF ARI estimates of IPCC AR5 (Boucher et al., 2013), see Table 10. Since the AR5 range has been evaluated for the period 1750–2011 and ERF ARI there includes the semi-direct effect, the numbers are not entirely comparable, however.

Figure 10 similarly shows the shortwave (SW) and long-wave (LW) cloud radiative forcing (due to anthropogenic aerosols)

at TOA. Also here we obtain positive SW forcing in some areas, mainly in the SH subtropics and at high latitudes, consistent with the lower PD than PI cloud droplet concentrations (CDNC) and liquid water path (LWP) found in these areas. Some of the positive cloud forcing is due to a reduction in the organic emissions from biomass burning since 1850 (e.g., in England, Australia, and the Eastern United States). Also over the Southern Ocean there are areas with slightly positive values, coinciding with areas with slightly smaller column vertically integrated CDNC and LWP values in the PD than in the PI simulations. This

pattern has been found to be even more prominent when the PI simulations apply PI oxidant levels (instead of PD as in this study), see Karset et al. (2018) for a more thorough discussion on the effect of different oxidant levels on the cloud forcing. Areas with a negative SW cloud forcing term are however dominating due to the general increase in CDNC from PI to PD conditions, being large (negative) over oceans downstream areas with high aerosol emissions from industrial activity, biofuel consumption or biomass burning. The negative SW cloud forcing peaks over the northern Pacific Ocean, near the coast of East

Asia. The global and annual average value is estimated at -1.50 W m$^{-2}$ (-1.45 W m$^{-2}$ for the AMIP simulations). The LW cloud forcing is smaller and is in most regions of opposite sign to the SW contribution. Its global and annual average is 0.161 W m$^{-2}$ (0.155 W m$^{-2}$ for the AMIP simulations). Compared to the ERF ACI estimates in Table 10, the total global cloud radiative forcing in CAM5.3-Oslo is thus on the high side, lying just outside the 5 to 95% confidence range given by IPCC AR5. As mentioned, the AR5 range in Table 10 is for the period 1750–2011 instead of 1850–2000. Compared to this extended period

we should expect a somewhat smaller negative forcing contribution, since the reference state in 1850 is less pristine than in 1750 while changes due to aerosols in the latter part of the period are of less importance (Carslaw et al., 2013). For the same time period, however, we should expect a stronger negative cloud forcing than that of AR5 since the second indirect effect is included in our model (although not calculated separately), whereas the ERF ACI range in IPCC AR5 is (mainly) for the first indirect effect.



The expert judgment 5 to 95% (medium confidence) uncertainty range for ERF ARI + ERF ACI is in IPCC AR5 estimated to be -1.9 to -0.1 W m$^{-2}$, while the 17–83% (likely) range is -1.5 to -0.4 W m$^{-2}$ (Boucher et al., 2013). These estimates take into account that the GCM studies calculate stronger aerosol ERF values than what is found in satellite studies. Our model values

-1.4 W m$^{-2}$ (NUDGE_PD – NUDGE_PI) and -1.36 W m$^{-2}$ (AMIP_PD – AMIP_PI) lie within both ranges of uncertainty. Our model estimates are also very close to one standard deviation away from the multi-model estimate for the period 1850–2000 in Boucher et al. (2013), which is given as -1.08 ± 0.32 W m$^{-2}$, based on results from CMIP5 and ACCMIP (Atmospheric Chemistry and Climate Model Intercomparison Project) models.

Table 11 lists some globally and annually averaged variables relevant for understanding the above estimates of effective radiative forcing by aerosols, both for the NUDGE and AMIP simulations. Although the globally averaged all-sky aerosol optical depth at 550 nm (OD550) for PD is found to be larger than in CAM4-Oslo (0.135, see Table 7 in K13), we now obtain an anthropogenic (PD-PI) AOD which is 29% smaller than in CAM4-Oslo, mainly due to lower atmospheric residence times and burdens of sulfate, BC and OM. The simulated anthropogenic AOD fraction of total AOD (about 16% in both NUDGE_PD

and AMIP_PD) are therefore considerably smaller than in CAM4-Oslo (26%), which is about the same as in the average AeroCom Phase I model (25%, Schulz et al., 2006). Anthropogenic absorption AOD (ABS) is about 40–45% smaller than in CAM4-Oslo (0.020), and the anthropogenic ABS fraction is estimated at about 25%, compared to 43% in CAM4-Oslo. Considering that the anthropogenic absorption optical depth has decreased more (39%) than the anthropogenic optical depth itself (28%), one would perhaps expect a more negative direct radiative forcing in CAM5.3-Oslo. It is instead found to be

nearly the same, -0.095 (or -0.092 for the AMIP simulations) vs. -0.10 W m$^{-2}$ (as an instantaneous direct forcing) globally averaged. This can be partly understood as an effect of the substantial increase in the cloud fraction (and thus planetary albedo) for low clouds, 0.43 vs. 0.34, with the largest increase found at mid- to high latitudes. The surface albedo is also higher in CAM5.3-Oslo, 0.163 vs. 0.156 in CAM4-Oslo. Regionally the largest increases (> 0.1) are also here found at mid- and high latitudes, over continents in the NH. The shift towards smaller anthropogenic BC concentrations and to lower altitudes (Fig.

5), which reduces the absorption in the atmospheric column and therefore leads to a less positive direct RF (e.g., Samset et al., 2013), is in other words counteracted by the effect of increased surface or near-surface albedos from CAM4-Oslo to CAM5.3-Oslo. The reduction in anthropogenic atmospheric absorption is reflected in the difference in SW direct radiative forcing between the TOA and the surface, which in CAM5.3-Oslo is estimated at 0.51 W m$^{-2}$ for NUDGE and 0.47 W m$^{-2}$ for AMIP, compared to 0.95 W m$^{-2}$ in CAM4-Oslo (K13).

In-cloud cloud droplet number concentrations and effective droplet radii are here defined differently than in CAM4-Oslo, by (for each time step) weighting the respective model variables with the stratiform liquid cloud fraction (a number between 0 and 1) in CAM5.3-Oslo, instead of the frequency of cloud occurrence (being either 0 or 1). These two model parameters are therefore not directly comparable between the two model versions. We can see, however, that the vertically integrated liquid


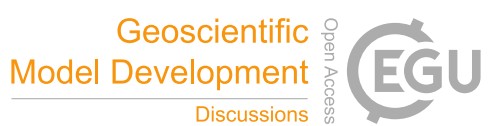

water path (LWP) in CAM5.3-Oslo (~54 g m$^{-2}$) is much smaller than in CAM4-Oslo (~130 g m$^{-2}$, see K13). Some of this drop in LWP may be due to the changes in aerosol treatment, but the relative low sensitivity of LWP to aerosol concentration levels (Table 11, and Table 4 in K13) suggests that much of it is a result of switching from the RK cloud microphysics scheme (Rasch and Kristjánsson, 1998) in CAM4-Oslo to MG1.5 (see Sect. 3) in CAM5.3-Oslo, and how the respective schemes are tuned.

This may have contributed to an increase in the modeled cloud susceptibility (Albrecht, 1989), thus leading to enhanced cloud forcing by anthropogenic aerosols. A more thorough investigation of this falls outside the scope of this study, and has not been pursued.

The size and even sign of the Albrecht (life-time) effect is very uncertain, and has in a recent observationally based study been

shown to be small, or more specifically, not detectable above the level of natural variability for the Holuhraun volcanic eruption (Malavelle et al., 2017). In CAM5.3-Oslo the anthropogenic change in LWP is estimated to be about 3.56 g m$^{-2}$ in NUDGE_PD – NUDGE_PI (3.42 g m$^{-2}$ in the AMIP simulations). Compared to 4.37 g m$^{-2}$ in CAM4-Oslo (K13), this constitutes a much larger relative change in LWP, being 6.6% (6.4%) instead of 3.4%. The life-time effect was in CAM4-Oslo calculated as a radiative forcing, however, by use of double calls to both the radiation and stratiform cloud microphysics modules, following

Kristjánsson (2002). Since the cloud cover is independent of liquid water content (mainly depending on RH), that approach does not take into account changes in cloud life-time from changes in the cloud cover, which may result in a low-end estimate of the indirect effect (Kristjánsson, 2002). The relative (anthropogenic divided by total) change in vertically integrated CDNC is about the same in CAM5.3-Oslo (21% both in NUDGE and AMIP) and in CAM4-Oslo (21%, not shown). Hence, a considerable part of the increase in cloud effective radiative forcing from -0.90 W m$^{-2}$ to -1.34 W m$^{-2}$ is probably due to the

very uncertain life-time indirect effect.

Since the modeled ice crystal number concentrations (ICNC) can be directly affected by aerosols only through hetero-geneous freezing of mineral dust and BC in mixed-phase clouds, it is quite insensitive to anthropogenic aerosols. Vertically integrated ICNC is practically unchanged from PI to PD in both the NUDGE and AMIP simulations (Table 11), so that the effect of this

on the total cloud radiative forcing is probably negligible.

## 6 Summary and conclusions

We have described in quite some detail changes in the treatment of aerosols and aerosol–cloud interactions in going from the predecessor model version CAM4-Oslo (Kirkevåg et al., 2013; Iversen et al., 2013) to CAM5.3-Oslo. In broad terms the changes consist of taking explicitly into account nucleation and secondary organic aerosols (based on Makkonen et al., 2014),

using new sea-salt emissions and emission sizes (Salter et al., 2015), applying interactive DMS and primary organics emissions by use of prescribed ocean surface layer concentration and wind driven parameterizations (Nightingale et al., 2000; Vignati et al., 2010), and now also online dust emissions (Zender et al., 2003). Aerosol hygroscopicity and some other microphysical





properties have also been updated, and heterogeneous ice nucleation has been implemented based on Wang et al. (2014). An updated overview of the main principles behind the production tagged aerosol module, which is used in CAM5.3-Oslo and a number of predecessor versions, has also been presented.

We have furthermore made an attempt to validate CAM5.3-Oslo with respect to its simulated aerosol properties and aerosol cloud interactions by comparing monthly model output with in situ observations and remote retrievals. This is meant to complement more thorough several ongoing inter-comparison studies, mainly under the AeroCom project (see http://aerocom.met.no), which focus on various model diagnostics at monthly as well as finer time resolutions (down to 1 h) using results from the same model version as in this study, along with other AeroCom models.

It is shown that the simulated vertical profile of BC concentrations is more realistic in CAM5.3-Oslo than in CAM4-Oslo, when comparing to in situ measurements from the HIPPO Aircraft campaign in the Pacific Ocean. The new model version produces much less excessive BC mass concentrations in the upper troposphere and in the stratosphere, although the concentrations are still overestimated at the highest altitudes. This may be related to aerosol aging and to how aerosols are

transported and scavenged in deep convective clouds: also the mass concentrations of the other aerosol components have been reduced (aloft) from CAM4-Oslo to CAM5.3-Oslo. This issue is to a large degree dependent on the choice of host model (which is CAM5.3 in this case), and will most likely continue to be an area of focus in future research and development of the model.

With a ca. doubled DMS emission and column burden, and a subsequent increase in the $SO_2$ source term, near surface mass concentrations of $SO_2$ now seem to be considerably over-estimated (normalized mean bias NMB ~ 150%) compared to in situ observations available via the AeroCom intercomparison project (http://aerocom.met.no), more so than in CAM4-Oslo. However, the modeled concentrations are not adjusted with respect to representative height above the ground surface before comparing with observations, which is an important factor for $SO_2$ and therefore hampers reliable evaluation of the model

performance.

Near surface sulfate concentrations are biased slightly high (22%), more so than in CAM4-Oslo (-5%), which instead exhibits a slightly lower Pearson correlation coefficient, R. All of the 8 AeroCom Phase III (AP3) models with available information at aerocom.met.no have higher correlations, and half of them also have smaller (in absolute value) biases. The sulfate

concentrations in CAM5.3-Oslo are found to be less biased than only 6 of the 23 available AeroCom Phase II (AP2) models, although with similar or better correlations than 14 of the models.

Near surface BC concentrations are mainly biased low (-28%), but less than in CAM4-Oslo (-54%), which together with the more realistic vertical profiles indicate an improvement in modelling of BC. The bias is also found to be smaller than in 6 of



8 AP3 models, but only in 7 of the 23 AP2 models. The correlation values lie within the ranges spanned by the AP2 and AP3 models, although in the lower range for both AeroCom phases.

Since OsloAero5.3 (like earlier module versions) does not trace OM from different source types and thus different assumed

OM/OC ratios separately, a reliable evaluation of the modeled mass concentrations for OM cannot be obtained without doing further work with this particular aim in mind. However, if we simply assume that the OM/OC ratio in the model is 1.4 or 2.6 for all OC, which is assumed to be representative for cases with *no* biomass burning and *only* biomass burning emission sources, respectively, we find respective biases of 122% or 19%, compared to 108% or 12% in CAM4-Oslo. Unless the sparsely distributed in situ observation data represent OC very poorly globally (which is a possibility since only North America

and Europe are represented), these results do indicate an overestimation which is now slightly larger than in the predecessor model, despite the increased level of sophistication in the new parameterization of SOA and primary biogenic OM emissions from the ocean. The correlation value of 0.29 is just below that of CAM4-Oslo. The correlation is also lower than in most of the AP2 models and all of the AP3 models. If we assume that OM/OC = 1.4, the bias is also larger than in the AP3 models and in all but one of the AP2 models. Although CAM4-Oslo apparently performs slightly better in this particular evaluation, and

for the current OM/OC ratio assumption, we should keep in mind that both SOA (treated as primary OM) and biogenic OC from the ocean use prescribed emissions, rendering that model version less useful for Earth System Modelling and studies of past and future climates, as well as for more detailed process studies and sensitivity studies in general.

The sea-salt aerosol concentrations are found to have a bias of 22%, which is an improvement compared to CAM4-Oslo,

although the correlation is slightly lower. Both model versions apply wind and temperature dependent emissions, but CAM5.3-Oslo is using particle size parameters at the point of emission which are closer to observed values in (and fully consistent with) the updated treatment. Our model outperforms the AP3 models bias-wise and has the second highest Pearson correlation. It also ends up among the best in comparison to the AP2 models.

The surface concentrations of mineral dust are biased low by -39%, but with a decent correlation of 0.52. The available observation sites are not representative for the source regions of dust, however, and we have reasons to believe that the negative bias is a result of an underestimate in dust transport rather than in the emissions, see the summary for aerosol optics below. The dust concentrations have quite large year to year variations, and differ the most between the nudged and the free running AMIP simulations, where the bias is smaller. Compared to the 8 AP3 models, CAM5.3-Oslo performs better than 3 bias-wise

and 4 with respect to correlation. Compared to the 23 AP2 models it performs better than 14 models bias-wise, but only 7 with respect to correlations. CAM4-Oslo is less biased, but uses prescribed dust emissions and is therefore less applicable for Climate and Earth System Modelling studies.



We have also compared column integrated optical parameters with estimates from other models, and most importantly, with ground-based remote sensing data (AERONET). Looking first at the modeled mass extinction coefficients (MEC), we find changes in all components compared to CAM4-Oslo: a ca. 13% decrease in MEC for sulfate and 30% for OM, while it has increased by ca. 17% for BC, 18% for mineral dust, and as much as 63% for sea-salt, where considerable changes in assumed
particle size at the point of emission have had a large impact. The new estimates are all within the range of models that participated in AeroCom Phase I. The globally averaged mass absorption coefficient (MAC) for BC is smaller or larger than in the predecessor and in observations, depending on how it is being calculated. The practice for evaluating this parameter in climate models is to our knowledge not standardized for internally mixed aerosols, and is often estimated based on the assumption that BC is the only aerosol component which contributes to absorption. This approach yields high MAC values of
about 21-23 $m^2 \, g^{-1}$ in CAM5.3-Oslo and 13.6 $m^2 \, g^{-1}$ in CAM4-Oslo, globally averaged. Adopting the more realistic assumption that also mineral dust and OM contribute to the absorption, a lower bound of the globally averaged modeled MAC is estimated to be approximately 5 $m^2 \, g^{-1}$. If we take this lower bound as a representative model value, it just touches the lower end of a recommended range of 5 to 11 $m^2 \, g^{-1}$ based on in situ measurements. However, even here we find areas regionally where MAC exceeds the recommended central value of 7.5 $m^2 \, g^{-1}$.

Comparing clear-sky aerosol optical depth at 550nm (OD550CS) with remote retrieved values from AERONET sun-photometer stations world-wide, we find a negative bias of -16% globally, compared to -22% in CAM4-Oslo. The respective all-sky bias for CAM5.3-Oslo is positive, 15%. OD550CS is generally biased low at high NH latitudes and high over and downstream of major mineral dust emission areas. Compared to the 8 AP3 models, half of these have smaller bias values
globally, while 6 perform better than CAM5.3-Oslo with respect to correlations. Compared to 20 AP2 models, only 7 of these have lower biases, while correlations are higher in 17 of the models.

For clear-sky absorption optical depth (ABS550CS) there is a slightly stronger negative bias, -25%, but smaller than in CAM4-Oslo. The all-sky model variable is slightly less biased. The ABS550CS bias is of same sign and roughly the same magnitude
as for OD550CS for most regions world-wide. The one AP3 model with data available has a stronger low bias, and only 5 of 16 AP2 models have smaller biases than CAM5.3-Oslo. All of these AeroCom models yield better correlation values, however.

The clear-sky Ångström parameter (ANG4487CS) is found to have a relatively small negative bias globally, -17%, while the all-sky variable has a much stronger negative bias. ANG4487CS is most underestimated in northern Africa, which is consistent
with exaggerated dust emissions. Comparing with 13 AP2 models, CAM5.3-Oslo is outperformed by 7 models bias-wise and 5 models with respect to correlation.

In an attempt to also evaluate an important aspect of the cloud microphysics with respect to calculation of cloud–aerosol interactions, we have made compared modeled droplet concentrations (CDNC) at the cloud top with remotely retrieved CDNC



from MODIS. This is done for ocean areas only, but these are the areas contributing most to the global effective radiative effect due to aerosol–cloud interactions. While overestimating droplet concentrations downwind of major emissions of mineral dust and biomass burning aerosols, CAM5.3-Oslo (in NUDGE_PD) mainly under-estimates CDNC over the other coastal areas in East Asia and North America. This might be related to biases in aerosol concentrations in the respective continental

source regions, but this cannot be known for sure as long as we only have near surface concentrations for very limited areas covered to compare with, and only mass (not number) concentrations. The largest regional biases in OD550CS from AERONET, which have mainly continental sites, seem to be consistent with the positive biases in CDNC, however. "Globally averaged" (low- to mid-latitude ocean grid points only), cloud top CDNC has a low bias of -32%.

Finally, we have presented and discussed model estimates of effective radiative forcing (ERF) by anthropogenic aerosols, for comparison with previous radiative forcing (RF) results from CAM4-Oslo and RF and ERF estimates from IPCC AR5. Globally averaged the SW direct effect is estimated at -0.095 W m$^{-2}$, compared to -0.100 W m$^{-2}$ in CAM4-Oslo. The LW direct effect was not taken into account in CAM4-Oslo, and is in CAM5.3-Oslo estimated to be 0.026 W m$^{-2}$. The joint SW and LW direct effective radiative forcing (-0.069 W m$^{-2}$) lies well within the range of estimates in IPCC AR5. The effective radiative

cloud forcing due to anthropogenic aerosols is for SW and LW radiation estimated at -1.50 W m$^{-2}$ and 0.16 W m$^{-2}$, respectively, compared to -0.91 W m$^{-2}$ and 0.01 W m$^{-2}$ in CAM4-Oslo. The joint SW and LW cloud forcing by anthropogenic aerosols in CAM5.3-Oslo (-1.34 W m$^{-2}$) is at the lower end of the 5–95% confidence interval of IPCC AR5 based on model and satellite studies, but lies just within one standard deviation of the reported multi-model range of CMIP5 and ACCMIP models.

Whether we use the short (7 year) simulations which have been nudged to ERA-Interim meteorology or the longer (30 years) free AMIP simulations does not make much of a difference for the global averaged results, e.g., for the ERF estimates (only 4% weaker total aerosol ERF in the free running simulations). Regionally differences are larger, however, both for ERF estimates and for anthropogenic contributions to model fields in general (i.e., differences PD – PI).

After the simulations for use in this study were finalized, it was found that the median radius for mixture no. 12 (Aitken mode BC) with respect to dry deposition had not been increased to the new number in Table 2, as intended. Instead the old value of 0.0118 µm (K13) has been used. This only affects the dry deposition (in OsloAero5.3), while the treatment of aerosol optics and sizes for use in cloud droplet activation (in AeroTab5.3, as well as in the look-up tables and the use of those in the model) is correct and unaffected. The impact of the bug has been tested by rerunning two of the least time-consuming simulations

(NUDGE_PI and NUDGE_PI) with the bug fixed. This reveals that the code which has been used in this study has underestimated the BC lifetime and column burden by about 9%, and the direct effective radiative forcing by 0.02 W m$^{-2}$, globally averaged. Since the bug affects only a small part of the results discussed in this study, and since the exact same model version has been used in several ongoing AeroCom Phase III inter-comparison experiments (with additional simulations with finer time resolved model output), we have decided to keep this model version and results as they are for this particular study.



In addition to correcting this bug for BC, the presented results suggest that we should re-tune (reduce) the dust emissions strength in future work with CAM5.3-Oslo, in order to better match remotely retrieved aerosol optical depths over the most dust dominated areas. The somewhat surprisingly small changes in OM results (including the validation) compared to the predecessor model, where the SOA treatment is very simplistic, should also be investigated in more detail. Work on vertical

transport as well as aerosol cloud interactions in convective clouds are other areas of great interest.

*Code and data availability.* The source code for CAM5.3-Oslo is stored within the NorESM development repository. The code and simulation output data produced in this study can be made available within the framework of a user agreement. The CAM4-

Oslo and CAM5.3-Oslo data in Tables 5–8 and Figs. 6 and 7 are available from the AeroCom data base at http://aerocom.met.no, under the Project label NorESM, Subset NorESM-Ref2017. Most of the discussed model data (in the form of tables and figures) are furthermore available at http://ns2345k.web.sigma2.no/nudged_NorESM_c12, see especially 53OSLO_PDandPIwPDoxi_vs_AMIP_PDandPIwPDoxi for comparisons of NUDGE_PD with AMIP_PD and NUDGE_PD – NUDGE_PI with AMIP_PD – AMIP_PI.

*Competing interests.* The authors declare that they have no conflict of interest.

*Acknowledgements:* This study has been financed by the research council of Norway (RCN) through the project EVA (229771) and the NOTUR/Norstore projects (Sigma2 accounts nn2345k and nn9448k, and Norstore account NS2345K), by the Nordic
projects eSTICC (57001) and CRAICC (26060), and by the EU projects BACCHUS (FP7-ENV-2013-603445), CRESCENDO (641816), and IS-ENES2. X. Liu was supported by the Office of Science of the US Department of Energy as part of the Earth System Modeling Program. The general NorESM1.2/CAM5.3-Oslo model development has also benefited from contributions by other scientists, affiliated with NCAR and PNNL in USA, member institutions of The Norwegian Climate Centre: BCCR, MET Norway, MetOs-UiO, NERSC, Cicero, NILU and NP; MISU, as well as The Bolin Centre in Sweden and the University
of Helsinki, Finland. Special thanks go to NCAR for granting early access to development versions of CESM, to the AeroCom community for making their model data available at aerocom.met.no, and to Prof. Jón Egill Kristjánsson at the Department of Geosciences, University of Oslo (UiO), for his dedicated work on cloud microphysics and aerosol–cloud interactions and his leading role in this field at UiO until he passed away on August 14'th 2016.

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



**Table 1:** Aerosol tracers included in OsloAero5.3.

| Tracer variable id | Meaning<br>S4=SO$_4$=particulate sulfate, SOA=secondary organic aerosols; BC=black carbon; OM=primary organic matter; SS=Sea-salt; DU=DST=mineral dust. | Notation in Fig. 1 |
|---|---|---|
| SO4_NA | SO$_4$ formed by co-nucleation with SOA | S4(n) |
| SO4_A1 | SO$_4$ condensate on existing particles from H$_2$SO$_4$(gas) | S4 (yellow) |
| SO4_A2 | SO$_4$ formed from aqueous phase chemistry | S4 |
| SO4_AC | SO$_4$-particles coagulated with other particles | S4(ac), S4(c) |
| SO4_PR | SO$_4$ primary emissions - emitted as particles | S4(ac) |
| SOA_NA | SOA formed by co-nucleation with SO$_4$ | SOA(a) |
| SOA_A1 | SOA condensate on existing particles from SOAG$_{sv}$(gas) | SOA (yellow) |
| BC_N | BC emitted externally mixed as nucleation sized mode | BC(n) |
| BC_AX | BC emitted externally mixed as fractal accumulation mode | BC(ac) |
| BC_NI | BC emitted internally mixed with OM, aitken mode | OM/BC(a) |
| BC_A | BC coated with water-solubles, aitken mode | OM/BC(a) |
| BC_AI | BC co-existing with OM and coated aitken mode | OM/BC(a), BC(a) |
| BC_AC | BC particles coagulated with other aerosols (coagulate) | BC(ac), BC(c) |
| OM_NI | OM emitted internally mixed with BC, aitken mode | OM/BC(a) |
| OM_AI | OM co-existing with BC and coated, aitken mode | OM/BC(a) |
| OM_AC | OM and SOA particles coagulated with other aerosols (coagulate) | OM(ac), OM(c), SOA(ac), SOA(c) |
| DST_A2 | Mineral dust, accumulation mode | DU(ac) |
| DST_A3 | Mineral dust, coarse mode | DU(c) |
| SS_A1 | Sea-salt aerosol, aitken mode | SS(a) |





| SS_A2 | Sea-salt aerosol, accumulation mode | SS(ac) |
|-------|-------------------------------------|--------|
| SS_A3 | Sea-salt aerosol, coarse mode | SS(c) |



**Table 2:** Distribution of aerosol tracers in the particle mixtures treated in the model. Shaded cells indicate background tracers. The cells which are not shaded, indicate tracers which modify the size distribution. The initial number median radius (NMR) and standard deviation (SIGMA) of each background mode are listed in the second and third column. Note that for historical reasons, no particle mixture number 3 and 13 exist in the present model version. For the sake of consistency and transparency, the numbering is the same as in the model code.

| Particle mixture no. | NMR [μm] | SIGMA | Aerosol tracers (cf. Table 1) contributing to the particle mixture | | | | | |
|---|---|---|---|---|---|---|---|---|
| 0 | 0.0626 | 1.6 | BC_AX | | | | | |
| 1 | 0.0118 | 1.8 | SO4_NA | SOA_NA | SO4_A1 | SOA_A1 | | |
| 2 | 0.024 | 1.8 | BC_A | SO4_A1 | SOA_A1 | | | |
| 4 | 0.04 | 1.8 | OM_AI | BC_AI | SO4_A1 | SO4_A2 | SOA_A1 | |
| 5 | 0.075 | 1.59 | SO4_PR | BC_AC | OM_AC | SO4_A1 | SO4_AC | SO4_A2 | SOA_A1 |
| 6 | 0.22 | 1.59 | DST_A2 | BC_AC | OM_AC | SO4_A1 | SO4_AC | SO4_A2 | SOA_A1 |
| 7 | 0.63 | 2.0 | DST_A3 | BC_AC | OM_AC | SO4_A1 | SO4_AC | SO4_A2 | SOA_A1 |
| 8 | 0.0475 | 2.1 | SS_A1 | BC_AC | OM_AC | SO4_A1 | SO4_AC | SO4_A2 | SOA_A1 |
| 9 | 0.3 | 1.72 | SS_A2 | BC_AC | OM_AC | SO4_A1 | SO4_AC | SO4_A2 | SOA_A1 |
| 10 | 0.750 | 1.6 | SS_A3 | BC_AC | OM_AC | SO4_A1 | SO4_AC | SO4_A2 | SOA_A1 |
| 12 | 0.024 | 1.8 | BC_N | | | | | |
| 14 | 0.04 | 1.8 | OM_NI | BC_NI | | | | |





**Table3:** Overview of the experiments in this study. Note that the land model (CLM4.5) set-up is for a PD climate, so that BVOC emissions are based on PD land-use. All simulations have been run with 0.9°x1.25° horizontal resolution and with 30 layers in the vertical.

| Name | Meteorology | Emission year | years simulated (years analyzed) |
|------|-------------|---------------|----------------------------------|
| NUDGE_PD | ERA-Interim | 2000 | 2004–2010 (2006–2010) |
| NUDGE_PI | ERA-Interim | 1850 | 2004–2010 (2006–2010) |
| AMIP_PD | CAM5.3-Oslo/AMIP | 2000 | 1–30 (3–30) |
| AMIP_PI | CAM5.3-Oslo/AMIP | 1850 | 1–30 (3–30) |





**Table 4:** Aerosol budgets for the different components in the ERA Interim nudged and the AMIP (shown in square brackets) simulations for year 2000 (PD) and 1850 (PI) emissions. Emission and burdens for DMS, $SO_2$ and $SO_4$ are given as Tg(S) yr$^{-1}$. For each cell the upper row shows results from NUDGE_PD [AMIP_PD], and the lower row shows results from NUDGE_PI [AMIP_PI]. The burdens are calculated from interstitial aerosols only. Results in round brackets are from the PD 2000 experiment in Kirkevåg et al. (2013), for comparison.

| | DMS | $SO_2$ | $SO_4$ | Dust | Sea-salt | BC | OM |
|---|---|---|---|---|---|---|---|
| **Emissions (Tg)** | 34.3 [34.6] (18.1) | 65.0 [65.0] (66.3) | 1.67 [1.67] (1.70) | 3104 [2508] (1672) | 1937 [2003] (6462) | 7.93 [7.93] (7.70) | 86.9 [87.4] (122) |
| | 34.3 [34.7] (18.1) | 14.6 [14.6] (16.4) | 0.373 [0.373] (0.42) | 3135 [2552] (1672) | 1937 [2005] (6462) | 3.15 [3.15] (3.06) | 61.4 [61.9] (97.5) |
| **Chemical sources (Tg yr$^{-1}$)** | N/A | 31.5 [31.8] (13.2) | 56.2 [56.4] (62.2) | N/A | N/A | N/A | 87.3 [83.2] (16.2) |
| | N/A | 31.5 [31.9] (13.2) | 26.5 [26.3] (23.2) | N/A | N/A | N/A | 89.5 [85.3] (15.5) |
| **Dry dep. (% of sinks)\*** | N/A | 22.5 [22.5] (23.0) | 13.2 [12.9] (8.4) | 80.7 [80.7] (74.8) | 43.6 [43.3] (54.6) | 24.8 [23.5] (28.1) | 13.6 [13.0] (21.4) |
| | N/A | 17.7 [18.0] (10.5) | 13.3 [13.1] (6.3) | 80.5 [80.5] (74.8) | 43.5 [43.2] (54.6) | 21.7 [20.7] (27.3) | 13.2 [12.8] (22.4) |
| **Wet dep. (% of sinks)** | N/A | 19.3 [19.2] (7.9) | 86.8 [87.1] (91.6) | 19.3 [19.3] (25.2) | 56.4 [56.7] (45.4) | 75.2 [76.5] (71.9) | 86.4 [87.0] (78.6) |
| | N/A | 24.6 [24.2] (11.1) | 86.7 [86.9] (93.7) | 19.5 [19.5] (25.2) | 56.5 [56.8] (45.4) | 78.3 [79.3] (72.7) | 86.8 [87.2] (77.6) |
| **Chemical loss (%)** | 100 [100] (100) | 58.2 [58.3] (69.1) | N/A | N/A | N/A | N/A | N/A |
| | 100 [100] (100) | 57.7 [57.8] (78.4) | N/A | N/A | N/A | N/A | N/A |
| **Lifetime (days)** | 1.48 [1.50] (2.39) | 1.35 [1.33] (1.11) | 3.70 [3.65] (3.80) | 1.92 [1.93] (2.55) | 1.07 [1.04] (0.28) | 4.98 [4.77] (8.12) | 5.13 [4.84] (7.58) |
| | 1.48 [1.50] | 1.25 [1.26] | 3.25 [3.21] | 1.91 [1.94] | 1.07 [1.04] | 5.03 [4.83] | 4.87 [4.62] |





| | | | | | | | |
|---|---|---|---|---|---|---|---|
| | (2.39) | (1.07) | (3.21) | (2.55) | (0.28) | (7.12) | (7.32) |
| **Burden (Tg)** | 0.140 [0.143] (0.12) | 0.357 [0.352] (0.24) | 0.584 [0.579] (0.59) | 16.3 [13.3] (11.7) | 5.70 [5.72] (4.94) | 0.108 [0.103] (0.17) | 2.44 [2.26] (2.87) |
| | 0.140 [0.144] (0.12) | 0.158 [0.160] (0.087) | 0.239 [0.238] (0.21) | 16.4 [13.6] (11.7) | 5.67 [5.71] (4.94) | 0.043 [0.042] (0.060) | 2.01 [1.86] (2.27) |

* Calculated as 100% – chemical loss (%) – wet deposition (%)





**Table 5**: Seasonal and annual normalized mean biases (NMB) and Pearson correlation coefficients (R) for NUDGE_PD vs. observed climatological surface concentrations (see http://aerocom.met.no, cf. Fig. 6).

| | BC | | SO$_2$ | | SO$_4$ | | OM (OA) | | SS | | DUST | |
|---|---|---|---|---|---|---|---|---|---|---|---|---|
| | NMB | R | NMB | R | NMB | R | NMB | R | NMB | R | NMB | R |
| **DJF** | -53% | 0.32 | 154 | 0.45 | -19% | 0.66 | -34% | 0.31 | 20% | 0.49 | -8.4% | 0.43 |
| **MAM** | -21% | 0.47 | 124 | 0.23 | 19% | 0.69 | 63% | 0.44 | 13% | 0.57 | -39% | 0.82 |
| **JJA** | 8.2% | 0.61 | 143 | 0.21 | 46% | 0.87 | 294% | 0.37 | 28% | 0.59 | -52% | 0.47 |
| **SON** | -28% | 0.38 | 180 | 0.26 | 31% | 0.70 | 96% | 0.25 | 26% | 0.53 | -42% | 0.45 |
| **ANN** | -28% | 0.38 | 150% | 0.35 | 22% | 0.72 | 122% | 0.29 | 22% | 0.54 | -39% | 0.52 |




**Table 6:** Normalized mean bias (NMB, in %) statistics from one year of monthly European and Global data (see AeroCom web interface for details on coverage and networks). Compared are NMB for the near surface aerosol mass concentrations and column integrated optical properties for CAM5.3-Oslo, as well as for CAM4-Oslo and AeroCom models in the aerocom.met.no data base (here represented by a NMB range). The top row indicates the meteorological year for observations and nudged simulations, where climatology means that all available years from the model or observations are used for the statistics. The regional coverage areas for observations are abbreviated as follows: E = Europe, N = North America, A = Asia, and Global = nearly all continents (or world oceans for the case of dust) are represented. The control version of the AeroCom Phase II (AP2) and Phase III (AP3) models used in the model inter-comparison are listed below the table, with names as on the AeroCom web interface. Optics diagnostics listed for the AP2 and AP3 models are mostly clear-sky values, in the sense that the clear-sky humidity of the grid cell is used for calculating the hygroscopic swelling. CAM4-Oslo and CAM5.3-Oslo compute all-sky optical properties using the average humidity of the grid cell. Clear-sky (CS) properties are represented by a cloud fraction weighted average of the all-sky properties. Data from CAM4-Oslo and the two simulations with CAM5.3-Oslo, all run with 2000 (PD) emissions, can be found in the aerocom.met.no data base under the Project label NorESM, Subset NorESM-Ref2017.

| NMB (%) | Climatology | | | 2006 | | | 2010 | | |
| | Coverage | CAM4-Oslo | CAM5.3-Oslo NUDGE_PD (AMIP_PD) | Coverage | AP2 range (≤ 23 models*) | CAM5.3-Oslo NUDGE_PD | Coverage | AP3 range (≤ 8 models*) | CAM5.3-Oslo NUDGE_PD |
|---|---|---|---|---|---|---|---|---|---|
| $SO_2$ conc. | E;N;A | 16 | 150 (137) | E;N | 65 – 977 | 223 | E | NA | 328 |
| $SO_4$ conc. | E;N;A | -5 | 22 (27) | E;N | -61 – 186 | 37 | E | -40 – 199 | 31 |
| BC conc. | E | -54 | -28 (-34) | E | -40 – 64 | -32 | E | -65 – 35 | -16 |
| OA (OM) conc. | E;N | 108 | 122 (125) | E;N | -60 – 335 | 141 | E | -70 –71 | 23 |
| sea-salt conc. | E;N;A | 50 | 22 (40) | E;N | -97 – 477 | 66 | E | -56 – 301 | 36 |
| dust conc. | Global | -14 | -39 (-24) | Global | -64 – 106 | -34 | Global | -82 – 4 | -46 |
| OD550CS | Global | -22 | -16 (-27) | Global | -50 – 133 | -18 | Global | -53 – -3 | -24 |
| OD550 | Global | -8 | 15 (3) | Global | | 11 | Global | | 12 |
| ABS550CS | Global | -32 | -25 (-30) | Global | -80 – 21 | -38 | Global | NA | -36 |
| ABS550 | Global | -33 | -20 (-30) | Global | | -30 | Global | | -35 |
| ANG4487CS | Global | NA | -17 (-15) | Global | -30 – 31 | -15 | Global | NA | -16 |
| ANG4487 | Global | -19 | -44 (-42) | Global | | -44 | Global | | -45 |

* excluding models with missing data or with NMB < -99% or NMB > 1000% (see the main text for more details)

AP2 models: CAM5.1-MAM3-PNNL.A2.CTRL, ECHAM-SALSA.A2.CTRL, ECHAM-SALSA.A2.CTRL.emi2000, GISS-MATRIX.A2.CTRL, GISS-modelE.A2.CTRL, GLOMAPbin1pt1.A2.CTRL, GLOMAPmodev4.A2.CTRL, GLOMAPmodev6R.A2.CTRL, GMI.A2.CTRL, GMI-v3.A2.CTRL, GOCART-v4.A2.CTRL GOCART-v4Ed.A2.CTRL, HADGEM2-ES.A2.CTRL, HADGEM3-A-GLOMAP.A2.CTRL, INCA.A2.CTRL, MPIHAM_V1_KZ.A2.CTRL, MPIHAM_V2_KZ.A2.CTRL, OsloCTM-v2.A2.CTRL, OsloCTM.A2.CTRL, SALSA_v1_TB.A2.CTRL, SPRINTARS-v384.A2.CTRL, SPRINTARS-v385.A2.CTRL, and TM5.V3.A2.CTRL



AP3 models: CNRM-CM6.2Nut127_AP3-CTRL2015, CNMR-CM6.2t127_AP3-CTRL2015, ETHZ-ECHAM-HAM2_CTRL2015, GEOS-Chem-v10-01_AP3-CTRL2015, OsloCTM3_AP3-CTRL2015, SPRINTARS-T106_AP3-CTRL2015, SPRINTARS-T213_AP3-CTRL2015, and TM5_AP3-CTRL2015





**Table 7:** Pearson correlation coefficient (R) statistics for the same data as in Table 9.

| R | Climatology | | | 2006 | | | 2010 | | |
|---|---|---|---|---|---|---|---|---|---|
| | Coverage | CAM4-Oslo | CAM5.3-Oslo NUDGE_PD (AMIP_PD) | Coverage | AP2 range (≤ 23 models*) | CAM5.3-Oslo NUDGE_PD | Coverage | AP3 range (≤ 8 models*) | CAM5.3-Oslo NUDGE_PD |
| SO$_2$ conc. | E;N;A | 0.28 | 0.35 (0.32) | E;N | 0.27 – 0.57 | 0.61 | E | NA | 0.59 |
| SO$_4$ conc. | E;N;A | 0.61 | 0.72 (0.70) | E;N | 0.47 – 0.76 | 0.70 | E | 0.34 – 0.65 | 0.32 |
| BC conc. | E | 0.14 | 0.38 (0.17) | E | 0.15 – 0.57 | 0.42 | E | 0.20 – 0.43 | 0.24 |
| OA (OM) conc. | E;N | 0.31 | 0.29 (0.28) | E;N | 0.15 – 0.39 | 0.25 | E | 0.00 – 0.52 | -0.02 |
| sea-salt conc. | E;N;A | 0.60 | 0.54 (0.51) | E;N | 0.12 – 0.83 | 0.80 | E | 0.34 – 0.72 | 0.71 |
| dust conc. | Global | 0.45 | 0.52 (0.59) | Global | 0.25 – 0.74 | 0.35 | Global | 0.27 – 0.66 | 0.55 |
| OD550CS | Global | 0.67 | 0.59 (0.62) | Global | 0.44 – 0.77 | 0.53 | Global | 0.40 – 0.76 | 0.57 |
| OD550 | Global | 0.67 | 0.64 (0.69) | Global | | 0.61 | Global | | 0.63 |
| ABS550CS | Global | 0.58 | 0.47 (0.50) | Global | 0.45 – 0.70 | 0.45 | Global | 0.60 | 0.40 |
| ABS550 | Global | 0.58 | 0.47 (0.50) | Global | | 0.42 | Global | | 0.47 |
| ANG4487CS | Global | NA | 0.75 (0.74) | Global | 0.56 – 0.72 | 0.68 | Global | NA | 0.69 |
| ANG4487 | Global | 0.47 | 0.46 (0.46) | Global | | 0.45 | Global | | 0.43 |

* excluding models with missing data




**Table 8:** Percentage of model near surface concentration values within a factor 2 of the observations (Fact2, given in %) for the same data as in Table 6.

| Fact2 (%) | Climatology | | | 2006 | | | 2010 | | |
|---|---|---|---|---|---|---|---|---|---|
| | Coverage | CAM4-Oslo | CAM5.3-Oslo NUDGE_PD (AMIP_PD) | Coverage | AP2 range (≤ 23 models*) | CAM5.3-Oslo NUDGE _PD | Coverage | AP3 range (≤ 8 models*) | CAM5.3-Oslo NUDGE _PD |
| SO$_2$ conc. | E;N;A | 36 | 12 (12) | E;N | 4 – 33 | 10 | E | NA | 7 |
| SO$_4$ conc. | E;N;A | 68 | 57 (53) | E;N | 17 – 85 | 45 | E | 14 – 70 | 39 |
| BC conc. | E | 68 | 75 (72) | E | 20 – 51 | 46 | E | 26 – 64 | 50 |
| OA (OM) conc. | E;N | 36 | 40 (40) | E;N | 12 – 53 | 35 | E | 21 – 52 | 39 |
| sea-salt conc. | E;N;A | 34 | 31 (28) | E;N | 0 – 37 | 31 | E | 2 – 40 | 34 |
| dust conc. | Global | 34 | 24 (18) | Global | 9 – 32 | 18 | Global | 7 – 23 | 19 |
| OD550CS | Global | 75 | 42 (41) | Global | 45 – 80 | 39 | Global | 38 – 74 | 49 |
| OD550 | Global | 69 | 68 (71) | Global | | 64 | Global | | 58 |
| ABS550CS | Global | 54 | 47 (51) | Global | 10 – 51 | 45 | Global | NA | 40 |
| ABS550 | Global | 53 | 50 (50) | Global | | 49 | Global | | 48 |
| ANG4487CS | Global | NA | 83 (85) | Global | 68 – 90 | 82 | Global | NA | 83 |
| ANG4487 | Global | 81 | 49 (52) | Global | | 54 | Global | | 51 |

* excluding models with missing data or with NMB < -99% or NMB > 1000% (see the main text for more details)




**Table 9:** Globally and annually averaged PD mass extinction coefficients at 550 nm for each of the main aerosol components in CAM5.3-Oslo, compared to CAM4-Oslo and to AeroCom Phase I models. For a component X we calculate $MEC_X = AOD_X/B_X$, where $B_X$ is the burden of the component.

| MEC ($m^2$ $g^{-1}$) decomposition | CAM4-Oslo | CAM5.3-Oslo NUDGE_PD (AMIP_PD) | AeroCom Phase I median (min–max) |
|---|---|---|---|
| Sulfate ($SO_4$) | 6.7 | 5.84 (5.78) | 8.5 (4.2–28.3) |
| OM | 8.6 | 5.99 (6.06) | 5.7 (3.2–11.4) |
| BC | 6.5 | 7.56 (7.64) | 8.9 (5.3–18.9) |
| Dust | 1.4 | 1.64 (1.66) | 0.95 (0.46–2.1) |
| Sea-salt | 3.1 | 5.04 (5.05) | 3.0 (0.88–7.5) |
| Reference | Kirkevåg et al. (2013) | This work | Kinne et al. (2006) |



**Table 10:** Globally and annually averaged aerosol Radiative Forcing (RF) and Effective Radiative Forcing (ERF) decomposed into its SW and LW components for CAM5.3-Oslo and CAM4-Oslo, compared with the respective mean values and ranges reported in IPCC AR5. Note that the estimates from IPCC AR5 are only available as sums of the SW and LW contributions and have been estimated for the period 1750 to 2011 (with one exception, see the foot-note), whereas the CAM4-Oslo and CAM5.3-Oslo estimates are for year 1850 to 2000.

| RF / ERF decomposition | CAM4-Oslo RF (W m⁻²) | CAM5.3-Oslo ERF (W m⁻²) NUDGE_PD (AMIP_PD) | IPCC AR5 RF (W m⁻²) | IPCC AR5 ERF (W m⁻²) |
|---|---|---|---|---|
| SW ARI | -0.10 | -0.095 (-0.092)* | -0.35 (-0.85 to 0.15) | -0.45 (-0.95 to 0.05) |
| LW ARI | - | 0.026 (0.026)* | | |
| SW ACI | -0.91 | -1.50 (-1.45)* | Not assessed | -0.45 (-1.20 to 0.0) |
| LW ACI | 0.01 | 0.161 (0.155)* | | |
| ARI & ACI | -1.00 | -1.41 (-1.36)* | Not assessed | -0.9 (-1.9 to -0.1) -1.08 (-1.40 to -0.76)** |
| Reference | Kirkevåg et al. (2013) | This work | Boucher et al. (2013) | Boucher et al. (2013) |

5    * the semi-direct effect is here embedded in the ERF ACI term (Ghan, 2013), not in ERF ARI as in the IPCC AR5 estimates

** mean ± one standard deviation for CMIP5 and ACCMIP models for the period 1850 – 2000





**Table 11:** All-sky and clear-sky aerosol optical depth (OD) and absorptive optical depth (ABS) at 550 nm, Liquid Water Path (LWP), in-cloud cloud droplet number concentrations (CDNC=AWNC/FREQL from model output) and averaged effective cloud droplet radius (Reffl=AREL/FREQL) at 860 hPa (model layer 24), and Ice Water Path (IWP). Also shown are the column integrated CDNC (CDNCcol=CDNUMC) and Ice Crystal Number Concentration values (ICNCcol, calculated as part of the post-processing).

| Experi-ment | OD550 (OD550CS) | ABS550 (ABS550 CS) | LWP (g m⁻²) | CDNCcol (1.e6 cm⁻²) | CDNC 860 hPa (cm⁻³) | Reffl 860 hPa (μm) | IWP (g m⁻²) | ICNCcol (cm⁻²) |
|---|---|---|---|---|---|---|---|---|
| NUDGE PD | 0.152 (0.1244) | 0.0048 (0.0049) | 53.85 | 1.39 | 58.93 | 11.25 | 10.00 | 6874.16 |
| NUDGE PI | 0.128 (0.109) | 0.0036 (0.0037) | 50.29 | 1,10 | 49.12 | 11.56 | 10.03 | 6876.01 |
| NUDGE PD-PI | 0.025 (0.015) | 0.0012 (0.0012) | 3.56 | 0.29 | 9.81 | -0.31 | -0.03 | -1.9 |
| AMIP PD | 0.142 (0.113) | 0.0042 (0.0044) | 53.52 | 1.37 | 57.57 | 11.50 | 10.25 | 6882.92 |
| AMIP PI | 0.119 (0.098) | 0.0031 (0.0032) | 50.10 | 1.08 | 47.78 | 11.86 | 10.29 | 6882.97 |
| AMIP PD-PI | 0.023 (0.014) | 0.0011 (0.0012) | 3.42 | 0.29 | 9.79 | -0.36 | -0.04 | -0.05 |



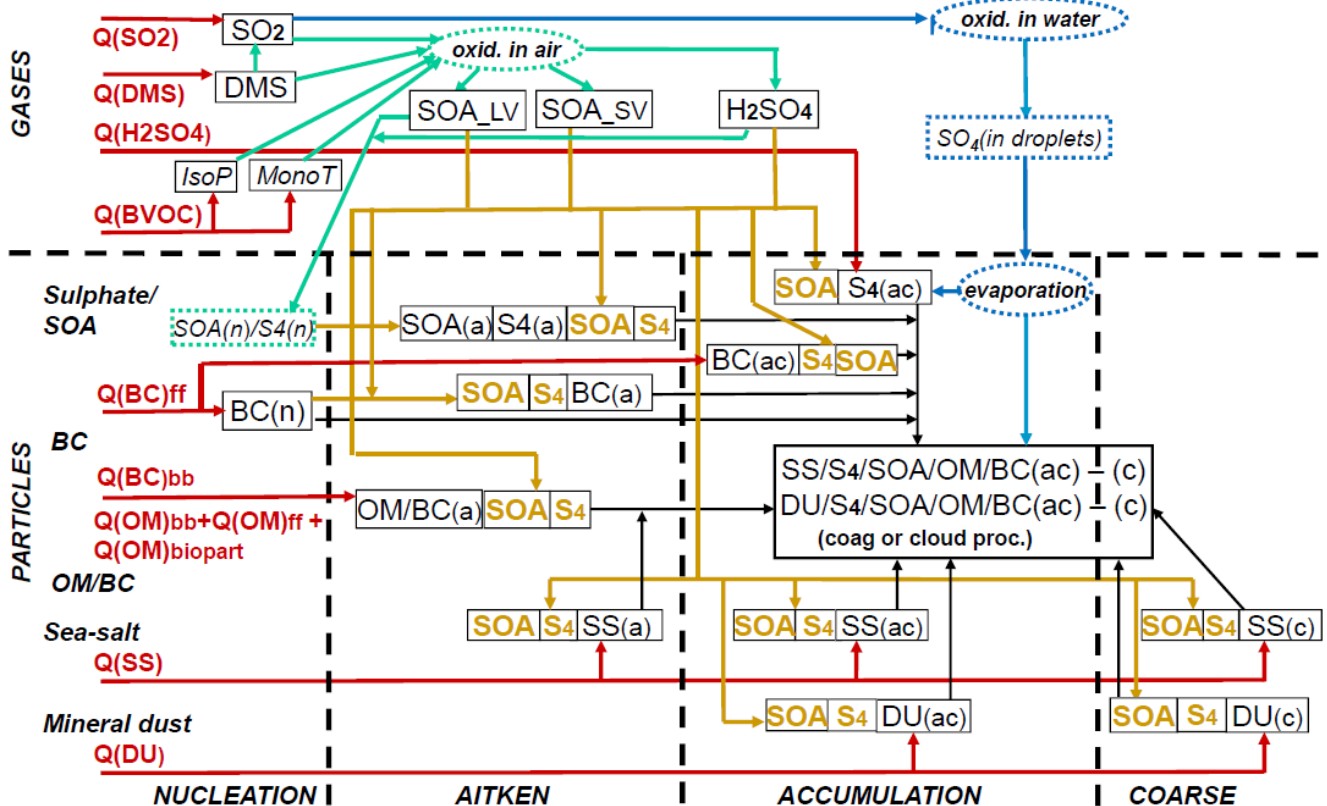

**Figure 1:** Flow diagram of processes in the aerosol module OsloAero5.3. The source terms to the left, labeled Q(X)y, where X is the constituent name and y is the source type, can be primary emissions or secondary production. The source labels y = *bb*, *ff*, or *biopart* indicate biomass burning, fossil or biofuel combustion, and biogenic particle sources. Primary particles are emitted (red arrows) as accumulation mode sulfate (S4(ac)), nucleation and accumulation mode black carbon (BC(n), BC(ac)), Aitken mode BC (BC(a)), internally mixed Aitken mode organic matter and black carbon (OM/BC(a)), Aitken, accumulation, and coarse mode sea-salt (SS(a), SS(ac), SS(c)), and accumulation and coarse mode mineral dust (DU(ac), DU(c)). Model calculated gas-phase components are DMS, SO2, Isoprene (IsoP), Monoterpene (MonoT), H2SO4, and gaseous secondary organics (SOAG$_{LV}$ and SOAG$_{SV}$). SOAG$_{LV}$ partly co-nucleates with nucleation-mode sulfate (S4(n), SOA(n), turquoise arrows), and partly condensates (yellow arrows) on existing particle surfaces, while SOAG$_{SV}$ only forms SOA through condensation. Sulfate produced in cloud water droplets (SO4(in droplets), blue arrow) is partly added to S4(ac), and partly to a broad internal mixture of accumulation and coarse mode particles coagulated with either mineral dust or sea-salt. Black arrows represent coagulation which contributes to the latter two particle types. Components in dashed boxes are not explicitly calculated.



**Figure 2:** Hygroscopic growth factors (wet/ambient radius divided by dry radius) for aerosol components at some typical dry radii and for relative humidities up to $RH_{max} = 99.5\%$, as treated in AeroTab5.3 and the optics look-up tables. Note that the growth factor curve for sea-salt at dry radius 0.3 µm is not visible due to overlap with that for 0.75 µm. To relate this figure to the nomenclature in Table 2, $H_2SO_4$ (sulphuric acid) may come as SO4_NA, SO4_PR or SO4_A1, $(NH_4)_2SO_4$ (ammonium sulfate) as SO4_A2, BC as BC_AX, BC_N, BC_NI or BC_A, OM as OM_NI, OM_AI, SOA_NI or SOA_A1, mineral dust as DST_A2 or DST_A3, and sea-salt as SS_A1, SS_A2 or SS_A3.





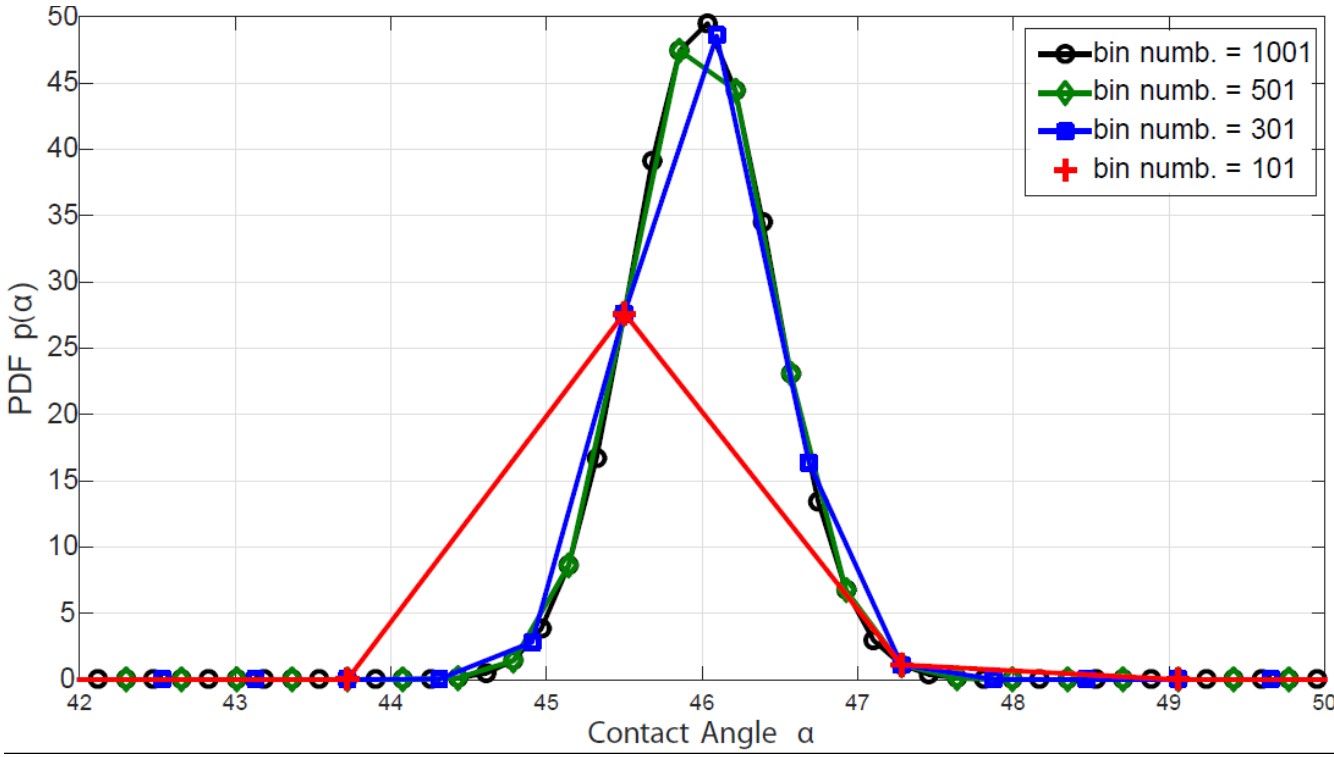

**Figure 3:** Probability p($\alpha$) of the $\alpha$-PDF model used for calculating the contact angle for immersion freezing. Different bin numbers are tested in order to correct the numerical formulation that is used in Wang et al. (2014).





**Figure 4:** Modeled zonal mean mass mixing ratios of a) SO₂ b) Sulfate (as S), c) BC, d) OM, e) Dust, and f) Sea-salt, in the NUDGE_PD (left panels) and the AMIP_PD (right panels) simulation. Note the different scales for mineral dust and sea-salt vs. the other components.





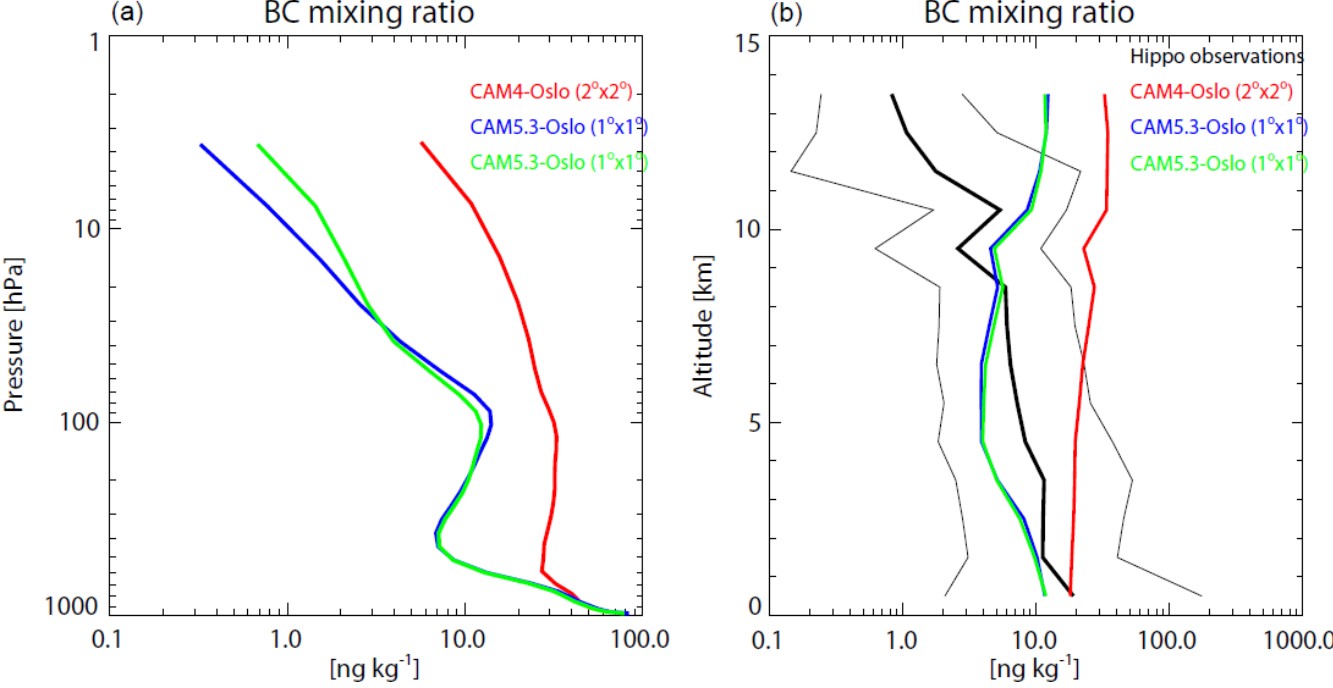

**Figure 5:** (a) Globally averaged annual BC mass mixing ration profiles as modeled in the NUDGE_PD (blue line) and AMIP_PD (green line) experiments, and in CAM4-Oslo for comparison. (b) Modeled BC mass mixing ratio profiles from the same simulations as in (a), compared to HIPPO aircraft campaigns averaged over the areas and months where the campaign took place (Schwarz et al., 2013; see also Samset et al., 2014).



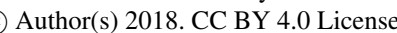


**Figure 6:** Surface concentrations in the NUDGE_PD experiment compared with EBAS and AEROCE data through the AeroCom tools.

OA represents modeled OM concentrations vs. observed OC concentrations multiplied by 1.4 (the assumed OM/OC ratio for fossil fuel OC in the model).





**Figure 7:** Scatter plots (left panels) and annual relative bias plots with respect to AERONET observations/retrievals (right panels) of clear-sky aerosol optical depth (top), all-sky absorption optical depth (middle) at 550 nm, and Angström parameter (bottom) for the wavelength range 440-870 nm, for the NUDGE_PD simulation.





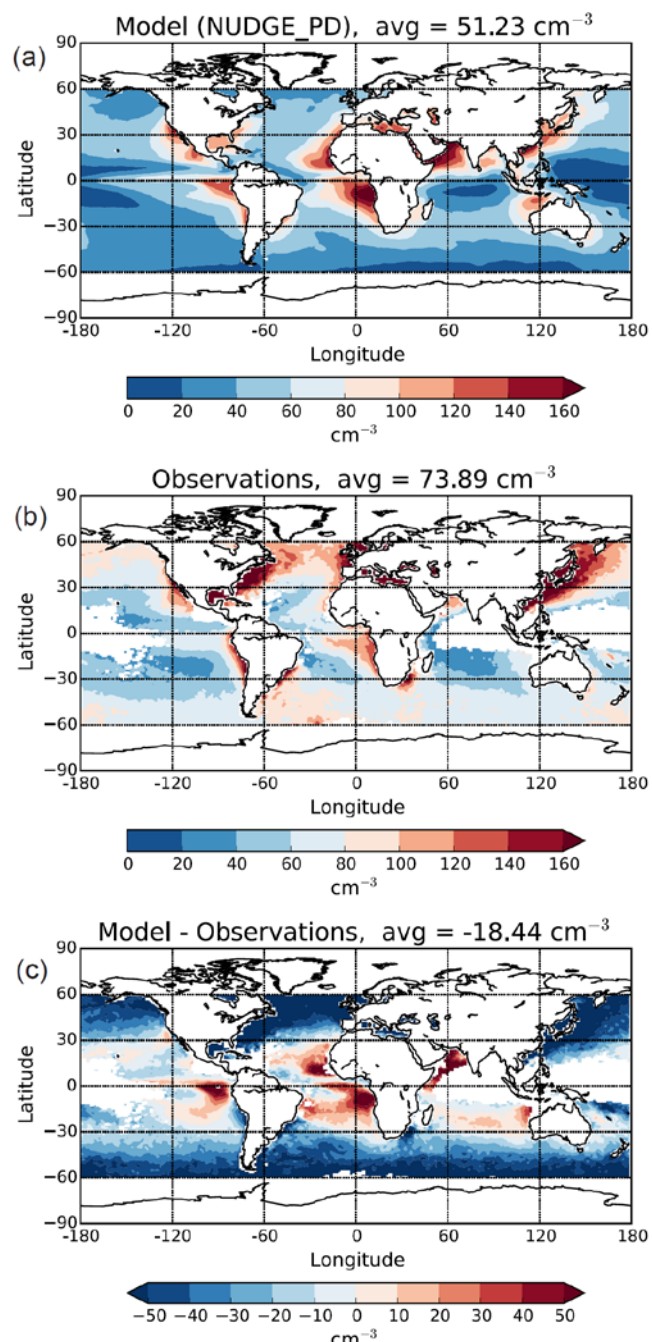

**Figure 8:** In-cloud cloud droplet number concentrations at cloud top in (a) CAM5.3-Oslo (NUDGE_PD) compared to (b) Bennartz et al. (2017), with the difference shown in (c). White areas indicate lack of observations from MODIS meeting the criteria on temperature and cloud fraction given by Bennartz et al. (2017).





**Figure 9:** Shortwave (SW, top panels) and longwave (LW, bottom panels) direct radiative forcing at top of the atmosphere (TOA) for the simulations NUDGE_PD – NUDGE_PI (left panels) and AMIP_PD – AMIP_PI (right panels). Note the different color scales.





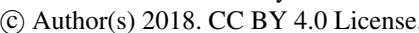

**Figure 10:** Shortwave (SW, top panels) and longwave (LW, bottom panels) anthropogenic cloud radiative forcing at top of the atmosphere (TOA) for the simulations NUDGE_PD – NUDGE_PI (left panels) and AMIP_PD – AMIP_PI (right panels).