# Peer review of "A production-tagged aerosol module for earth system models, OsloAero5.3 – extensions and updates for CAM5.3-Oslo"

_Geoscientific Model Development, 2018_

## Short Comment (SC1) · 22 Jun 2018

As outlined in https://www.geoscientific-model-development.net/about/manuscript_ types.html GMD is encouraging authors to provide access to the release of the source code matching the model described in manuscript preferable through a DOI. The information provided are incomplete. When copyright or licensing restrictions prevent the public release of model code, or in the cases where there is some other good reason for not allowing public access to the code, authors need to state the reasons for why access is restricted and need to explain how access can be obtained (e.g. signing a license agree or join a consortium).

Lutz Gross GMD Executive Editor

---

## Referee Comment (RC1) · Anonymous Referee #1 · 5 Jul 2018

The article untitled "A production-tagged aerosol module for earth system models, OsloAero5.3 – extensions and updates for CAM5.3-Oslo" by A. Kirkevag et al. presents in a very detailed way updates in the modelisation of aerosols that is used in the atmospheric component of the Norwegian Earth System Model (NorESM). This updated version called OsloAero5.3 is here tested in the CAMS5.3 atmospheric model which is part of the Community Earth System Model 1.2 (CESM). With regards to the CMIP6 project, OsloAero5.3 is planned to be integrated/merged with CEMS2 to form the NorESM2 model, but the version presented in this article could be used for the early phase of CMIP6. Therefore, in addition to being of value to the aerosol modelling community, the discussions in the article are fully relevant to the CMIP6 exercise.

[Figure]

The article is very well written, and provides a thorough review of changes from a previous version documented in Kirkevag et al. 2013, together with an analysis of several aerosol diagnostics. Several changes have been mode, including ones to the aerosol sources, aerosol nucleation, soa production, and aerosol-cloud interactions. The analysis presents comparisons not only with results from the previous version of the aerosol model, but also with observations, with results from other aerosols models, in particular those of the AeroCom community, and with other aerosol studies. The analysis attempts to document in details the advances and setbacks of this new version. Although the article is very long, in particular in the aerosol model description part, I would recommend it for publication in GMD as it is, as details in any part can be of interest to some scientists in the aerosol community. I include below a few comments that will require only minor corrections in the document.

- it would be interesting to include information of the added computational cost required to use this aerosol model, possibly in comparison to the other aerosol models of CESM (MAM3 and MAM7). In particular, which/now many of the tracers listed in Table 1 are transported by the model? Also, please add some details on the chemical mechanism used in conjunction to this aerosol model (line 6 page 8)

- page 2 line 2: "while it...": please clarify this part of the sentence

- page 2 line 4: "overestimated" shouldn't it be changed to "underestimated"

- page 13 line 19: is the proportional coefficient you use that of the references?

- page 15 line 8: please provide details on the vertical resolution (extension, distribution of levels)

- page 17 line 26: please comment on the fact that in Figure 5 the Nudge and Amip lines are quasi identical in the troposphere

**[GMDD](GMDD)**

Interactive
comment

- page 19 line 28: comment on the interest of self nudging

- page 21 line 3 change "better than" to "better"

- page 29 line 7: please clarify "as the source of the largest negative biases" where do you see that in Figure 7?

- page 30 line 1: please add a few details about this dataset

- page 34 line 13: is this excess of BC in the upper troposphere common/uncommon in other models?

- page 56: Table 6 legend: I suggest to change "monthly European and Global data" to "monthly data". Please explain "(or world oceans in the case of dust)".

  Please indicate for how many models you have clear-sky information.

- page 58 in Table 7 legend, change "Table 9" into "Table 6"

- page 59 : please add "and column integrated optical properties" in the legend

- page 62: indicate meaning of AWNC, FREQL, AREL. CDNUMC non needed in legend.

- page 66: meaning of etax1000? please add a couple of levels in the stratosphere

- page 71: I believe ERF ari are shown; it would be clearer to indicate this on top of the figures and in the legend

- finally, in general zooming in the figures degrades their quality; this needs to be corrected

---

## Referee Comment (RC2) · Anonymous Referee #2 · 16 Jul 2018

This manuscript describes the latest version of the OsloAero module used for modelling atmospheric aerosol in the aerosol-climate model CAM5.3-Oslo and earth-system model NorESM 1.2. OsloAero has a quite different formulation to most aerosol schemes, using a set of "background aerosol" tracers, whose prescribed size distributions and composition are then modified according to a second set of tracers which are tagged according to aerosol production processes. The resulting variations in size and composition, which are not restricted to e.g. a log-normal form, are then looked up in tables computed offline to determine the relevant optical and other properties of these mixtures. This is a novel approach to aerosol modelling which the authors have developed over a number of previous papers, adding distinct capabilities compared to the more widespread modal and sectional schemes. The present manuscript describes new developments for nucleation and secondary organic aerosol, as well as new online emission schemes for sea-salt, mineral dust and oceanic DMS and organics. There has also been an effort to ensure better consistency of parameters between the different components of the scheme, which is always welcome from a physical point of view. These are important additions, both in terms of updating this modelling framework with the latest understanding of physical processes, and for tighter coupling in the earth system context. A good basic set of evaluation plots and metrics are included, though the overall results compared to the previous CAM4-Oslo are somewhat mixed, with some biases and errors improved but others becoming larger; nevertheless this is a well-presented paper documenting a significant advance in the physical and chemical capabilities of the model and I would recommend it for publication in GMD subject to the minor comments below:

**p.3, line 28 – p.4, line 10** This section is quite programmatic in terms of discussing the evolution of the projects to which OsloAero is related; I would consider whether all of this is relevant to a model description paper.

**p.4, line 27** "same method of aerosol activation as Liu et al. (2012)": is this the primary reference for the parameterisation in use, or does it describe a particular implementation of a well-known parameterisation (e.g. Abdul-Razzak and Ghan, 2000) which should also be cited here.

**p.6, line 4** It's not clear either *why* a portion of the modifying tracer is redistributed to the background ones, or the basis on which the amount redistributed is calculated; some explanation here is needed.

**p.6, lines 12–13** In what way are these sized "augmented to take into account atmospheric growth"? This is a rather vague description.

**p.6, lines 13–16** A little more explanation of what is meant by lumping the size-modifying tracers would be good, as well as the meaning of the modal size parameters of these modifying tracers, if these are not particles in their own right, but act to adjust the sizes of the background modes. (This may be explained further in earlier papers e.g. that labelled K13, but should ideally be self-explanatory in this manuscript.)

**p.27–29** I assume "lost from the model" means that monoterpene and isoprene do not exist as tracers in the model, but only as "transient" species near the surface within a given model time step?

**p.11, line 19** This should be "…scheme is different from that…" or "…schemes are different from those…" for singular/plural consistency.

**p.14, line 8** It would be good to recap the emission sources here, even if they haven't changed from the previous paper.

**p.15, line 8** What is the height of the model top with 30 levels?

**p.15, line 9** Clarify that "microphysical schemes" here refers to cloud and precipitation rather than aerosol, if that is the case. Also, a brief description of the nature of these schemes (single/double-moment, what is prognostic etc.) would be welcome to provide context for aerosol-cloud interactions in this model.

**p.15, line 24** A reference for the nudging technique would be welcome (e.g. Jeuken et al., 1996, 10.1029/96JD01218 or equivalents in other models).

**p.16, line 18** Is there a reason why CAM4-Oslo couldn't be tested at the higher resolution, or CAM5.3-Oslo at the lower one, in order to assess the impact of the resolution change separately from the actual model updates?
**p.16, line 25** If the DMS burden is doubled compared to the earlier version, are there some observations which could be cited here to indicate which is more realistic, or is the uncertainty even larger than this?

**p.17, line 29–30** The attribution of upper-tropospheric excess to the treatment of convective processes based on comparisons to HIPPO was also made by Kipling et al. (2013, 10.5194/acp-13-59690-2013 in the context of another model.

**p.27, lines 8–12** The rationale for using the grid-box-mean RH and weighting by clear-sky fraction, rather than the more common approach is not clear.

**p.37, line 30** NUDGE_PI appears twice; one should be NUDGE_PD.

**p.38, line 9** Who should be contacted to obtain a user agreement for access to the code and data?

**Tables** Many of the tables contain a large number of numerical values, either mass budget terms or statistics. While the actual numbers may be useful for reference, summarising these in charts would probably be much easier for the reader to understand at a glance.

---

## Author Comment (AC1) · 24 Aug 2018

**Author's response to interactive comments by Exexutive Editor of GMD, Lutz Gross:**

The authors would like to thank the Executive Editor for the very constructive review comment, which is repeated below in bold font, followed by our reply.

**As outlined in https://www.geoscientific-model-development.net/about/manuscript_types.html GMD is encouraging authors to provide access to the release of the source code matching the model described in manuscript preferable through a DOI. The information provided are incomplete. When copyright or licensing restrictions prevent the public release of model code, or in the cases where there is some other good reason for not allowing public access to the code, authors need to state the reasons for why access is restricted and need to explain how access can be obtained (e.g. signing a license agree or join a consortium).**

This manuscript is a documentation of the aerosol and atmosphere module of NorESM1.2, CAM5.3-Oslo, but the rest of the ESM has not yet been documented and published. The code for NorESM1.2 is all on the same github repositoy, and CAM5.3-Oslo cannot easily be put on a separate repository. This would imply a huge additional effort by many scientists, which we are unable to defend and afford at the moment.

Making the code available to everyone through a doi number (or alternative solutions) is something that will be discussed in a new infrastructure project funded by the Norwegian Research Council (INES, lead by Mats Bentsen and Michael Schulz). Since NorESM1.2 is partly based on CESM1.2 from NCAR in USA, the code availablility indirectly relies on the user agreement between us as NorESM developers and NCAR. Potentially new users are also asked to become CESM users, which is necessary for obtaining access to new model input data (as well as code) from NCAR, most of which are also used in NorESM.

Based on the above, we suggest to rewrite the first part of the Code and data availability section on lines 8 and 9 as follows:

«The source code for CAM5.3-Oslo is part of a restricted NorESM2 pre-release and stored within the private github NorESM repository (https://github.com/metno/noresm/tree/NorESM1.2-v1.0.0). Access to the code and simulation output data produced in this study can be obtained upon reasonable request to noresm-ncc@met.no and requires entering a NorESM Climate modeling Consortium (NCC) user agreement.»

---

## Author Comment (AC2) · 24 Aug 2018

**The article untitled "A production-tagged aerosol module for earth system models, OsloAero5.3 – extensions and updates for CAM5.3-Oslo" by A. Kirkevag et al. presents in a very detailed way updates in the modelisation of aerosols that is used in the atmospheric component of the Norwegian Earth System Model (NorESM). This updated version called OsloAero5.3 is here tested in the CAMS5.3 atmospheric model which is part of the Community Earth System Model 1.2 (CESM). With regards to the CMIP6 project, OsloAero5.3 is planned to be integrated/merged with CEMS2 to form the NorESM2 model, but the version presented in this article could be used for the early phase of CMIP6. Therefore, in addition to being of value to the aerosol modelling community, the discussions in the article are fully relevant to the CMIP6 exercise.**

**The article is very well written, and provides a thorough review of changes from a previous version documented in Kirkevag et al. 2013, together with an analysis of several aerosol diagnostics. Several changes have been mode, including ones to the aerosol sources, aerosol nucleation, soa production, and aerosol-cloud interactions. The analysis presents comparisons not only with results from the previous version of the aerosol model, but also with observations, with results from other aerosols models, in particular those of the AeroCom community, and with other aerosol studies. The analysis attempts to document in details the advances and setbacks of this new version. Although the article is very long, in particular in the aerosol model description part, I would recommend it for publication in GMD as it is, as details in any part can be of interest to some scientists in the aerosol community. I include below a few comments that will require only minor corrections in the document.**

Thank you for these encouraging words. Having a good and well documented basis paper for the pre-CMIP6 model version and for present and planned AeroCom experiments are indeed among the main purposes of the manuscript, while at the same time hoping it can be of use for researchers within the field in general.

**• it would be interesting to include information of the added computational cost required to use this aerosol model, possibly in comparison to the other aerosol models of CESM (MAM3 and MAM7). In particular, which/now many of the tracers listed in Table 1 are transported by the model? Also, please add some details on the chemical mechanism used in conjunction to this aerosol model (line 6 page 8)**

All the aerosol tracers in Table 1 are transported by the model, as mentioned on p. 5, line 25.  To clarify this also in Table 1, we change the text in the caption to

«Transported aerosol tracers in CAM5.3-Oslo».

For completeness, we suggest to add also the list of gas tracers transported in the model, in a sentence at the end of the Table 1 caption:

«The aerosol precursor and oxidant gas tracers transported by the model are: $SO_2$, $H_2SO_4$, DMS, isoprene, monoterpene, SOAG_LV, SOAG_SV, and $H_2O_2$.»

To answer the question about model cost, we suggest to add a new paragraph at the end of Sect. 2.1 (p. 6):

«The total number of transported aerosol and gas tracers in OsloAero5.3 is 29 (21 aerosol and 8 gas tracers, see Table 1), compared to 20 (15 and 5) in MAM3 and 37 (31 and 6) in MAM7. Comparing CAM5.3-Oslo simulations using OsloAero5.3 with MAM3, we find a ca. 49% increase in model cost (50% for the atmosphere module alone). Much of the relatively large increase in model cost compared to MAM3 is due to the multi-dimensional table look-ups and interpolation calculations for aerosol optical properties and sizes in OsloAero5.3. For comparison, according to Liu et al. (2012), CAM5.1 set up with MAM7 runs about 30% slower than with MAM3.»

Although described to some degree in the first paragraph of Sect. 2, we agree that more detailed information about the chemical mechanisms should be provided, and suggest to add the following explanatory text after line 6 on page 8:

«The chemical mechanisms in OsloAero5.3, for sulfur and oxidant chemistry as well as the SOA chemistry in (R1) – (R6), have been described in more detail by Karset et al. (2018), Sect 2. For an overview of the chemical reactions and the respective reaction rate coefficients, see Table 2 in Karset et al. (2018).»

• **page 2 line 2: "while it...": please clarify this part of the sentence**

We here try to convey with just a few words that some regional biases are of opposite sign to the globally averaged bias, and we hope this is clarified by the following reformulation:

«Comparing clear-sky column integrated optical properties with data from ground based remote sensing, we find a negative bias in optical depth globally, however not as strong as in CAM4-Oslo, but with positive biasess in some areas typically dominated by mineral dust emissions.»

• **page 2 line 4: "overestimated" shouldn't it be changed to "underestimated"**

We here compare the biases in clear-sky absorption with those of optical depth (all in CAM5.3-Oslo), which are both overestimated in some of the most dust dominated areas, although they are underestimated globally. These regional overestimates are smaller (or even of opposite sign) for absorption than for optical depths, as described in more detail on p. 28, lines 11-16. So we do mean «less overestimated» for these particular regions. In case the meaning of this senstence is unclear to the referee because of how it is formulated, we suggest to reformulate it slightly, to:

«Aerosol absorption has a larger negative bias than the optical depth globally. This is reflected in a lower positive bias in areas where mineral dust is the main contributor to absorption.»

• **page 13 line 19: is the proportional coefficient you use that of the references?**

As mentioned in l. 11, C is just a (non-adjustable) unit conversion coefficient, not given in any of the references. Since k_600 in the model is given with units cm/h, M_DMS as g/mol, and C_DMS as nmol/L, C = 2.778E-15 in the model code, which gives F_DMS with units kg/m2/s in Eq. 4.  To clarify, we will change the text on l. 11 from «C is a unit conversion coefficient» to

«C is a unit conversion coefficient in the model code (not a tuning factor)».

• **page 15 line 8: please provide details on the vertical resolution (extension, distribution of levels)**

We may here add the following information

«In hybrid sigma pressure coordinates, the uppermost eta level mid (or top of the level) value is 3.64 (2.26), and for the lowermost level it is 992.56 (985.11). The number of layers below approximately 1 km and 2 km height a.s.l. are 5 and 8, respectively.»

**• page 17 line 26: please comment on the fact that in Figure 5 the Nudge and Amip lines are quasi identical in the troposphere**

We propose to add at the end of this paragraph:

«Note that NUDGE_PD and AMIP_PD yield almost identical results in the troposphere. This indicates that the nudging, as long as we are not nudging the temperature, only has modest effects on the convective transport and mixing of BC in the model (see also Fig. 4).»

**• page 19 line 28: comment on the interest of self nudging**

We are not sure what the referee would like to see added to the text here, but we tentatively add at the end of the paragraph, on line 3, p. 20:

«An important effect of nudging is that it constrains the model's natural variability (Kooperman et al., 2012), which is useful in calculations of the indirect effect of aerosols since it reduces the simulation length required to obtain sufficiently high signal to noise ratios. When nudging to a meteorology produced by the model itself (self-nudging) instead of using data from reanalysis (such as the ERA data), one additionally obtains model meteorology and climate which are more consistent with the model's own, innate behaviour, which in turn lends more confidence in, e.g., estimates of effective radiative forcing, but is less useful for the purpose of comparing modeled aerosol properties with observations.»

We furthermore suggest to add at p. 33, line7:

«Note, however, that nudging to the ERA data instead of the model's own meteorology only has small impacts on anthropogenic cloud forcing: Karset et al. (2018) applying self-nudging (and when using the same oxidant levels as in the present study) in CAM5.3-Oslo, estimated it to -1.32 W $m^{-2}$, very close to our estimate -1.34 W $m^{-2}$ (SW + LW ERF ACI in Table 10).»

**• page 21 line 3 change "better than" to "better"**

The sentence spanning lines 2-4 is awkwardly phrased, but «better than» is correct to use here, see Table 7. In order to make it clearer, we suggest to rewrite this sentence to:

«For the climatological averaged data, the Pearson correlation coefficient R (hereafter often just referred to as the correlation) is slightly better for the nudged than for the un-nudged AMIP simulation, where instead the normalized mean bias NMB (hereafter often just referred to as the bias) is slightly better.»

**• page 29 line 7: please clarify "as the source of the largest negative biases" where do you see that in Figure 7?**

Thank you for pointing this out: this part of the text is not very well formulated. We suggest to change this and the preceeding sentence to:

«Regionally, ANG4487CS is most underestimated in Northern Africa (defined as above, i.e. extended to include sites along the European coast of the Mediterranean, -35%), followed by Europe (defined as above, -32%), Australia (-31%), India (-20%), East Asia (-10%), and smallest in North America (-7%).

This pattern (see also Fig. 7) seems to point towards dust as a source of large negative biases, which is consistent with an excessive mineral dust contribution to the total aerosol (as also indicated by the regional OD550CS biases), or alternatively, overestimated dust particles sizes (opposite of what we found as a potential cause of the positive bias in ABS550CS).»

**• page 30 line 1: please add a few details about this dataset**

We here propose to add

«This data set is a climatology of cloud droplet number concentration (monthly mean, in-cloud 1°x1° CDNC values plus associated uncertainties for warm clouds) based on 13 years of Aqua-MODIS observations over the global ice-free oceans.».

**• page 34 line 13: is this excess of BC in the upper troposphere common/uncommon in other models?**

Although CAM4-Oslo was among the more extreme o the AeroCom models in that respect, it was not unique: e.g., HadGEM2 is more extreme in many regions, as can be seen from model – HIPPO intercomparison in Samset et al. (2014, see Fig. 2) (doi:10.5194/acp-14-12465-2014). We may therefore add at the end of this paragraph (on line 18):

«Note that there is a general tendency for models participatig in the AeroCom project to overestimate the aircraft measurements aloft in remote regions (Samset et al., 2014).».

We also suggest to add a reference to the AeroCom intercomparison paper by Kipling et al. (2016) in the sentence on lines 14-16:

«This may be related to aerosol aging and to how aerosols are transported and scavenged in deep convective clouds (see e.g. Kipling et al., 2016): also the mass concentrations of the other aerosol components have been reduced (aloft) from CAM4-Oslo to CAM5.3-Oslo.»

**• page 56: Table 6 legend: I suggest to change "monthly European and Global data" to "monthly data". Please explain "(or world oceans in the case of dust)". Please indicate for how many models you have clear-sky information.**

The suggested text change from "monthly European and Global data" to "monthly data" looks fine, and will be done in the updated manuscript.

The term «world oceans in the case of dust» is clumsy, but makes sense when looking at the actual map of available stations at aerosom.met.no, where we see that most of the observation sites are located on islands and along the coasts of the continent, see http://aerocom.met.no/cgi-bin/aerocom/surfobs_annualrs.pl?PROJECT=NorESM&MODELLIST=NorESM-Ref2017&FULL=explicit&INFO=nohover&PERFORMANCE=ind&YEARFILTER=ALLYEARS&PSFILTER=ALLVARS&Type0=SITELOCATION&Ref0=AEROCE&Run0=CAM53-Oslo_7310_MG15CLM45_5feb2017IHK_53OSLO_PD_UNTUNED&Parameter0=SCONC_DUST&Station0=WORLD&Year0=an9999&Period0=mALLYEAR. We suggest to reformulate «and Global = nearly all continents (or world oceans for the case of dust) are represented.» to:

«and Global = nearly all continents or world oceans (island sites) are represented.»

We do not know the exact number of AeroCom models (among those we compare results with) which have clear-sky information on aerocom.met.no. This question was taken up by the AeroCom community during phase II of the project, which resulted in an attempt of documentation via a wiki page: https://wiki.met.no/aerocom/optical_properties. Among the 10 AeroCom model versions (in

addition to our own) which are both appearing in the wiki and in Tables 6-8 in the present study (GISS-MATRIX, GISS-modelE, GOCART-v4Ed, HADGEM2-ES, INCA, MPIHAM_V2_KZ, OsloCTM-v2, SPRINTARS-v384, and TM5.V3), only one estimates clear-sky in a similar way as we do, i.e. by using weights based on total 2D cloud cover but all-sky RH for hygroscopic growth of the aerosol: SPRINTARS_v384 averages only AOD for time steps and grid points with CLDTOT<0.2, using all-sky RH in these almost cloudfree cases. GISS-MATRIX, GMI and GOCART_v4E average all-sky AOD (no cloud cover weighting), also based on growth factors from all-sky RH. GISS-MATRIX also provide clear-sky optics, but this has not been used at aerocom.met.no. Although not very clear from the wiki, the rest of the models seem to estimate AOD (tacitly meant to be a clear-sky parameter) for all cloud conditions, but based on hygroscopic particle growth using a clear-sky RH value, defined as the RH in the clear-sky fraction (the value which gives a grid-mean all-sky RH when the cloudy part of the grid is assumed to be saturated). Furthermore, all but one of the 10 models calculate AOD as averages over both night and daytime values, as in CAM4-Oslo and CAM5.3-Oslo.

As described, the CAM4-Oslo and CAM5.3-Oslo data on aerocom.met.no contain both clear-sky and all-sky aerosol optics (for some selected wavelengths) by using the 2D clear-sky fraction weighted AODs as clear-sky AOD. In these models the two measures do differ significantly. However, in an early version of CAM6-Oslo, where we have changed to using clear-sky RH hygroscopic growth for the optics calculations, and otherwise introduced only minor changes in aerosol treatment, the relative difference between all-sky and clear-sky weighted global optical depth has been reduced from ~20% (in CAM5.3-Oslo) to ~10% or less. Although the AeroCom wiki does not provide any quantative answers to how much the clear-sky and all-sky optical propertes differ for other AeroCom models, the models which assume hygroscopic growth based on RH in the clear-sky fraction of the model grid-cell, instead for the all-sky RH as in our model, may therefore also have relatively small differences between all-sky and clear-sky weighted optical properties.

This was a long answer to a short question, for which we unfortunately cannot give an exact reply. We suggest a very short summary of the above to be added in the manuscript, by changing a part of the caption of Table 6 (lines 8-11) from «Optics diagnostics listed for the AP2 and AP3 models are mostly clear-sky values, in the sense that the clear-sky humidity of the grid cell is used for calculating the hygroscopic swelling. CAM4-Oslo and CAM5.3-Oslo compute all-sky optical properties using the average humidity of the grid cell. Clear-sky (CS) properties are represented by a cloud fraction weighted average of the all-sky properties.» to

«Optics diagnostics listed for most of the AP2 and AP3 models (exact number is not available) are clear-sky values, in the sense that the clear-sky humidity of the grid cell is used for calculating hygroscopic swelling of the aerosol. Information about this for 11 of the AP2 models included here, plus some others, may be found at https://wiki.met.no/aerocom/optical_properties. CAM4-Oslo and CAM5.3-Oslo compute all-sky optical properties using the average humidity (RH) of the grid cell. Clear-sky (CS) properties are instead represented by a 2D cloud-free fraction weighted average of the all-sky properties. Only a few other AeroCom models follow a similar clear-sky optics definition, and the optics data submitted to AeroCom for a few of the models are all-sky values both in terms of cloud conditions and RH for hygroscopic growth.»

**• page 58 in Table 7 legend, change "Table 9" into "Table 6"**

Thank you, this will be corrected as suggested.

**• page 59 : please add "and column integrated optical properties" in the legend**

Thank you, this will be corrected along the lines suggested, from «near surface concentration values» to «near surface concentration and column integrated optical parameter values.

**• page 62: indicate meaning of AWNC, FREQL, AREL. CDNUMC non needed in legend.**

CDNUMC is the model variable name for what we in the manuscript denote CDNCcol, and we can removed from the table caption as requested. The first 3 variable names are perhaps not needed in the legend either, but they indicate how we calculate the in-cloud parameters from the variables in the model output. To describe this better without degrading the readability, we suggest to first simplify the caption text to:

«Table 11: All-sky and clear-sky aerosol optical depth (OD) and absorptive optical depth (ABS) at 550 nm, Liquid Water Path (LWP), in-cloud cloud droplet number concentrations (CDNC)* and effective cloud droplet radius (Reffl)** at 860 hPa (model layer 24), and Ice Water Path (IWP). Also shown are the column integrated CDNC (CDNCcol) and Ice Crystal Number Concentration values (ICNCcol, calculated as part of the post-processing).»

and instead add details about the calculation of in-cloud parameters as footnotes to the table:

*CDNC is calculated as the average cloud water concentration AWNC (a grid average multiplied with the fractional occurence of liquid at each time step), divided by the fractional occurence of liquid, FREQL

**Reffl is calculated as the average froplet effective radius AREL (a grid average multiplied with the fractional occurence of liquid at each time step), divided by the fractional occurence of liquid, FREQL

**• page 66: meaning of etax1000? please add a couple of levels in the stratosphere**

All model levels are already plotted, but I assume you mean a couple of extra tick marks and tick lables: this will be added to the figure. The meaning of etax1000 is the model (hybrid coordinate) eta level multiplied by 1000, which will be added to the figure caption at the end of the first sentence, as «(eta x 1000 is the model hybrid coordinate eta level multiplied by 1000)».

**• page 71: I believe ERF ari are shown; it would be clearer to indicate this on top of the figures and in the legend**

The terminology used in Fig. 9 is in line with that of Ghan et al. (2013). We realize that this may be a bit confusing, since we also use the terms ERFari and ERFaci in the main text and in Table 10. Due to the slightly different definition of ERFari and ERFaci from that of IPCC AR5, which is explained in the Introduction, mentioned specifically in Table 10 and slightly discussed in Sect. 5, we would prefer to keep the Ghan terminology in Fig. 9 (as in Fig. 10). This is to avoid unnecessary confusion when comparing with other studies using the IPCC AR5 terminology.

**• finally, in general zooming in the figures degrades their quality; this needs to be corrected**

The original figures are probably better than they look when encapsulated in the pdf document used for the review process, but this will be checked again and dealt with, as far as the maximum allowed figure file size of 5MB (and 30MB in total) permits.

---

## Author Comment (AC3) · 24 Aug 2018

**Author's response to interactive comments by anonymous referee #2:**

The authors would like to thank the referee for very constructive review comments. Each of the review comments are repeated below in bold font, followed by our replies.

This manuscript describes the latest version of the OsloAero module used for modelling atmospheric aerosol in the aerosol-climate model CAM5.3-Oslo and earth system model NorESM1.2. OsloAero has a quite different formulation to most aerosol schemes, using a set of "background aerosol" tracers, whose prescribed size distributions and composition are then modified according to a second set of tracers which are tagged according to aerosol production processes. The resulting variations in size and composition, which are not restricted to e.g. a log-normal form, are then looked up in tables computed offline to determine the relevant optical and other properties of these mixtures. This is a novel approach to aerosol modelling which the authors have developed over a number of previous papers, adding distinct capabilities compared to the more widespread modal and sectional schemes. The present manuscript describes new developments for nucleation and secondary organic aerosol, as well as new online emission schemes for sea-salt, mineral dust and oceanic DMS and organics. There has also been an effort to ensure better consistency of parameters between the different components of the scheme, which is always welcome from a physical point of view. These are important additions, both in terms of updating this modelling framework with the latest understanding of physical processes, and for tighter coupling in the earth system context. A good basic set of evaluation plots and metrics are included, though the overall results compared to the previous CAM4-Oslo are somewhat mixed, with some biases and errors improved but others becoming larger; nevertheless this is a well-presented paper documenting a significant advance in the physical and chemical capabilities of the model and I would recommend it for publication in GMD subject to the minor comments below:

p.3, line 28 – p.4, line 10 This section is quite programmatic in terms of discussing the evolution of the projects to which OsloAero is related; I would consider whether all of this is relevant to a model description paper.

The first part of this text, p.3, line 28 – p.4, line 2, explains how CAM5.3-Oslo (or NorESM1.2) relates to CAM5.3 (CESM1.2), as well as to the online and offline parts of the aerosol schemes, OsloAero5.3 and AeroTab5.3, which we think is an important part of the description of the model, and being of high relevance also for potential users of the model. The remaining part links the present model version, already being used as a pre-CMIP6 version (in the CRESCENDO project, not mentioned in the text since that at least may be considered too project centered) of the final model versions planned to be used in CMIP6. We think this is of relevance for the reader, especially when seen together with the subsequent lines (p.4, linea 9 - 11) which in a similar way mention the participation of NorESM1 in CMIP5. We advocate for retaining this text as it is, for the sake of completeness. We will instead remove the last sentence of this paragraph (on lines 11-13) about NorESM1 participation in PDRMIP, since this is not imortant for the history or description of CAM5.3-Oslo. Two papers in the reference list will therefore also be removed: Samset et al. (2016) and Myhre et al. (2017).

p.4, line 27 "same method of aerosol activation as Liu et al. (2012)": is this the primary reference for the parameterisation in use, or does it describe a particular implementation of a well-known parameterisation (e.g. Abdul-Razzak and Ghan, 2000) which should also be cited here.

This was orginally meant to be written in more general terms, not only to concern the activation treatment, which indeed follows Abdul-Razzak and Ghan (2000) (referred to later, on the last line of p. 12). We suggest to rewrite the sentence starting at line 27 as follows:

«OsloAero5.3, as it is implemented in CAM5.3, applies the same method of aerosol activation (Abdul-Razzak and Ghan, 2000) and transport of and transition between aerosols in interstitial and cloud-phase as in Liu et al. (2012), with the simplifications proposed by Ghan and Easter (2006) that cloud-borne aerosols are not advected, except by vertical turbulent mixing.»

**p.6, line 4 It's not clear either why a portion of the modifying tracer is redistributed to the background ones, or the basis on which the amount redistributed is calculated; some explanation here is needed.**

The confusion seems to be due to a misunderstanding. What we are trying to explain is that the mass from the modifying tracers are distributed onto the modes (or mixtures) which initially only are constituted by the background aerosol tracers. Hence, no redistribution from modifying to background tracers is being performed at any point. While the tracers operate in «chemistry space», the mixtures (containg mass from various tracers) are treated separately in «physics space», if we can put it that way. How this is done quantitatively is outlined in the subsequent paragraph. Aiming to clarify this point, we suggest to rewrite this sentence (from line 4) to:

«OsloAero calculates how much of each "modifying" tracer should be distributed onto each of the background modes (thus forming mixtures of mass from the various tracers) within a time-step.».

**p.6, lines 12–13 In what way are these sized "augmented to take into account atmospheric growth"? This is a rather vague description.**

The sizes are increased (for mixtures 1-4 in Table 2) to typical sizes in the lower troposphere, which are rough estimates (prescribed values) originally from OsloAero4.0 (Kirkevåg et al., 2013), but with minor adjustments based on table look-ups of median radii (for the log-normal fits to modified size-distributions) from earlier CAM5.3-Oslo model simulations. To make this more clear, we suggest to add a second row with these modified sizes in Table 2, and rewrite the sentence on lines 11-16 (moving some of the detailed size info on lines 15 and 16 to the Table instead) to:

«First, during atmospheric transport the background tracers are assigned typical tropospheric dry sizes (i.e., the sizes at the point of emission, augmented to take into account atmospheric growth for the finest particles, mixtures no. 1 - 4 in Table 2). The size-modifying tracers are also assigned prescribed sizes (Table 2). Calculation of sizes (for dry deposition) after hygroscopic growth is as in OsloAero4.0 (K13).»

The «augmented» dry sizes (as NMR) for mixture 1, 2 and 4, to be listed in Table 2, are 0.025, 0.025 and 0.06  $\mu m$ , respectively.

In addition to the above changes in the main text, we modify the fourth sentence in the caption of Table 2 slightly to

«The initial number median dry radius (NMR) and standard deviation (SIGMA) of each background mode are listed in the second and third column.», and follow up with:

«Also listed (with numbers in brackets) are the prescribed dry NMR values assumed during transport (including atmospheric growth), for the finest particle mixtures (no. 1, 2 and 4). For other mixtures, the dry sizes of transported tracers are assumed to be identical to the initial sizes.»

Similarly, we add at the end of the table caption: «Assumed dry size parameters for the size-modifying tracers during transport are NMR = 0.04  $\mu$ m and SIGMA = 1.8 for SO4\_A1, and NMR = 0.1  $\mu$ m and SIGMA = 1.59 for SO4\_A2, SO4\_AC, OM\_AC, BC\_AC and SOA\_A1."

**p.6, lines 13–16 A little more explanation of what is meant by lumping the size-modifying tracers would be good, as well as the meaning of the modal size parameters of these modifying tracers, if these are not particles in their own right, but act to adjust the sizes of the background modes. (This may be explained further in earlier papers e.g. that labelled K13, but should ideally be self-explanatory in this manuscript.)**

The «lumping of size-modifying tracers (with respect to transport and deposition)» alludes to the fact that, e.g., sulfate condensate has only one tracer even though it is used in (distributed onto) several background modes to form mixtures, as shown in Table 2. But the different size-modifying tracers are here not lumped together to even fewer tracers, so this expression is a bit misleading. We have therefore chosen to change the text to that above, in the reply to the previous remark.

The modal size parameters were already defined in the same sentence, as «number median radius (NMR) and standard deviation (SIGMA)», but this has now been moved to the caption of Table 2 and rephrased (to size parameters). These tracers, as transported tracers in the model, behave as if they were «particles in their own right». Only in the microphysics part of the model (through use of look-up tables) do they adjust the sizes of the backround modes. To make this more clear in the text, we suggest to add the following sentence at line 15, just before the sentence starting with «Secondly»:

«In the parts of OsloAero5.3 which deal with aerosol chemistry, transport and dry deposition (the aerosol life-cycle scheme), both the background tracers and size-modifying tracers are treated as if they were particles.»

**p.27–29 I assume "lost from the model" means that monoterpene and isoprene do not exist as tracers in the model, but only as "transient" species near the surface within a given model time step?**

Both Isoprene and Monoterpene exist as gas tracers in the model, see e.g. Fig. 1. What we mean here on p. 7, line 27-29, is that the fractions of these gases which do not form SOA gas (85% or 95% in R1 – R5), are no longer tracked and taken into account, as they are assumed not to produce SOA aerosols. This (that they are not just transient species) can also be seen from Karset et al. (2018), Table 4., where their lifetimes for present-day conditions are estimated to 3.2 and 2.6 days, respectively. To clarify this in the text, we suggest to rewrite this sentence to

«The fractions of monoterpene and isoprene which do not react to form SOA gas in (R1) - (R6) are not taken into account, assuming that they form other gas or aerosol products which we do not track in the model.»

To make the list of aerosol and gas tracers complete and easy to find in the manuscript, we will also list of the 8 transported gas tracers at the end of the caption of Table 1:

«The aerosol precursor and oxidant gas tracers transported by the model are:  $SO_2$ ,  $H_2SO_4$ , DMS, isoprene, monoterpene, SOAG\_LV, SOAG\_SV, and  $H_2O_2$ .»

**p.11, line 19 This should be "... scheme is different from that..." or "... schemes are different from those..." for singular/plural consistency.**

Thank you, we will correct this to «...scheme is different from that...»

**p.14, line 8 It would be good to recap the emission sources here, even if they haven't changed from the previous paper.**

The exact meaning of this comment is a bit unclear, but we suggest to modify and add to this paragraph (on lines 6-9) as follows:

«For aerosol and precursors not mentioned above, as in K13, the emissions are taken from the IPCC AR5/CMIP6 (Lamarque et al., 2010) for the years 2000 (for simplicity called present day, PD), and 1850 (preindustrial, PI) conditions. The emissions and their vertical distribution are essentially the same as those used by Liu et al. (2012): the IPCC AR5 emission data set includes anthropogenic emissions for primary aerosol species OC and BC, as well as the precursor gas SO2. We assume that 2.5% of the sulfur emissions are emitted directly as primary sulfate aerosols, and the rest as SO2. Anthropogenic emissions are defined as originating from industrial, energy, transportation, domestic and agriculture activity sectors.»

**p.15, line 8 What is the height of the model top with 30 levels?**

We may here add the following extra nformation (as a reply also to referee#1):

«In hybrid sigma pressure coordinates, the uppermost eta level mid (or top of the level) value is 3.64 (2.26), and for the lowermost level it is 992.56 (985.11). The number of layers below approximately 1km and 2km height a.s.l. are 5 and 8, respectively.»

**p.15, line 9 Clarify that "microphysical schemes" here refers to cloud and precipitation rather than aerosol, if that is the case. Also, a brief description of the nature of these schemes (single/double-moment, what is prognostic etc.) would be welcome to provide context for aerosol-cloud interactions in this model.**

Yes, thank you, we here mean cloud and precipitation microphysical schemes. To add a brief description of the two schemes and the main difference between them, we suggest to rewrite and add to the sentence on lines 9-11 as follows:

«CAM5.3, and therefore also CAM5.3-Oslo, has two choices for stratiform microphysical cloud schemes: MG1.0 (Morrison and Gettelman, 2008) and MG1.5 (Gettelman and Morrison, 2015). Both are double-moment (i.e., mass and number predicting) bulk cloud microphysics schemes with prognostic cloud droplet and cloud ice mass mixing ratios and number concentrations. MG1.5 is an update of the original formulation MG1, where the location for updating prognostic droplet number mixing ratios with the tendency from droplet activation has been moved to the beginning of the scheme. We have in this study used MG1.5.»

p.15, line 24 A reference for the nudging technique would be welcome (e.g. Jeuken et al., 1996, 10.1029/96JD01218 or equivalents in other models).

The references for the nudging technique applied in CAM5 are Koopermann et al. (2012) and Zhang et al. (2014). These papers are already mentioned a bit later in Sect. 3, but we may add the references also at the end of the sentence at lines 24-26, p. 15:

«... using a relaxation time scale of six hours (Koopermann et al., 2012; Zhang et al., 2014).»

**p.16, line 18 Is there a reason why CAM4-Oslo couldn't be tested at the higher resolution, or CAM5.3-Oslo at the lower one, in order to assess the impact of the resolution change separately from the actual model updates?**

This was not originally done with the present model version and in this study, but two unnudged AMIP simulations (PI and PD) with 2° resolution have later been run to help answer this review comment. The main focus of this study is to document the changes in aerosol treatment from CAM4-Oslo to CAM5.3-Oslo, together with the comparison of their performance, while the effort to understand the differences in simulated results are more of a spin-off from the main study (not mentioned in the abstract). We will therefore not include the results of the 2° simulations (hereafter referred to as AMIP2degPD and AMIP2degPI) in the tables and figures of the manuscript, which would mean an immense amount of extra work, especially when it comes to the AeroCom intercomparison with other models and with maesurements (which are regularly updated on the aerocom.met.no server). We instead add comments on the effect of differences due to resolution some places in the manuscript where differences between the two model versions are being discussed (the basis diagnostics for this may be found here, together with diagnostics from the other simulations: http://ns2345k.web.sigma2.no/nudged\_NorESM\_c12/7310AMIP20002nAMIP\_PI\_wPDoxi\_vs \_AMIP2deg\_PDnPI/ModIvsModII.htm):

At the end of the paragraph on p. 17, line 2:

«An additional test simulation with CAM5.3-Oslo with AMIP PD set-up and 2° resolution show that the effect of increased resolution (to 1°) on DMS emissions and lifetime alone is only about 5% and 0.2%, respectively (not shown).»

At the end of the paragraph on p. 17, line 21:

«The additional 2° test simulation with CAM5.3-Oslo reveals that the effect of increased resolution on vertical profiles is very small compared to the differences between the two model versions, for all species (not shown).»

At the end of p. 17, line 32:

«The additional 2° test simulation reveals that the effect of increased resolution on the lifetime of BC is only about 0.3% (not shown).»

At the end of the paragraph on p. 18 line 11:

«The additional 2° test simulation (note that this by default set-up has a slightly different cloud tuning) reveals that the effect of increased resolution on LWP and on total (low) cloud cover is small compared to the differences between the two model versions, only about 1% and -1% (-3%), respectively (not shown).»

At the end of the paragraph on p. 18, line 17:

«The effect of increased resolution from 2° to 1° is here found to be 11% for the emissions (due to stronger winds), 9% for the burden, and only -2% for the lifetime (not shown).»

At the end of the paragraph on p. 18, line 24:

«The additional 2° test simulation reveals that the effect of increased resolution on the OM lifetime is only about 1% (not shown).»

At the end of p. 18, line 32: «The additional 2° test simulation reveals that the effect of increased resolution on the SO2 lifetime is only about -0.4% (not shown).»

At the end of the paragraph on p. 19, line 12: «The additional 2° test simulation reveals that the effect of increased resolution on the mineral dust lifetime is only about 2% (not shown).»

Since the above differeces due to increased resolution alone are so small, the impacts on effective radiative forcing are also small, compared to other changes from CAM4-Oslo to CAM5.3-Oslo. E.g., the SW cloud radiative forcing at TOA from AMIP2degPD minus AMIP2degPI is estimated to be -1.47 W/m2, compared to -1.45 W/m2 in AMIP\_PD – AMIP\_PI (Fig. 10). We therefore believe that the above additions to the text are sufficient, especially given that the weight on the parts of the mansucript where understanding differences in model results between the two models to some degree are discussed, is small (not mentioned in the abstract).

**p.16, line 25 If the DMS burden is doubled compared to the earlier version, are there some observations which could be cited here to indicate which is more realistic, or is the uncertainty even larger than this?**

The text should here have read «emissions», not «column burdens», and the background for these changes (with references) are discussed directly below. The text will here be corrected to:

«The result of the change in DMS emission parameterization described in Sect. 2.4 is an almost doubled DMS emission (34-35 Tg S yr-1) compared to the 18.1 Tg S yr-1 found in K13,...».

The increase in column burden (for which we do not have any obersvations) is much smaller (see Table 1), going from 0.12 Tg S in CAM4-Oslo to 0.14 Tg S in CAM5.3-Oslo. For completeness, we add at the end of the paragraph, on line 3, p. 17 (after the additional text concerning model resolution), as well as including the new reference under the «References» section:

«Note also that the increase in column burden from CAM4-Oslo to CAM5.3-Oslo is much smaller than the increase in emissions (see Table 1), going from 0.12 to 0.14 Tg S. These both lie well within the range of estimates (0.015 - 0.17 Tg S) from other model studies reported by Liu et al. (2007), see their Table 1.»

**p.17, line 29–30 The attribution of upper-tropospheric excess to the treatment of convective processes based on comparisons to HIPPO was also made by Kipling et al. (2013, 10.5194/acp-13-59690-2013 in the context of another model.**

Thank you. We here suggest to change the text as follows, and include the new reference in «References»:

«This is probably related to the way aerosols are transported and scavenged in deep convective clouds in the model (see e.g. Kipling et al., 2013; 2016).»

The latter reference is already used elsewhere in the text.

**p.27, lines 8–12 The rationale for using the grid-box-mean RH and weighting by clearsky fraction, rather than the more common approach is not clear.**

Different models make use of different assumptions about what can be considered representative for optical properties from observations or remotely retrievals under clear-sky conditions (such as those from AERONET), and it is not obvious to us which ones are the most accurate. We would argue that the assumed RH for hygroscopic growth is not the only relevant parameter, since aerosol optical properties also depend on the general abundance of aerosol, its size distribution and chemical composition, which again have links to the cloud cover (and precipitation). Although using the clear-sky RH fraction and not taking into account the actual cloud cover is the most common approach, it is not the only approach applied. The question of how AeroCom modellers calculate clear-sky optical properties was taken up by the AeroCom community during phase II of the project, which resulted in an attempt of documentation via a wiki page: https://wiki.met.no/aerocom/optical properties. Among the 10 AeroCom model versions (in addition to our own) which are both appearing in the wiki and in Tables 6-8 in the present study (GISS-MATRIX, GISS-modelE, GOCART-v4Ed, HADGEM2-ES, INCA, MPIHAM\_V2\_KZ, OsloCTM-v2, SPRINTARS-v384, and TM5.V3), only one estimates clear-sky in a similar way as we do, i.e. by using weights based on total 2D cloud cover but all-sky RH for hygroscopic growth of the aerosol: SPRINTARS\_v384 averages only AOD for time steps and grid points with CLDTOT<0.2, using all-sky RH in these almost cloudfree cases. GISS-MATRIX, GMI and GOCART\_v4E average all-sky AOD (no cloud cover weighting), also based on growth factors from all-sky RH. GISS-MATRIX also provide clear-sky optics, but this has not been used at aerocom.met.no. Although not very clear from the wiki, the rest of the models seem to estimate AOD (tacitly meant to be a clear-sky parameter) for all cloud conditions, but based on hygroscopic particle growth using the clear-sky RH value. As a short summary of this (also as reply to a related comment by referee #1), we suggest to rewrite parts of the caption of Table 6 (lines 8-11) from

«Optics diagnostics listed for the AP2 and AP3 models are mostly clear-sky values, in the sense that the clear-sky humidity of the grid cell is used for calculating the hygroscopic swelling. CAM4-Oslo and CAM5.3-Oslo compute all-sky optical properties using the average humidity of the grid cell. Clear-sky (CS) properties are represented by a cloud fraction weighted average of the all-sky properties.»

**to**

«Optics diagnostics listed for most of the AP2 and AP3 models (exact number is not available) are clear-sky values, in the sense that the clear-sky humidity of the grid cell is used for calculating hygroscopic swelling of the aerosol. Information about this for 11 of the AP2 models included here, plus some others, may be found at

https://wiki.met.no/aerocom/optical\_properties. CAM4-Oslo and CAM5.3-Oslo compute allsky optical properties using the average humidity (RH) of the grid cell. Clear-sky (CS) properties are instead represented by a 2D cloud-free fraction weighted average of the allsky properties. Only a few other AeroCom models follow a similar clear-sky optics definition, and the optics data submitted to AeroCom for a few of the models are all-sky values both in terms of cloud conditions and RH for hygroscopic growth.»

**p.37, line 30 NUDGE\_PI appears twice; one should be NUDGE\_PD.**

Thank you, this will be corrected.

**p.38, line 9 Who should be contacted to obtain a user agreement for access to the code and data?**

The contact person has traditionally been Øyvind Seland, but we have a more general mailing list which should be used: noresm-ncc@met.no. Based also on another interactive comment, we suggest to rewrite the second line of the Code and data availability paragraph to:

«The source code for CAM5.3-Oslo is part of a restricted NorESM2 pre-release and stored within the private github NorESM repository

(https://github.com/metno/noresm/tree/NorESM1.2-v1.0.0). Access to the code and simulation output data produced in this study can be obtained upon reasonable request to noresm-ncc@met.no and requires entering a NorESM Climate modeling Consortium (NCC) user agreement.»

**Tables Many of the tables contain a large number of numerical values, either mass budget terms or statistics. While the actual numbers may be useful for reference, summarising these in charts would probably be much easier for the reader to understand at a glance.**

Since the tabulated values are still needed for reference in the main text and are valuable for inter-comparison with past and future studies, such charts would cause a drastic increase in the number of figures and therefore the total length of the mansucript, which is already quite long. Each table lists quite many model variables (and observed variables for some tables), meaning that each of the tables would require several fugures/charts in order to avoid overbusy plots with multiple axes. We therefore strongly suggest to keep the tables as they are, but with one small change to make it easier to spot the largest biases at a glance in Tables 5 and 6: Using bold fonts for NMB numbers with absolute values larger than (e.g.) 50%.

**Reference:**

Liu, X., Penner, J. E., Das, B., Bergmann, D., Rodriguez, J. M., Strahan. S., Wang, M., and Feng, Y.: Uncertainties in global aerosol simulations: Assessment using three meteorological data sets, J. Geophys. Res., 112, D11212, doi:10.1029/2006JD008216, 2007.

---

## Author Response (AR1)

**Authors' response to interactive comments**

The authors' replies to interactive comments by the Executive Editor and by anonymous referee #1 and referee #2 have already been posted under "interactive discussions", see the respective supplement files (pdf) there. The only
5    other changes we have made in the revised manuscript are:

-    We have updated and corrected the affiliation of one of the co-authors, Risto Makkonen.
-    We have updated the reference for Boy et al. (2018), which in the previous manuscript version was "in preparation" but now exists as an ACP discussion paper.

[revised manuscript text omitted]

---

## Author Response (AR2)

The authors would like to thank the Topical editpr again for very constructive comments. Each of the comments are repeated below in bold font, followed by our replies.

**Topical Editor Decision: Publish subject to technical corrections (30 Aug 2018) by Graham Mann**

5 **Comments to the Author:**

**I have read through the authors response to the reviewers and confirm that I agree their replies to the comments are accurate and that I also approve the suggested revisions that have been made to the manuscript, which in response to the helpfully thorough reviews have made some significant improvements**

10 **to the explanations in the manuscript.**

**There are however several minor technical issues which I list below, and once these revisions have been made, I can confirm my approval for the manuscript to proceed to publication in the main Atmospheric Chemistry and Physics journal.**

**The required technical revisions are listed below numerically:**

**1) Re: reviewer 1's request for detail on the vertical resolution, the suggested addition is fine, but are the units for the values given there in hPa? (I.e. is the mid-point of the lowest level at 992.56 hPa [not**

20 **withstanding pressure variations from the terrain-following element of the hybrid sigma-pressure co-ordinate system]?). If so please add that unit to the eta values given there.**

Yes, thank you, the unit should here be hPa, now added to the numbers in the text referred to.

**2) Re: reviewer 1's comment on the "self-nudging", the first of the two suggested added sentences here is**

25 **fine, but in the 2nd sentence please rephrase the last half of the sentence. In particular, the phrase "which in turn leads to more confidence in, e.g. estimates of effective radiative forcing, but is less useful for the purposes of comparing modeled aerosol properties with observations." Although this explains the issue re: the distinction between nudging to "best estimate fields" from meteorological re-analyses, and nudging to some previously archived sequential instantaneous fields (e.g. 3-hourly) from some reference simulation with**

30 **the same model) the phrase "which in turn lends more confidence in e.g. estimates of effective radiative forcing" needs to be revised. I think the main point is not that the self-nudging gives more confidence re: radiative forcings necessarily (although it does for comparisons to observations), rather the approach suppresses (or over-writes) fast feedbacks in the simulations, which then derive "radiative forcing" rather than "effective radiative forcing" (the latter term including the fast feedback effects but the former term**

35 **explaining the effects have been deliberately excluded as part of the experiment-design).**

We thought of the suggested formulation "which in turn lends more confidence in e.g. estimates of effective radiative forcing" as a natural consequence of the first part, that self-nudging is more true to the model's own

behaviour. However, we did not mean that self-nudging gives more realistic/true ERF estimates, just that it can give ERF estimates that are more similar to estimates from un-nudged simulations than what we can expect when nudging to reanalyses or (in general) data from other models. So, we agree that «lends more confidence» may indeed be misleading here. We suggest to remove that part of the sentence and rewrite the rest, to:

5   «When nudging to an atmospheric circulation produced by the model itself (self-nudging) instead of using data from reanalysis (such as the ERA data), the circulation mean and variability characteristics are less affected, resulting in ERF estimates which are more consistent with the model's own, innate behaviour. However, since the circulation variability is here not «synchronised» with the observed variability of a specific time period, self-nudging does not facilitate comparison of modeled aerosol properties with observations for that time period.»

As stated in the last paragraph of section 3, we do not nudge temperature, and our intention is not to estimate aerosol instantaneous radiative forcing (which ideally requires double calls to radiation and cloud physics, as applied e.g. in CAM4-Oslo): fast feedback processes are still allowed to modify the temperature and are thus not overwritten (possibly slightly damped if large scale circulation feedbacks – which indeed are overwritten by

15  the nudging – play a role). In this study, the main purpose of the nudging is to synchronise the circulation variability and resulting temperature/humidity and particle transport variations in order to reduce the statistical uncertainty (in our estimates) that arises from such variability.

**3) On the next reviewer 1 suggested addtion (page 33 line 7), just a terminological point to please change**

20  **"ERF ACI" to "ERFaci", the lower-case aci rather than ACI being the recommended terminology from the radiative forcing of the IPCC 5th assessment report (Myhre et al., 2013). Please also replace other instances of "ERF ACI" and of "ERF ARI" with "ERFaci" and "ERFari" respectively.**

Thank you, we have now replaced all instances of  "ERF ACI" and of "ERF ARI" with "ERFaci" and "ERFari", respectively.

**4) Re: the added text in reply to reviewer 1's comment page 21 line 3, that text is fine but since that statement refers only to SO2 (some of the aerosol metrics indicating an apparent worsening of skill with the addition of the nudging) please add "SO2" after "climatological averaged".**

We agree. This is now changed in the text as «For the climatological averaged $SO_2$ data, the…»

**5) Re: reviewer 1's comment re: the BC in the UT (page 34, line 13), please add "BC compared to" between "overestimate" and "the aircraft measurements" so that is also more specific, also replacing "aloft" with "in the free troposphere" for the same reason. Please also correct the typo "participatig" with "participating".**

Thank you for pointing this out. The text now instead reads «Note that there is a general tendency for models
35  participating in the AeroCom project to overestimate BC compared to the aircraft measurements in the free troposphere in remote regions (Samset et al., 2014).».

**6) That's great re: the reply to reviewer 1's comment re: the query on the clear-sky flux info from the models, but when I checked that link https://wiki.met.no/aerocom/optical_properties that seemed to be a broken link or typo in the http-address (or maybe the site was down for maintenance at the time I checked?)**

There could have been a temporary glitch, which may occasionally happen due to server down-time. I am not able to verify whether the server was down during parts of the day August 27'th, but the given URL is the correct one, so we hope you will be able to try it again. This wiki page has been accessible for years already.

**7) Re: reviewer 1's question about page 62, please correct spelling of "froplet" and add the word "cloud" before "droplet effective radius" so it's absolutely clear you mean cloud droplet reff not aerosol particle reff.**

Thank you, this is now updated as suggested.

**8) In the reply to reviewer 1's question about page 71, I don't think this reply is correct because the wording "direct radiative forcing" and "cloud radiative forcing" are still shown in Figures 9 and 10, and in the caption. Please correct these to "ERFari/RFari" for Figure 9 and "ERFaci/RFaci" for Figure 10 in both the image and the caption.**

In our reply to reviewer #1 we referred to Ghan et al. (2013), which should have been Ghan (2013). There the recommended decomposition of aerosol radiative forcing (in terms of ERF, as I understand it) is as follows: direct radiative forcing, cloud radiative forcing, and surface albedo forcing. This is in our manuscript described in the first paragraph of Sect. 5. ERF is discussed in the introduction of Ghan (2013), but the terms ERFari and ERFaci are not mentioned. The reason why we suggested to stick to the «Ghan terminology», is that the IPCC terms ERFari and ERFaci have a slightly different meaning, as also explained in the beginning of Sect. 5 and pointed out specifically in a footnote to Table 10, where our ERF values are compared to those from IPCC AR5: «*the semi-direct effect is here embedded in the ERF ACI term (Ghan, 2013), not in ERF ARI as in the IPCC AR5 estimates». Admittantly, here in the footnote we have mixed the two terminologies.

We therefore suggest the follwing changes based on the comments of the referee and the topical editor, while still informing the reader about the distinction in terminology.

1) We change the headlines to "SW ERFari" and «LW ERFari» in Figure 9 and "SW ERFaci" and «LW ERFaci» in Figure 10 (while keeping the globally averaged values for each panel, as before).

2) We then rewrite the Figure captions as follows:

«**Figure 9:** Shortwave (SW, top panels) and longwave (LW, bottom panels) ERFari at top of the atmosphere (TOA) for the simulations NUDGE_PD – NUDGE_PI (left panels) and AMIP_PD – AMIP_PI (right panels). Note the different color scales. Note also that the semi-direct effect is here not included, since ERFari in this study corresponds to the «direct radiative forcing» component in Ghan (2013).»

«**Figure 10:** Shortwave (SW, top panels) and longwave (LW, bottom panels) ERFaci at top of the atmosphere (TOA) for the simulations NUDGE_PD – NUDGE_PI (left panels) and AMIP_PD – AMIP_PI (right panels). Note

that the semi-direct effect is here embedded in the ERFaci term, which corresponds to the «cloud radiative forcing» component in Ghan (2013).»

**9) In the reply to reviewer 2's suggestion re: p6 lines 13-16, please revise the existing text to clarify what is meant by "the background tracers and size-modifying tracers are treated as if they were particles". Is this referring to the parts of OsloAero5.3 (or another module in the host model) that deals with the oxidation/conversion of the gas phase tracers (i.e. the aerosol precursor gases)?**

To keep in mind what we are talking about, the full sentence inserted as a reply to reviewer #2 was:

«In the parts of OsloAero5.3 which deal with aerosol chemistry, transport and dry deposition (the aerosol life-cycle scheme), both the background tracers and size-modifying tracers are treated as if they were particles.»

To be very specific, this refers only to the part of OsloAero5.3 which calculates the dry deposition, not the transport or chemistry. I usually think of these as a whole, the so-called life-cycling module, which is the background for the previously suggested text version. Note also that the background tracers and size modifying tracers (mass from condensation, coagulation and cloud processing, i.e. sulfate mass produced in aqueous phase reactions ending up as interstitial aerosol after evaporation of cloud droplets) both pertain to aerosols, not gases. To clarify, we therefore suggest to rewrite this and the prevous sentence as follows:

«First, during atmospheric transport the background aerosol tracers are assigned typical tropospheric dry sizes (i.e., the sizes at the point of emission, augmented to take into account atmospheric growth for the finest particles, mixtures no. 1 - 4 in Table 2). The size-modifying aerosol tracers are also assigned prescribed sizes (see Table 2). Their respective sizes after hygroscopic growth, calculated as in OsloAero4.0 (K13), are eventually used for calculation of dry deposition, where both types of aerosol tracers are treated as if they were separate particles.»

For dry deposition the size-modifying aerosol mass should in principle be removed with the same rate as the background aerosol it is living on. However, as these size-modifying aerosols represent mass living on various modes/mixtures simultaneously (lumped approach), we attribute representative aerosol size parameters to the size-modifying aerosol tracers to be used in the dry deposition calculation.

**And is that what is meant by "aerosol chemistry" (i.e. shorthand for "aerosol precursor chemistry")? If so, suggest to change "aerosol chemistry" to "oxidations", add "of aerosol precursor gases" and replace "(the aerosol lifecycle scheme)" with "(the gas phase chemistry scheme)" or similar. Isn't that the key thing here re: the difference between gas-phase and aerosol-particle tracers here in the context of the "aerosol chemistry, transport and dry deposition" stated? Is that what you meant in terms of the tracers are treated as particles (i.e. are they given non-zero sedimentation velocity according to their particle size in the dry deposition routines [and surface area density and uptake coeff for the "aerosol chemistry"? i.e. is it actually heterogeneous chemistry you mean here when you say aerosol chemistry?]. Please can you re-word the suggested text-addition to make this clearer.**

The reference to chemistry was not relevant, please see our reply above.

**10) Re: reviewer 2's comment re: the isoprene and monoterpene reactions, suggest also to add mention that the chemistry follows offline oxidant approach, so the products from the chemical conversion of the second-reactants (the oxidants) do not need to be included in the products on the right-hand side of the chemical-equations.**

This is already mentioned on p. 8, line 7 (as requested by the topical editor already for the first GMDD version of the manuscript):

"We also note that, since the model uses the "offline-oxidant approach", reactions (R1) to (R6) need only to resolve one product (on the right-hand-side of the equations)."

We may elaborate a little further on this, along the lines suggested:

"We also note that, since the model uses the "offline-oxidant approach", reactions (R1) to (R6) need only to resolve one product, meaning that the products of the second-reactants (the oxidants) do not need to be included on the right-hand side of the chemical equations."

**11) Re: reviewer 2's comments re: the AR5 emissions (Lamarque et al., 2010) you have written "CMIP6" but you mean "CMIP5" here. Please correct.**

Yes, this should read CMIP5, and is now corrected.

**12) Re: reviewer 2's request for reference for the nudging, I'm wondering about the stated "relaxation time of 6 hours". I'm wondering if maybe you meant "frequency of 6 hours" in that the model relaxes back to (interpolations between?) different successive meteorological re-analysis field every 6 hours? If that is what was meant please replace "relaxation time" with "frequency"**
**as I think that's clearer what's meant. I guess the relaxation time must be on a shorter timescale than that?**

Concerning the statement that relaxation time should be on a shorter timescale: We could have agreed with this statement if we also nudged temperature and had the intention of overwriting fast feedback processes in order to estimate something closer to the instantaneous radiative forcing or to obtain a meteorology closer to observations, for even better facilitating the comparison of aerosol properties with observations.

As elaborated in the response to comment 2, this has however not been our objective. In addition to the possibility of comparing modeled properties with observations, we have sought estimates of the effective radiative forcing of anthropogenic aerosols, which may be calculated from relatively short nudged simulations (constraining the model's natural variability) using the recommended relaxation time of 6 hours, as described by both Kooperman et al. (2012) and Zhang et al (2014).

The nudging is applied every time step (i.e. half-hourly) in the model using observations that are read in at a 6-hourly frequency and then interpolated in time. The relaxation time informs about the nudging strength rather than the nudging frequency. For example, we could have chosen zero hours, in which case the model would "replace" the model fields with the interpolated observational fields.

5   We suggest to extend the text referred to, based on the above but without taking too much space (considering that we already have two references), as follows:

"The difference between the AMIP and the Nudged configuration is that the latter includes additional terms to the dynamical equations which push (nudge) the model meteorology towards the observed (or reanalyzed, read in 6 hourly and interpolated in time) meteorology, using a relaxation time of six hours (Koopermann et al., 2012;
10  Zhang et al., 2014).»

**13) Re: reviewer 2's request to explain the rationale for weighting the averaging of the "instantaneous-calculated AODs" by the clear-sky fraction rather than the recommended approach to calculate the hygroscopic-growth applying the clear-sky-RH rather than the gridbox-mean RH, OK, yes that response in**
15  **explaining this alternative approach. I see that, when calculating means, doing so with a weighting to one-minus-cloud-fraction will then make that mean-AOD be more representative of clear-sky conditions (and therefore minimise the effects from overestimated-aerosol-optical-depth due to elevated humidity in cloudy gridboxes). And the added text now explains that the CAM4-Oslo and CAMS5.3-Oslo diagnosed this differently than the model, so that is good, and clear. However, again -- I agree it would help in the reply to**
20  **reviewers comments to be able to refer to the discussion on the Wiki (although maybe contributors might I guess potentially object to that then being referenceable and citeable in this way for some other reason).**
**In any case the link there did not seem to work when I tried it on 27th Aug.**
**As per the other comment, please check this and add the corrected link (assuming the AeroCom team approve the link can be used in this way for the review without the consent of the contributors).**

25  The wiki web link has been checked and verified to be correct, see also our reply above. We would also like to include a personal communication reference to Michael Schulz, as the leader of the AeroCom project. Two of the sentences in the caption of Table 6 then become, after adding the personal communcation citation and rewriting the second sentence slightly:

[revised manuscript text omitted]

**Slettet:**

**Slettet:** anthropogenic cloud radiative forcing